# KV Cache Transform Coding for Compact Storage in LLM Inference

**Konrad Staniszewski**[1,2] **& Adrian Łańcucki**[1]
NVIDIA[1], University of Warsaw[2]
kstaniszewsk@nvidia.com

## Abstract

Serving large language models (LLMs) at scale necessitates efficient key-value (KV) cache management. KV caches can be reused across conversation turns via shared-prefix prompts that are common in iterative code editing and chat. However, stale caches consume scarce GPU memory, require offloading, or force recomputation. We present `kvtc`, a lightweight transform coder that compresses KV caches for compact on-GPU and off-GPU storage. Drawing on classical media compression, `kvtc` combines PCA-based feature decorrelation, adaptive quantization, and entropy coding. It requires only a brief initial calibration and leaves model parameters unchanged. By exploiting redundancies in KV caches, `kvtc` achieves up to 20× compression while maintaining reasoning and long-context accuracy, and 40× or higher for specific use cases. We test `kvtc` with Llama 3, Mistral NeMo, and R1-Qwen 2.5 models across benchmarks including AIME25, GSM8K, LiveCodeBench, LongBench, MATH-500, MMLU, Qasper and RULER. It consistently outperforms inference-time baselines such as token eviction, quantization, and SVD-based methods, while achieving higher compression ratios. These results support `kvtc` as a practical building block for memory-efficient LLM serving with reusable KV caches.

## 1 Introduction

Chat-based interfaces, commonly used for interacting with large language models (LLMs), enable users to iteratively refine answers across open-domain dialogues and specialized tasks, such as code generation (Chiang et al., 2024; Köpf et al., 2023). Each conversational turn extends the key–value (KV) cache associated with a conversation, storing hidden activations for every previous token. For modern Transformer models, this cache can easily occupy multiple gigabytes. As models scale up in size and reasoning capability, generating increasingly long reasoning chains (OpenAI et al., 2024), the KV cache footprint increases, posing a significant bottleneck for throughput and latency. During user turns, stale KV caches left on-chip occupy memory, which is needed for serving other users, yet ensure the fastest responses in the future. Conversely, caches could be discarded, incurring the cost of recomputation, or offloaded to CPU DRAM or local/network storage, leading to transfer overheads. This tension creates a latency–throughput dilemma in production systems and necessitates careful configuration.

Crucially, inference frameworks view the local KV caches as databases. Strategies such as block-level



Figure 1: The `kvtc` transform-coding pipeline. Features are linearly decorrelated via PCA, and the resulting coefficients are quantized using variable bit widths. The PCA basis $V$ is computed once on a calibration dataset and reused for all caches.

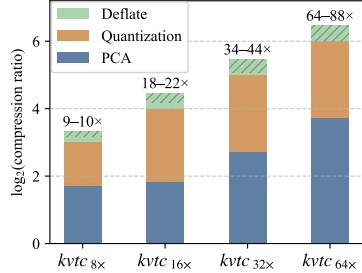

Figure 2: KV cache compression ratios contributed by parts of the `kvtc` pipeline for Llama 3.1 8B. DEFLATE's variability is marked with black stripes.

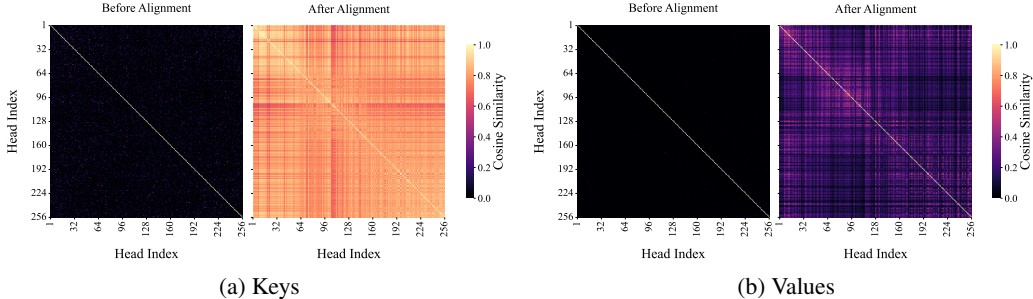

Figure 3: Cosine similarity before and after alignment between key (a) and value (b) heads calculated using Llama 3.1 8B on inputs from Qasper (Dasigi et al., 2021; Shaham et al., 2022). For each example, we calculate cosine similarity between all keys/values from the same position and then average across the batch. Orthonormal alignment matrices were produced using 20 samples from the RedPajama v2 (Weber et al., 2024).

paging and prefix sharing promote reuse of caches whenever prompt prefix matches (Kwon et al., 2023). Scaling LLM serving increasingly hinges on KV cache management and reuse (Liu et al., 2024b; Cheng et al., 2024; Yao et al., 2025), but current systems struggle to store, move, and refresh these caches efficiently. CacheGen (Liu et al., 2024b) compresses caches for transmission, offering at most $8.6\times$ KV cache reduction in comparison to a 16-bit baseline. SVDq (Yankun et al., 2025) and xKV (Chang et al., 2025) pursue low-rank compression during prefill, but both require calculation of per-prompt SVD. Long and frequently used prompts may justify investing more compute for offline training of corpus-specific caches (Eyuboglu et al., 2025).

Meanwhile, intensively studied KV cache compression methods, aimed at improving the runtime efficiency of autoregressive generation, offer interim measures to the cache retention problem (Yuan et al., 2024). Prior work hinges on observations that KV cache can be quantized (Frantar et al., 2023) sparsified (Liu et al., 2024d; Hooper et al., 2024), average-pooled (Nawrot et al., 2024), or shared between layers (Brandon et al., 2024); the cache itself is compressible (Yuan et al., 2024), and dimensions of keys and values for separate heads show a high degree of correlation (Zhang et al., 2023). For long contexts, these methods offer substantial throughput and latency improvements, by lowering KV cache sizes and thus the memory traffic during next token prediction. However, due to tight latency constraints, often coupled with refraining from modifying weights of the model, these techniques tend to be brittle (Tang et al., 2024), and accuracy degradation prohibits combining methods for compounded benefits. Finally, these methods seldom exploit the strong low-rank structure of KV tensors.

In this paper, we introduce `kvtc`: a simple yet powerful transform coding scheme, compressing KV caches for storage. Inspired by classical image codecs, it applies a learned orthonormal transform followed by channel-wise scalar quantization, which dynamically allocates bits, and entropy coding. The resulting bitstream is on average $20\times$ smaller than the original 16-bit one, while maintaining comparable accuracy. The method also exposes a smooth compression–accuracy trade-off, with $40\times$ or higher compression attainable at modest accuracy decrease. Thus, `kvtc` largely mitigates the problem of KV cache management: lowering the cost of its on-chip retention and the bandwidth required for offloading, without compromising interactive latency.

## 2 PRELIMINARIES

**KV Cache Structure** During decoding in autoregressive Transformers with multi-head self-attention, the keys and values produced for each processed token are cached to avoid recomputation. The collection of these tensors is the *KV cache*. For $l$ layers, $h$ heads, head dimension $d_{\text{head}}$ and sequence length $t$, a 16-bit KV cache occupies $(4\,l\,h\,d_{\text{head}}\,t)$ bytes.

Table 1: KV cache size in 16 bits for 1K tokens of context.

| Model | Size |
|---|---|
| Qwen 2.5 R1 1.5B | 28 MiB |
| Qwen 2.5 R1 7B | 56 MiB |
| Llama 3.1 8B | 128 MiB |
| Llama 3.3 70B Instruct | 320 MiB |
| Mistral NeMo 12B | 160 MiB |
| MN-Minitron 8B | 160 MiB |

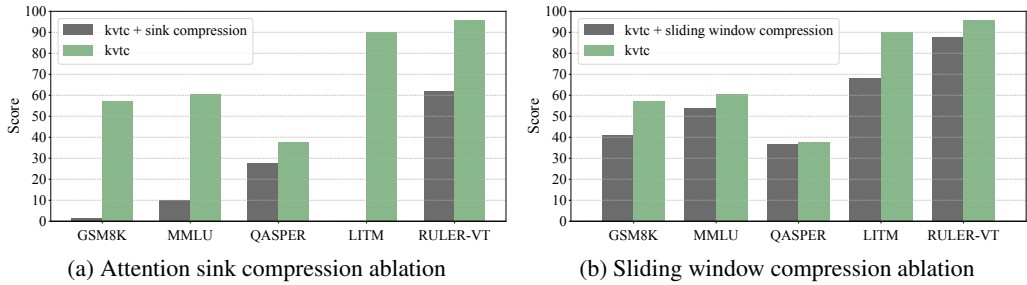

(a) Attention sink compression ablation          (b) Sliding window compression ablation

Figure 4: Ablation of `kvtc` with compression ratio $64\times$ on Llama 3.1 8B: (a) compression disabled for attention sink tokens; (b) compression disabled for the final 128 tokens. All other settings are fixed. Additional ablations are provided in Appendices B.3 and B.7.

Motivated by work on cross-layer KV cache sharing and compression (Brandon et al., 2024; Chang et al., 2025), we ask whether keys from different attention heads, and analogously values, lie in a shared latent space. To be more precise, we examine if it is possible to align key or value caches, produced by different attention heads, via linear transformations. Specifically, for each pair of attention heads $h_i, h_j$ in the model, we attempt to align their caches $K_i, K_j \in \mathbb{R}^{t \times d_{head}}$ with an orthogonal map found by solving the Procrustes problem (Gower & Dijksterhuis, 2004):

$$R^\star = \arg\min_R \|K_i - K_j R\|_F \quad \text{s.t. } R^\top R = I. \tag{1}$$

We then compute token-wise cosine similarity between $K_i$ and $K_j$ before alignment, and between $K_i$ and $K_j R^\star$ after alignment. We repeat the same procedure for values using $V_i, V_j$. Before alignment, inter-head cosine similarity is typically below 0.2. After orthogonal alignment, similarity increases substantially for keys and moderately for values (Figure 3). This pattern suggests that key heads largely inhabit a common subspace up to an orthogonal transformation; their dissimilarity before alignment likely stems from random initialization of key and value projection matrices. We note that for a data matrix $A \in \mathbb{R}^{n \times d}$, if $k$ directions suffice to explain all of the variance of $A$, then $k$ directions suffice to explain all of the variance of $B = [A, AR] \in \mathbb{R}^{n \times (d+d')}$, for $R \in \mathbb{R}^{d \times d'}$. This motivates our choice of PCA as a dimensionality reduction method.

Another motivation comes from the work on efficient attention kernels (Jiang et al., 2024). To be more precise, from the observation that different attention heads can show similar attention patterns. In a simplified setting without RoPE (Su et al., 2024), where keys are equal to queries and we assume the exact equality of dot products that create the attention patterns, the key spaces are equal up to an orthogonal transform by the uniqueness of Gram realizations (Horn & Johnson, 2013).

**Sliding Windows and Sink Tokens**   We avoid compressing both the $w$ most recent tokens and $s$ oldest tokens (attention sinks) due to their disproportionately high contribution to typical attention patterns (Jiang et al., 2024). In transform coding for vision and audio, bits are allocated to transform coefficients so that quantization induces minimal perceptual distortion. By loose analogy, the attention mass allocated to a token can be viewed as a proxy for its importance. In addition, when PCA is used to reduce the dimensionality of keys and values, the initial tokens yield higher reconstruction errors, as shown in Figure 6.

Foreshadowing the experiments described in Section 4, we have chosen $w = 128$ and $s = 4$ for evaluation. To illustrate their influence on accuracy, we ablate on their compression by setting either $w = 0$ or $s = 0$, as shown in Figure 4. These experiments show that compressing these tokens can significantly lower, or even entirely collapse the accuracy at high compression ratios.

**Multi-Turn Conversations**   Let a conversation be an ordered sequence

$$\mathcal{C} = ((x_0, y_0), (x_1, y_1), \dots),$$

where $x_t$ denotes the user (or system) input at turn $t$ and $y_t$ the generated reply. Generation of $y_t$ consists of a prefill pass, which produces the KV cache for all preceding tokens $(x_0, y_0, \dots, x_t)$, followed by iterative decoding, which generates the tokens of $y_t$ one at a time.

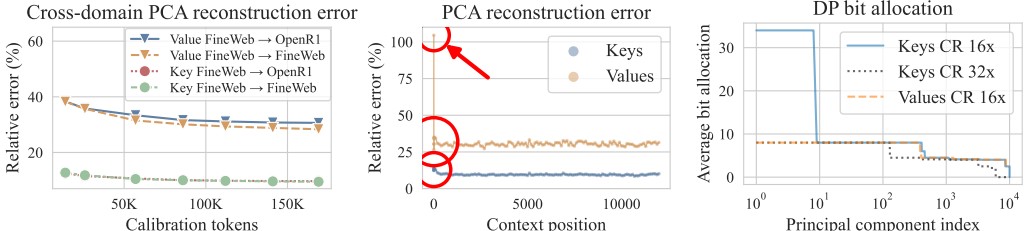

Figure 6: Calibration of Llama 3.1 8B with `kvtc`. **Left:** Reconstruction error as a function of the size of calibration set. The arrow $A \to B$ denotes fitting PCA on dataset A and calculating the error on B. **Middle:** The reconstruction error as a function of position in the context. The error is higher for the initial tokens. **Right:** Bit assignment computed via dynamic programming, counting in the per-group scaling factors.

When a new user prompt $x_i$ is received, the existing KV cache can be re-used and only newly added tokens have to be forwarded through the model, reducing computation and time-to-first-token (TTFT). However, if the cache has been deleted, the model must reprocess the entire conversation as a prompt, resulting in quadratic recomputation of attention across the input.

**KV Cache Management while Serving** Efficient LLM deployments often split prefill and decode across separate nodes due to their distinct performance profiles (Zhong et al., 2024), as shown in Figure 5. The prefill node produces the KV cache and transmits it to the decode node over a high-speed fabric, typically RDMA-capable such as InfiniBand or RoCE, minimizing latency and CPU overhead on the host. Both nodes may maintain tiered KV caches: GPU HBM (hot), CPU DRAM (warm), and NVMe/SSD (cold), managed with a retention policy. Long-term storage might require sending caches to a

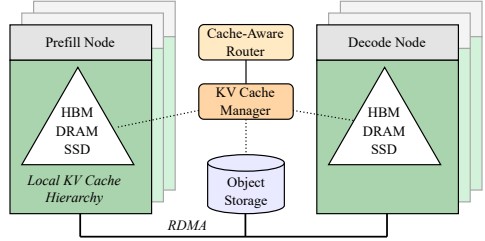

Figure 5: A high-level architecture of KV-cache-aware LLM serving environment.

remote location. Crucially, the decision to select a node for either prefill or decode of a certain input can be dictated by the already-held KV cache with a matching prefix. In such setups, KV cache transfers are typically the dominant cross-node traffic.

Why compress KV caches after the prefill or decode phase? Compression can: (i) extend the effective capacity of a KV cache database, and thus the lifetimes of caches, roughly in proportion to the compression ratio, and (ii) reduce network traffic. We discuss both angles below.

(i) Extending KV cache lifetimes increases cache hit rates in higher tiers (HBM/DRAM), often avoiding or substantially reducing prefill time for prompts with long, recently processed prefixes. For example, during a session with a coding assistant, a single 1,000-line file tokenized at $\sim 10$ tokens per line and processed by Llama 3.3 70B yields about 1.6 GiB of 8-bit KV cache. Subsequent conversation turns, or a few parallel conversations about this file, might reuse the corresponding cache. However, on a node which serves multiple clients, the volume of generated KV cache might shorten its hot/warm residency even at moderate batch sizes. A 20× lifetime extension via compression might determine whether a KV cache remains hot/warm until it becomes useful, or needs to be recomputed from scratch.

(ii) Prefill time scales as $\mathcal{O}(n^2)$ with prompt length and typically dominates transfer time. In addition, KV caches can be streamed by layer during prefill in order to further reduce TTFT (Qin et al., 2025). However, KV cache compression reduces memory traffic proportionally to the compression ratio, which might be critical when the network bandwidth is saturated and becomes the bottleneck.

## 3 METHOD

Our Key-Value Transform Coder (`kvtc`), shown in Figure 1, builds upon the transform-coding framework (Ahmed et al., 1974; Goyal, 2001), a widely adopted methodology for designing image and video compression algorithms such as JPEG (Joint Photographic Experts Group JPEG, 1994). It applies feature decorrelation by projecting onto an orthonormal basis matrix $V$ obtained via the singular value decomposition (SVD) of centered calibration data (i.e., principal component analysis, PCA). Quantization parameters are selected using a dynamic programming algorithm, and the resulting symbols are entropy-coded with DEFLATE (Wu, 2017). The three modes of operation of `kvtc` are:

- **Calibration** This step is performed only once for every model and compression ratio. During calibration, we obtain the key and value caches for the calibration dataset, calculate $\mathrm{SVD}(C-\mu) = U\Sigma V^\top$ where $\mu$ is the mean of the data, and store the projection matrices $V^\top$ (for key and value caches separately). Next, we run a dynamic programming algorithm that produces the optimal bit allocation for each of the principal components. This step is fast; for instance, calibration for a 12B model can be completed within 10 minutes on an H100 GPU (Appendix B.5). In Appendix B.14 we show that the additional amount of data stored per model is relatively small (2.4% of model parameters for Llama 3.3 70B). In Appendix B.10 we perform an ablation study on the number of layers over which we concatenate the KV caches.
- **Compression** Keys and values are compressed independently using the $(V, \mu)$ parameters and bit allocation obtained during calibration. Compression is performed between inference phases (e.g., after decoding or between prefill and decoding) and can be executed on either GPU (affecting TTFT) or CPU (if the cache is already in storage). Importantly, during decoding, the model operates on decompressed KV caches; compression is used only for storage or transfer.
- **Decompression** Decompression reverses the compression steps. The most computationally intensive operation—the inverse projection using $V^\top$—can be performed layer-by-layer using submatrices of $V^\top$, allowing generation to begin early.

In the following sections we describe in detail the components of `kvtc` during calibration, compression and decompression.

### 3.1 FEATURE DECORRELATION

Unlike prior SVD-based methods that calculate a separate decomposition for each prompt (Yankun et al., 2025; Chang et al., 2025), we compute the KV cache projection matrices **once** using a calibration dataset $\mathcal{C}$, and reuse them across all requests at inference time. Preparing a single, generalizable $V$ rests on three observations. First, SVD must be computed on a large, representative sample; sampling token positions from a diverse calibration set suffices for generalization and is computationally tractable. Second, excluding keys and values corresponding to the most recent tokens and attention sinks improves the achievable compression ratio (see Tables 6 and 13 in Appendix B). Third, positional embeddings distort the apparent low-rank structure of keys and should be removed before compression (Sun et al., 2025).

During calibration, we forward all sequences from $\mathcal{C}$ through the model and collect their KV caches. For each sequence, cache entries are concatenated along the time dimension to form a global pool of positions. We then sample $n$ token positions from this pool, excluding attention sinks. For each sampled position, we take the corresponding keys (and, equivalently, values) from $l$ layers and $h$ heads, undo positional rotations, and concatenate them along the hidden dimension $d_{\text{head}}$. This yields a data matrix $C \in \mathbb{R}^{n \times p}$ with $p = l\,h\,d_{\text{head}}$, whose rows index sampled token positions. Let $\mu \in \mathbb{R}^p$ be the per-feature mean of $C$. We compute the SVD of the centered matrix,

$$C - \mu = U\,\Sigma\,V^\top, \tag{2}$$

with singular values on $\mathrm{diag}(\Sigma)$ sorted in descending order (equivalently, PCA of $C$). For any $X \in \mathbb{R}^{m \times p}$, the decorrelated representation and its inverse are

$$D = (X - \mu)\,V, \qquad X = D\,V^\top + \mu, \tag{3}$$

where the equality holds when all $p$ components are used. When $r < p$ and the basis is truncated to $V \in \mathbb{R}^{p \times r}$, then $X \approx D\,V^\top + \mu$. For scalability, we use randomized SVD (Halko et al., 2011) calculated on a GPU with target rank $r < p$, substantially reducing runtime and memory. Results for this variant are shown in Figure 6, with additional details and ablations in Appendix B.1.

## 3.2 QUANTIZATION

PCA orders principal components by explained variance. We exploit this ordering to allocate a fixed bit budget across PCA coordinates so that high-variance components receive more bits. The allocation is computed once on a calibration set and reused at inference. We quantize $D$ coordinate-wise to obtain $D^{q_1,\ldots,q_d}$, where $q_i \in \mathbb{Z}_{\geq 0}$ is the bit width assigned to the $i$-th principal component. Under a global bit budget, we minimize the Frobenius reconstruction error

$$\left\| DV^\top - D^{q_1,\ldots,q_k} V^\top \right\|_F^2 . \tag{4}$$

Because right-multiplication by an orthonormal matrix preserves the Frobenius norm, we have

$$\left\| DV^\top - D^{q_1,\ldots,q_k} V^\top \right\|_F^2 = \left\| (D - D^{q_1,\ldots,q_k}) V^\top \right\|_F^2 = \left\| D - D^{q_1,\ldots,q_k} \right\|_F^2 . \tag{5}$$

Thus, optimal bit allocation can be found directly in the decorrelated domain.

We solve the constrained allocation with a simple dynamic programming (DP) algorithm that keeps two tables: (1) the minimum reconstruction error achievable using the first $i$ principal components under a payload of $b$ bits; and (2) a backpointer storing the optimal local decision. Pseudocode and a proof sketch of optimality under these constraints are in Appendix B.17. Furthermore, inspired by the Microscaling data formats (Rouhani et al., 2023), we quantize groups of *subsequent* PCA coordinates together, each group with shared 16-bit shift and scaling factors. The DP optimizes both the per-group bit width and group size, restricted to $\{1, 16, 64, 256, 1024\}$ components per group. The total budget equals the sum of payload bits across all coordinates plus per-group shift and scaling factors.

An example allocation is shown in Figure 6. As expected, the learned bit widths decrease monotonically for subsequent principal components. Crucially, the DP assigns zero bits to a substantial number of trailing principal components. This observation motivates reducing the dimensionality early during the calculation of PCA, lowering the cost of calibration, and trimming $V$ to the dimensions with nonzero bit widths for faster compression/decompression during inference. We show the benefits of DP guided quantization over pure PCA in Appendix B.9.

## 3.3 ENTROPY CODING

Finally, the quantized values are packed into a single byte array and further compressed using the DEFLATE algorithm (Wu, 2017). Crucially, we leverage nvCOMP (NVIDIA, 2020), which enables parallel operation directly on a GPU. This step is lossless, but the added compression ratio is content-dependent. We ablate the choice of the lossless compression algorithm and the influence on final compression in Appendix B.8.

## 4 EXPERIMENTS

**Models** We use models from three families: Llama 3 (Grattafiori et al., 2024), Mistral NeMo (Mistral AI team, 2024; Sreenivas et al., 2024), and R1-distilled Qwen 2.5 (DeepSeek-AI et al., 2025). The selection includes models ranging from 1.5B to 70B parameters of base, instruct, and reasoning kind. Table 1 lists the models along with their KV cache sizes. Notably, MN-Minitron 8B has been pruned from the Mistral NeMo 12B base, retaining the original KV cache size.

**Methods** We compare our method against KIVI (Liu et al., 2024d), GEAR (Kang et al., 2024) and an FP8 quantization of KV cache to the E4M3 format (Micikevicius et al., 2022). For eviction baselines, we compare with TOVA (Oren et al., 2024) and H$_2$O (Zhang et al., 2023). For SVD baselines, we compare our method with xKV (Chang et al., 2025). Finally, we compare `kvtc` with DMS (Łańcucki et al., 2025), a trained token eviction method, on reasoning tasks.

For all methods, we follow their original intended protocols, performing prefill in the vanilla mode and compressing the KV cache only after the self-attention has been calculated. For every method except xKV, we simulate a sequence of short conversations by running compression/eviction on the cache every $c$ tokens, where $c$ depends on the method's original sliding window policy. For `kvtc` which is run with a sliding window $w = 128$, we compress/decompress every $c = 16$ tokens, leaving the window in the 112–128 token range. In the case of xKV, we compress only the prefill tokens,

Table 2: Accuracy of KV cache compression methods. Results within 1 score point of vanilla are in **bold**. See Appendix B.12 for standard error analysis. $\texttt{kvtc}_{CR\times}$ denotes $\texttt{kvtc}$ set for CR× before DEFLATE.

| | Vanilla | GEAR$_{2\text{-bit}}$ | KIVI$_{2\text{-bit}}$ | H$_2$O | TOVA | xKV | FP8 | $\texttt{kvtc}_{8\times}$ | $\texttt{kvtc}_{16\times}$ | $\texttt{kvtc}_{32\times}$ | $\texttt{kvtc}_{64\times}$ |
|---|---|---|---|---|---|---|---|---|---|---|---|
| | | | | | Llama 3.1 8B | | | | | | |
| CR | 1 | 5 | 5 | 8 | 8 | 1-5 | 2 | 9-10 | 18-22 | 34-44 | 60-88 |
| GSM8K | 56.8 | 52.8 | 52.8 | 54.3 | 54.5 | **56.6** | 55.2 | **57.0** | **56.9** | **57.8** | **57.2** |
| MMLU | 60.5 | **59.6** | **59.6** | 44.3 | 44.8 | **59.5** | **60.1** | **59.8** | **60.1** | **60.6** | **60.7** |
| QASPER | 40.4 | **40.4** | 39.1 | 34.3 | 38.6 | 35.6 | **40.8** | **40.1** | **40.7** | 39.4 | 37.8 |
| LITM | **99.4** | 96.9 | 88.8 | 20.2 | 1.2 | **99.9** | **99.4** | **99.3** | **99.3** | **99.1** | 90.2 |
| RULER-VT | **99.8** | **99.8** | 98.9 | 50.4 | **99.7** | **99.8** | **99.9** | **99.1** | **99.1** | 98.9 | 95.9 |
| | | | | | MN-Minitron 8B | | | | | | |
| CR | 1 | 5 | 5 | 8 | 8 | 1-5 | 2 | 10-11 | 17-21 | 32-46 | 53-95 |
| GSM8K | 59.1 | 57.9 | 58.0 | 55.3 | **59.2** | **59.3** | **60.1** | **60.6** | **60.3** | **59.1** | 57.8 |
| MMLU | **64.3** | **63.6** | 63.2 | 43.5 | 48.1 | 63.1 | **64.3** | **64.2** | **64.1** | 63.7 | 62.1 |
| QASPER | 38.2 | **38.2** | **38.2** | 30.0 | 33.9 | 34.5 | **38.3** | 39.1 | 38.6 | 37.7 | **38.1** |
| LITM | **99.8** | 96.0 | 86.3 | 16.6 | 0.3 | **99.6** | **99.8** | **99.4** | **99.3** | 86.9 | 59.5 |
| RULER-VT | **99.4** | 98.3 | 96.8 | 39.2 | **99.3** | **99.1** | **99.2** | 98.8 | 98.8 | 96.0 | 93.4 |
| | | | | | Mistral NeMo 12B | | | | | | |
| CR | 1 | 5 | 5 | 8 | 8 | 1-5 | 2 | 10-11 | 17-20 | 31-43 | 51-87 |
| GSM8K | 61.9 | 59.8 | 59.7 | 57.0 | 60.3 | **61.9** | **61.7** | **62.5** | **62.0** | **62.2** | **61.9** |
| MMLU | 64.5 | **64.0** | **64.3** | 45.4 | 49.0 | **63.9** | **64.5** | **64.6** | **64.4** | **64.4** | 61.4 |
| QASPER | 38.4 | **38.6** | **38.2** | 29.5 | 36.0 | 33.5 | **37.9** | 37.6 | 37.6 | 37.5 | **38.0** |
| LITM | **99.5** | 96.9 | 91.9 | 16.2 | 8.7 | 97.9 | **99.0** | **99.9** | **99.8** | **99.6** | 95.3 |
| RULER-VT | **99.8** | **99.4** | 98.3 | 35.2 | **99.6** | **99.4** | **99.8** | **99.5** | **99.5** | 98.9 | 98.0 |

providing it with an advantage, since xKV is specifically designed for prefill optimization, and re-computing SVD matrices for newly decoded tokens would be prohibitively time-consuming. For a fair comparison, we only report the prefill compression ratios for non-Qwen models.

Notation-wise, $\texttt{kvtc}_{CR\times}$ denotes $\texttt{kvtc}$ in the default setting, where CR is the target compression for the DP. For all methods, we calculate CR only on the compressed tokens, not counting the sliding window tokens.

**Tasks** We evaluate compression effects on Llama 3.1 8B, MN-Minitron 8B and Mistral NeMo 12B across the following task categories, with results presented in Table 2:

- **Math & Knowledge**: 8-shot Chain of Thought (CoT) GSM8K (Cobbe et al., 2021), 4-shot CoT MMLU (Hendrycks et al., 2021a)
- **Long Context Performance**: 0-shot key-value retrieval task from (Liu et al., 2024a) (denoted LITM), 1-shot RULER (Hsieh et al., 2024) Variable Tracking (denoted RULER-VT), and 2-shot Qasper (Shaham et al., 2022). In Appendix B.13 we provide an extended long context evaluation using 2WikiMultiHopQA (Ho et al., 2020), MultiFieldQA (Bai et al., 2024), MuSiQue (Trivedi et al., 2022), QMSum (Zhong et al., 2021), SAMSum (Gliwa et al., 2019) from LongBench (Bai et al., 2024) and Common/Frequent Words Extraction (CWE/FWE), Needle in a Haystack (NIAH) (Kamradt, 2023), HotPotQA (Yang et al., 2018), SQuAD (Rajpurkar et al., 2016; 2018) from RULER (Hsieh et al., 2024), showing that $\texttt{kvtc}$ can still maintain performance comparable to vanilla with approximately $20\times$ compression ratio.

We evaluate R1-distilled models on challenging mathematical competitions AIME 2024-2025 (Art of Problem Solving, 2025) and coding tasks from LiveCodeBench (Jain et al., 2025), with results presented in Table 3. We additionally evaluate $\texttt{kvtc}$ with Llama 3.3 70B Instruct on MATH-500 (Hendrycks et al., 2021b; Lightman et al., 2023), the key-value retrieval task from (Liu et al., 2024a) and Needle in a Haystack (NIAH) (Kamradt, 2023; Hsieh et al., 2024), with results presented in Table 4. Detailed evaluation protocols can be found in Appendix A, and ablations and details about parameter choices for $\texttt{kvtc}$ in Appendix B.

**Calibration Data** We sample a 1:1 mixture of short and long documents, with lengths in the 1–8K and 8–32K ranges, respectively. Rotary positional embeddings are removed for calibration; further details are provided in Appendix B.1. Ablations showing $\texttt{kvtc}$ stability and generalization across domains of the calibration data are provided in Appendices B.5 and B.6.

## 4.1 RESULTS

In all experiments, `kvtc` applies the same compression ratio to both the key and value caches. An ablation of their individual compressibility is presented in Table 7 (Appendix B), suggesting that further adjustments could yield additional gains.

**General-Purpose Base Models** We evaluate `kvtc` on general-purpose models at the 8–12B scale, featuring three GQA-enabled models (Table 2). The compression ratio of `kvtc` varies due to the data-dependent nature of the DEFLATE algorithm, which, on average, achieves a compression ratio of approximately $1.23\times$ **on top** of quantization. Crucially, `kvtc` maintains high accuracy across tested tasks, even at substantial compression ratios of $32\times$ and $64\times$. Conversely, quantization methods—GEAR and KIVI—exhibit signs of performance degradation on GSM8K and Lost in the Middle tasks at $5\times$ CR; cache eviction methods such as $H_2O$ and TOVA perform poorly as generic KV cache compressors. We also note that xKV performs well across most tasks, except for Qasper. Interestingly, in certain cases, `kvtc` at very high compression ratios even surpasses the performance of the vanilla models. A similar observation was made in (Łańcucki et al., 2025) for a token eviction method; we provide additional insights in Appendix B.6, Table 11. Crucially, `kvtc` at $16\times$ compression (approximately $20\times$ after DEFLATE) consistently maintains results within $< 1$ score point (accuracy or F1, depending on the task) of the vanilla models. The standard errors for these results are reported in Table 19 (Appendix B.12).

**Reasoning Models** In order to test `kvtc` under more challenging conditions where context plays a critical role, we use complex math and coding tasks (Table 3). Due to high variability, AIME results are averaged over eight independent runs with results reported as $\mathrm{score}_{\pm\mathrm{std}}$. On coding tasks, $\mathrm{kvtc}_{8\times}$ shows minor accuracy drops of 0.3pp for the 1.5B model and 0.2pp for the 7B model. Notably, the KV cache size of the 1.5B model is already small at $29\,\mathrm{KiB/token}$, compared to $131\,\mathrm{KiB/token}$ for Llama 3.1 8B, and a $9\times$ compression shrinks it to only $3.2\,\mathrm{KiB/token}$. We also compare our method against DMS, a state-of-the-art autoregressive KV cache token eviction method. DMS achieves competitive results, and since it employs token eviction, it could potentially be combined with `kvtc` for even lower KV cache footprint.

Table 3: Reasoning quality (sampling temp 0.6, top-p 95%) of DeepSeek-R1-distilled Qwen2.5. DMS results as reported by Łańcucki et al. (2025).

| Method | CR | AIME24 | AIME25 | LCB |
| --- | --- | --- | --- | --- |
| | | Competition Math | | Coding |
| Qwen 2.5 R1 1.5B | | | | |
| Vanilla | 1 | $26.2_{\pm 4.8}$ | $21.7_{\pm 2.9}$ | 16.4 |
| $\mathrm{kvtc}_{8\times}$ | 9 | $25.4_{\pm 5.7}$ | $24.2_{\pm 4.0}$ | 16.1 |
| $\mathrm{kvtc}_{16\times}$ | 18 | $27.9_{\pm 6.7}$ | $22.5_{\pm 5.2}$ | 13.3 |
| $\mathrm{DMS}_{8\times}$ | - | 23.3 | N/A | 16.1 |
| Qwen 2.5 R1 7B | | | | |
| Vanilla | 1 | $50.9_{\pm 4.9}$ | $40.8_{\pm 4.3}$ | 36.7 |
| $\mathrm{kvtc}_{8\times}$ | 9-11 | $52.5_{\pm 3.6}$ | $40.8_{\pm 5.2}$ | 36.5 |
| $\mathrm{kvtc}_{16\times}$ | 18-21 | $50.9_{\pm 6.8}$ | $38.3_{\pm 5.5}$ | 31.6 |
| $\mathrm{DMS}_{8\times}$ | - | 50.0 | N/A | 33.4 |

**Multi-GPU Inference** To investigate attainable compression ratios for models distributed across multiple GPUs, we evaluate `kvtc` using Llama 3.3 70B (Table 4). The model runs in a pipeline-parallel setting (Hu et al., 2021) on four GPUs, each handling 20 layers. We maintain a local KV cache on each GPU, applying `kvtc` separately. Accuracy drops on the MATH-500 task are within $1.5 \times \mathrm{stderr}$, with accuracy decreasing by 1.2pp at $10\times$ compression and 3.0pp at $20\times$.

**Latency** We calculate the latency of elements of the compression pipeline and provide the results in Table 5. In contrast to full re-computation of KV cache for 8K context length, $\mathrm{kvtc}_{16\times}$ can reduce time-to-first-token (TTFT) up to $8\times$.

## 5 LIMITATIONS AND FUTURE WORK

**Online Compression and Composability with Other Methods** `kvtc` was designed for efficient storage and reducing time-to-first-token. However, the advantage of having a single, generalizable PCA matrix for initial compression of cache renders it suitable for further exploration of inference directly in the principal component space. We leave this as future work. Notably, `kvtc` does not alter the structure of KV cache and does not change how the attention is calculated. Consequently, it is directly compatible with token eviction methods, including but not limited to these used in the

Table 4: Compression applied to a model which is split across four GPUs (pipeline parallel), with KV cache chunks being compressed separately with `kvtc`. In certain scenarios, like offloading to CPU RAM, these chunks could be compressed jointly for higher accuracy.

| Method | MATH-500 | NIAH | LITM |
|---|---|---|---|
| Llama 3.3 70B Instruct | | | |
| Vanilla | $75.6_{1.92}$ | 100.0 | 100.0 |
| $\text{kvtc}_{8\times}$ | $73.2_{1.98}$ | 100.0 | 100.0 |
| $\text{kvtc}_{10\times}$ | $74.4_{1.95}$ | 100.0 | 100.0 |
| $\text{kvtc}_{16\times}$ | $73.2_{1.98}$ | 100.0 | 100.0 |
| $\text{kvtc}_{20\times}$ | $72.6_{1.99}$ | 100.0 | 100.0 |

Table 5: Latency of a simple implementation based on the Transformers library (Hugging Face, 2025b), measured on an NVIDIA H100 GPU using Mistral NeMo 12B in bfloat16. In addition, we compare TTFT during recomputation of KV caches with decompression of compressed caches. BS denotes batch size, CTX context length.

| Module | BS=8 CTX=8K | | BS=2 CTX=16K | |
|---|---|---|---|---|
| | Comp | Decomp | Comp | Decomp |
| Project | 153 ms | 156 ms | 78 ms | 75 ms |
| Quantize | 67 ms | 37 ms | 39 ms | 27 ms |
| Deflate | 137 ms | 64 ms | 66 ms | 36 ms |
| Total | **379 ms** | **267 ms** | **194 ms** | **143 ms** |
| Vanilla recompute TTFT | **3098 ms** | | **1780 ms** | |
| `kvtc` decomp TTFT | **380 ms** | | **208 ms** | |

experimental section, such as TOVA. Finally, `kvtc` could be used to compress the latent state in Multi-head Latent Attention (DeepSeek-AI et al., 2024).

**Scalability and Generalization Limits** We approximate deployment by evaluating `kvtc` on benchmark tasks in simulated multi-turn settings, which may not fully reflect real content distributions or interaction patterns. Our experiments cover dense decoder-only models from 1.5B to 70B parameters; evaluating larger models under conditions that more accurately mirror production is left for future work. For calibration, we process approximately 200K tokens on a single NVIDIA H100 SXM 80GB GPU. Under these settings, computing the PCA basis with the randomized algorithm of Halko et al. (2011) completes within minutes. As shown in Figures 6, 10 and 11, increasing the calibration set size consistently reduces the Frobenius-norm reconstruction error. Scaling `kvtc` beyond 200K tokens, primarily a matter of scaling the computation of PCA, is left for future work. Finally, we report Frobenius-norm reconstruction error as a proxy for downstream task accuracy. While convenient, this metric does not guarantee task-level gains and its predictive power may be task-dependent. We provide an initial correlation analysis in Appendix B.5 and defer a systematic study of alternative proxies to future work. Finally, the compression and decompression times of `kvtc` can be substantially reduced via kernel fusion and hierarchical PCA: first at the level of individual layers, then across groups of layers. Nevertheless, even a simple implementation yields substantial benefits and, in many cases, incurs only marginal overhead.

# 6 RELATED WORK

**Tuning-Free Quantization** Quantization-based methods that avoid model fine-tuning offer a straightforward path to KV cache compression (Zhao et al., 2024; Sheng et al., 2023). Works such as KIVI (Liu et al., 2024d) and KVQuant (Hooper et al., 2024) have advanced this direction by developing separate quantization strategies for key and value embeddings. These methods leverage the observation that keys benefit from per-channel quantization, while values are better suited to per-token quantization. Our approach diverges from these methods by first projecting concatenated embeddings from attention layers using SVD matrices derived from a calibration set. Quantization is then applied in this transformed space, with the precision dynamically optimized via dynamic programming bit allocation. While we adopt KIVI's uniform quantization scheme, our application occurs in the SVD-transformed domain. Similar to KVQuant, we apply compression before RoPE (Su et al., 2024) to preserve model quality.

**Tuning-Based Quantization** A complementary approach to post-training quantization involves fine-tuning models to adapt to quantized activations. LLM-QAT (Liu et al., 2024c) leverages generations from the pre-quantized model for fine-tuning, while BitDistiller (Du et al., 2024) merges Quantization Aware Training (Jacob et al., 2018) with Knowledge Distillation (Hinton et al., 2015) In contrast, our method eliminates the need for parameter modifications.

**Singular Value Decomposition Approaches** SVD has emerged as a straightforward method for removing the redundancy in KV caches and exploiting its low-rank structure. GEAR (Kang et al., 2024) improves quantization through low-rank correction mechanisms, whereas LoRC (Zhang et al., 2024) minimizes computational overhead by directly reducing the rank of key and value matrices. Eigen Attention (Saxena et al., 2024) restructures attention computation by projecting into a truncated subspace defined by SVD, enabling efficient operations. A similar mechanism could be devised for value vectors in `kvtc`, and key vectors for layers that do not employ positional embeddings. GEAR (Kang et al., 2024) improves KIVI quantization through low-rank correction mechanisms. Building on ShadowKV (Sun et al., 2025), SVDq (Yankun et al., 2025) integrates SVD with quantization, leveraging singular value magnitudes for a simple precision allocation; xKV (Chang et al., 2025) aggregates KV caches across multiple layers before decomposition. This work differs in three respects: (i) it models rotational relationships between non-adjacent layers to enable cross-layer concatenation before decomposition; (ii) it selects ranks and bitwidths via a dynamic program under a compression budget; and (iii) it applies entropy coding to the quantized factors. Empirical comparisons and ablations (e.g., treatment of early sink tokens) are reported in Appendix B.

**Sparse Attention Strategies** Sparse attention mechanisms provide a complementary paradigm for managing sequence length dimensions by selectively discarding non-essential keys/values during inference. Techniques such as $H_2O$ (Zhang et al., 2023) and TOVA (Oren et al., 2024) employ prioritization strategies to dynamically prune less informative elements from the KV cache. In contrast, chunk-based approaches like Quest (Tang et al., 2024), Landmark Attention (Mohtashami & Jaggi, 2023), and Native Sparse Attention (Yuan et al., 2025) construct compressed representations of the KV cache by partitioning sequences into chunks. These methods retrieve only the most critical chunks during attention computation, significantly reducing the number of memory transfers. Concurrently, dynamic compression techniques such as Dynamic Memory Compression (Nawrot et al., 2024) and Dynamic Memory Sparsification (Łańcucki et al., 2025) optimize KV cache memory usage through pooling/eviction of keys/values. These strategies could be integrated with quantization and SVD methods to achieve further gains (Yankun et al., 2025).

**Transform Coding** Transform coding underpins media codecs because it decorrelates local structure and compacts energy into a few coefficients, enabling aggressive quantization and entropy coding of the resulting sparse residuals (Ahmed et al., 1974; Wallace, 1992; ISO & IEC, 1998; JVT, 2003; Sze et al., 2014). Similar low-frequency structure might appear in neural models, motivating transform coding for activations and compression-aware training (Baskin et al., 2021; Young et al., 2021), or repurposing hardware H.264/H.265 encoders as codecs for LLM weights and KV caches (Xu et al., 2025). Our approach similarly builds on the transform coding paradigm. However, rather than relying on existing algorithms, we design a novel combination of decorrelation, quantization and entropy coding, which exploits cross-layer dependencies to achieve higher compression ratios.

**Cache Management Systems** Finally, cache management systems address the operational challenges of KV cache handling in production environments. Paged Attention (Kwon et al., 2023) mitigates memory overhead by introducing chunked memory allocation for KV caches. Continuous batching techniques, as implemented in systems like vLLM (Kwon et al., 2023) and FasterTransformer (NVIDIA, 2021), optimize device utilization by enabling parallel processing of multiple sequences. CacheGen (Liu et al., 2024b) advanced the field with a distributed framework for long-term KV cache management, incorporating compression, streaming, and cross-node coordination. Our approach extends these systems by integrating fine-grained compression capabilities.

## 7 CONCLUSION

We introduce `kvtc`, a method for compressing KV cache up to $20\times$ with negligible quality degradation, and higher compression ratios of $40\times$ or more available for specific use cases. We empirically show that key and value caches exhibit substantial redundancy, which `kvtc` exploits through a simple, transform coding pipeline. It is built around linear dimensionality reduction and a dynamic programming algorithm, which assigns variable numbers of bits to principal components. We demonstrate the effectiveness of `kvtc` across both regular and thinking model families, evaluating models from 1.5B to 70B. We believe that `kvtc` paves the way towards more efficient LLM deployments, lowering the cost of LLM-assisted iterative workflows.

## REPRODUCIBILITY

To foster reproducibility of our results, we provide extensive details about the calibration of PCA matrices in Appendix B.1 along with ablations regarding `kvtc` parameters in Appendices B.3 (sink tokens), B.4 (key vs value compressibility), B.5 (amount of calibration data), B.6 (effect of calibration data domain) and B.7 (sliding window). In Appendix A, we present the details about the evaluation setup: tasks, used prompts, and baseline configuration. We note that we utilize LM Evaluation Harness (Gao et al., 2024) and RULER (Hsieh et al., 2024) for evaluation, which are publicly available. In Appendix B.17 we provide the pseudocode for the dynamic programming precision assignment algorithm along with the sketch of the optimality proof and complexity analysis.

## ETHICAL STATEMENT

As a method that aims to improve aspects regarding LLM usage, `kvtc` does not introduce new risks. However, we note that it can amplify existing ones. Therefore, we refer to the existing body of knowledge on the ethical risks of LLM development, such as Ethics Threats in LLM-Based Agents (Gan et al., 2024), potential reversal of safety alignment (Xu et al., 2024), and more general risks regarding LLMs (Li & Fung, 2025).

## ACKNOWLEDGMENTS

The authors thank Mikołaj Błaż, Przemysław Podczasi, Piotr Tarasiewicz, and Przemysław Strzelczyk for many helpful discussions; Kevin Shih and Dima Zhylko for valuable comments on earlier versions of the manuscript; Szymon Migacz for assistance with computing infrastructure; and Alex Fit-Florea and Michael Lightstone for their support for the publication of this work.

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

APPENDIX

# A    EVALUATION DETAILS

## A.1    TASKS

For evaluation, we utilize Language Model Evaluation Harness (Gao et al., 2024) and RULER (Hsieh et al., 2024) with the Transformers library (Hugging Face, 2025b) serving as an inference backend.

### A.1.1    GSM8K

GSM8K (Cobbe et al., 2021) is an established task for the evaluation of the reasoning of non-reasoning models (models without thinking phase (`<think>`...`</think>`) before answer) (Yang et al., 2025). We evaluate it in an 8-shot CoT setting with few-shot examples from (Wei et al., 2022). Following (Meta, 2024) we allow for the generation of up to 1024 tokens. Task name in LM Evaluation Harness is `gsm8k_cot`.

---

**GSM8K 8-shot prompt example**

```
Q: There are 15 trees in the grove. Grove workers will plant trees in the grove today. After they are
↪  done, there will be 21 trees. How many trees did the grove workers plant today?
A: There are 15 trees originally. Then there were 21 trees after some more were planted. So there
↪  must have been 21 - 15 = 6. The answer is 6.

Q: If there are 3 cars in the parking lot and 2 more cars arrive, how many cars are in the parking
↪  lot?
A: There are originally 3 cars. 2 more cars arrive. 3 + 2 = 5. The answer is 5.

Q: Leah had 32 chocolates and her sister had 42. If they ate 35, how many pieces do they have left in
↪  total?
A: Originally, Leah had 32 chocolates. Her sister had 42. So in total they had 32 + 42 = 74. After
↪  eating 35, they had 74 - 35 = 39. The answer is 39.

Q: Jason had 20 lollipops. He gave Denny some lollipops. Now Jason has 12 lollipops. How many
↪  lollipops did Jason give to Denny?
A: Jason started with 20 lollipops. Then he had 12 after giving some to Denny. So he gave Denny 20 -
↪  12 = 8. The answer is 8.

Q: Shawn has five toys. For Christmas, he got two toys each from his mom and dad. How many toys does
↪  he have now?
A: Shawn started with 5 toys. If he got 2 toys each from his mom and dad, then that is 4 more toys. 5
↪  + 4 = 9. The answer is 9.

Q: There were nine computers in the server room. Five more computers were installed each day, from
↪  monday to thursday. How many computers are now in the server room?
A: There were originally 9 computers. For each of 4 days, 5 more computers were added. So 5 * 4 = 20
↪  computers were added. 9 + 20 is 29. The answer is 29.

Q: Michael had 58 golf balls. On tuesday, he lost 23 golf balls. On wednesday, he lost 2 more. How
↪  many golf balls did he have at the end of wednesday?
A: Michael started with 58 golf balls. After losing 23 on tuesday, he had 58 - 23 = 35. After losing
↪  2 more, he had 35 - 2 = 33 golf balls. The answer is 33.

Q: Olivia has $23. She bought five bagels for $3 each. How much money does she have left?
A: Olivia had 23 dollars. 5 bagels for 3 dollars each will be 5 x 3 = 15 dollars. So she has 23 - 15
↪  dollars left. 23 - 15 is 8. The answer is 8.

Q: {QUESTION}
A:
```

---

### A.1.2 MMLU

MMLU (Hendrycks et al., 2021a) is a collection of multiple-choice questions spanning 57 subjects. For MMLU evaluation, we use 4-shot `mmlu_flan_cot_fewshot` from LM Evaluation Harness, allowing for generation of up to 256 tokens. We pick this setup instead of the perplexity-based forward-pass evaluation of MMLU, because our baselines perform a prefill using full precision KV cache. Therefore, for a fair evaluation, we need to make the model generate a non-zero number of tokens before providing the answer, as otherwise the performance would be identical to vanilla.

---

**MMLU 4-shot prompt example**

```
The following are multiple choice questions (with answers) about abstract algebra.Q: Statement 1 |
↪  Every element of a group generates a cyclic subgroup of the group. Statement 2 | The symmetric
↪  group S_10 has 10 elements.
 (A) True, True (B) False, False (C) True, False (D) False, True
A: Let's think step by step. A cyclic group is a group that is generated by a single element. Hence a
↪  subgroup generated by a single element of a group is cyclic and Statement 1 is True. The answer
↪  is (C).

Q: The symmetric group $S_n$ has $
actorial{n}$ elements, hence it is not true that $S_{10}$ has 10 elements.
Find the characteristic of the ring 2Z.
 (A) 0 (B) 3 (C) 12 (D) 30
A: Let's think step by step. A characteristic of a ring is R is $n$ if the statement $ka = 0$ for all
↪  $a\in 2Z$ implies that $k$ is a multiple of $n$. Assume that $ka = 0$ for all $a\in 2Z$ for some
↪  $k$. In particular $2k = 0$. Hence $k=0$ and $n=0$. The answer is (A).

Q: Statement 1| Every function from a finite set onto itself must be one to one. Statement 2 | Every
↪  subgroup of an abelian group is abelian.
 (A) True, True (B) False, False (C) True, False (D) False, True
A: Let's think step by step. Statement 1 is true. Let $S$ be a finite set. If $f:S
ightarrow S$ is a onto function, then $|S| = |f(S)|$. If $f$ was not one to one, then for finite
↪  domain $S$ the image would have less than $S$ elements, a contradiction.
Statement 2 is true. Let $G$ be an abelian group and $H$ be a subgroup of $G$. We need to show that
↪  $H$ is abelian. Let $a,b \in H$. Then $a,b \in G$ and $ab=ba$. Since $G$ is abelian, $ab=ba$.
↪  Since $H$ is a subgroup of $G$, $ab \in H$. Therefore, $ab=ba$ and $H$ is abelian. The answer is
↪   (A).

Q: Statement 1 | If aH is an element of a factor group, then |aH| divides |a|. Statement 2 | If H and
↪  K are subgroups of G then HK is a subgroup of G.
 (A) True, True (B) False, False (C) True, False (D) False, True
A: Let's think step by step. Statement 2 is false. Let $H$ be a subgroup of $S_3$ generated by the
↪  cycle $(1,2)$ and $K$ be a subgroup of $S_3$ generated by the cycle $(1,3)$. Both $H$ and $K$
↪  have two elements, the generators and the identity. However $HK$ contains cycles (1,2), (1,3) and
↪   (2,3,1), but the inverse of (2,3,1) is (2,1,3) and it does not belong to HK, hence HK is not a
↪  subgroup. The answer is (B).

Q: {QUESTION}
A: Let's think step by step.
```

---

### A.1.3 LOST IN THE MIDDLE

Lost in the Middle (Liu et al., 2024a) is evaluated in a 0-shot 100-keys (300 keys for Llama 3.3 70B) setup, allowing generation of 64 tokens using the prompt presented below. We implement the evaluation using the LM Evaluation Harness framework and utilize the UUID strings and code from (Liu et al., 2024a). This benchmark allows for methodical testing of the model's ability to access the input context. As an evaluation metric, we utilize the exact match between the UUID returned by the model and the gold answer.

---

**Lost in the Middle question example (base models)**

```
Extract the value corresponding to the specified key in the JSON object below.

JSON data:
{"1afcec1f-1acd-42e3-b833-e7882d5daada": "25f1a78d-a2f6-4c7d-8bd6-51226b263cbe",
 "94071d67-86df-455c-8ee9-691e492ff740": "0d7ba717-e034-410e-88ab-c13d37cc6499",
 "88b322bb-571c-4e55-9934-aa8df11b3349": "c54095cf-9931-460b-8a6b-e1f09afb2f72",
 ...
 "5a729e1f-6956-4c1d-b024-10b317ed5657": "cea37ae5-84e1-4deb-b4c6-19d04134d664",
 "aaed65fc-f80c-4090-a0a9-90592140b9de": "ffc2e314-2d0f-4b20-be32-916ba96d1ea9",
 "90b7fe08-8708-451a-badf-34cabe7930a4": "7bebff53-05c7-4ca3-9314-bca68bd65c04"}
 "1afcec1f-1acd-42e3-b833-e7882d5daada":
```

---

**Lost in the Middle question example (Llama 3.3 70B instruct)**

```
<|begin_of_text|><|start_header_id|>system<|end_header_id|>

Cutting Knowledge Date: December 2023
Today Date: 26 Jul 2024

<|eot_id|><|start_header_id|>user<|end_header_id|>

Extract the value corresponding to the specified key in the JSON object below.

JSON data:
{"2a85047d-fe61-4c53-8844-1d85668d6a7d": "a4852ee7-d94f-40ce-8c2f-f14e95377e79",
 "1698320e-2ba6-499f-b119-b8ffd74d53db": "e212083b-22f5-4d27-87d3-5c3cfcf9542f",
 ...
 "90ef90b0-7972-46d0-9a73-4b07be2f5aae": "ebb84b27-9156-4d23-8a7d-a34aef606f28",
 "c1736979-584d-4b93-8e25-a206770fcdae": "dcb5aace-7fb2-4288-bfc2-c5faacf89469"}
 What is the value associated with the key "2a85047d-fe61-4c53-8844-1d85668d6a7d"? Answer using the
↪   following format:

`The value associated with the key "2a85047d-fe61-4c53-8844-1d85668d6a7d" is ANSWER_HERE.`

Where ANSWER_HERE is the value associated with the key
↪   "2a85047d-fe61-4c53-8844-1d85668d6a7d".<|eot_id|><|start_header_id|>assistant<|end_header_id|>
```

### A.1.4 VARIABLE TRACKING

Variable Tracking is evaluated in 1-shot (with a relatively short example shown below) using RULER (Hsieh et al., 2024) with context length 8K, limiting the generation to 128 tokens. The benchmark tests the model's ability to track variable assignments across unrelated contexts.

> **Variable Tracking prompt and question example**
>
> ```
> Memorize and track the chain(s) of variable assignment hidden in the following text.
>
> The grass is green. The sky is blue. The sun is yellow. Here we go. There and back again.
>  The grass is green. The sky is blue. The sun is yellow. Here we go. There and back again.
>  The grass is green. The sky is blue. The sun is yellow. Here we go. There and back again.
>  The grass is green. The sky is blue. The sun is yellow. Here we go. There and back again.
>  The grass is green. The sky is blue. The sun is yellow. Here we go. There and back again.
>  The grass is green. The sky is blue. The sun is yellow. Here we go. There and back again.
>  VAR JUP = 97498 The grass is green. The sky is blue. The sun is yellow. Here we go. There and back
>  ↪   again.
>  The grass is green. The sky is blue. The sun is yellow. Here we go. There and back again.
>  The grass is green. The sky is blue. The sun is yellow. Here we go. There and back again.
>  The grass is green. The sky is blue. The sun is yellow. Here we go. There and back again.
>  The grass is green. The sky is blue. The sun is yellow. Here we go. There and back again.
>  The grass is green. The sky is blue. The sun is yellow. Here we go. There and back again.
>  The grass is green. The sky is blue. The sun is yellow. Here we go. There and back again.
>  VAR AGD = VAR JUP  The grass is green. The sky is blue. The sun is yellow. Here we go. There and
>  ↪   back again.
>  The grass is green. The sky is blue. The sun is yellow. Here we go. There and back again.
>  The grass is green. The sky is blue. The sun is yellow. Here we go. There and back again.
>  The grass is green. The sky is blue. The sun is yellow. Here we go. There and back again.
>  The grass is green. The sky is blue. The sun is yellow. Here we go. There and back again.
>  The grass is green. The sky is blue. The sun is yellow. Here we go. There and back again.
>  VAR KCB = VAR AGD  The grass is green. The sky is blue. The sun is yellow. Here we go. There and
>  ↪   back again.
>  The grass is green. The sky is blue. The sun is yellow. Here we go. There and back again.
>  The grass is green. The sky is blue. The sun is yellow. Here we go. There and back again.
>  The grass is green. The sky is blue. The sun is yellow. Here we go. There and back again.
>  The grass is green. The sky is blue. The sun is yellow. Here we go. There and back again.
>  The grass is green. The sky is blue. The sun is yellow. Here we go. There and back again.
>  The grass is green. The sky is blue. The sun is yellow. Here we go. There and back again.
>  VAR LJP = VAR KCB  The grass is green. The sky is blue. The sun is yellow. Here we go. There and
>  ↪   back again.
>  The grass is green. The sky is blue. The sun is yellow. Here we go. There and back again.
>  VAR LFP = VAR LJP  The grass is green. The sky is blue. The sun is yellow. Here we go. There and
>  ↪   back again.
>
> Question: Find all variables that are assigned the value 97498 in the text above. Answer: According
>  ↪   to the chain(s) of variable assignment in the text above, 5 variables are assgined the value
>  ↪   97498, they are:  JUP AGD KCB LJP LFP
>
>
> Memorize and track the chain(s) of variable assignment hidden in the following text.
>
> ...
> Question: Find all variables that are assigned the value 79092 in the text above. Answer: According
>  ↪   to the chain(s) of variable assignment in the text above, 5 variables are assgined the value
>  ↪   79092, they are:
> ```

### A.1.5 NEEDLE IN A HAYSTACK

Needle in a Haystack (NIAH) (Kamradt, 2023) is evaluated 0-shot using RULER (Hsieh et al., 2024) with context length 100K for Llama 3.3 70B and 8K for other models, limiting the generation to 128 tokens. The benchmark tests the model's ability to retrieve information hidden in long text. The name of the task in the RULER is `niah_single_2`.

```
NIAH prompt example

<|begin_of_text|><|start_header_id|>system<|end_header_id|>

Cutting Knowledge Date: December 2023\nToday Date: 26 Jul 2024

<|eot_id|><|start_header_id|>user<|end_header_id|>

A special magic number is hidden within the following text. Make sure to memorize it. I will quiz
↪  you about the number afterwards.

<essay text prefix>
<needle>
<essay text suffix>
What is the special magic number for abrasive-pathology mentioned in th
e provided text? The special magic number for abrasive-pathology mentioned in the provided text
↪  is<|eot_id|><|start_header_id|>assistant<|end_header_id|>
```

### A.1.6 QASPER

Qasper (Dasigi et al., 2021; Shaham et al., 2022) is evaluated in 2-shot using LM Evaluation Harness task `scrolls_qasper` with full autoregressive generation and F1 score (same reasons as with MMLU evaluation). We limit the generation to 128 tokens. This benchmark evaluates the model's ability to answer questions about a paper presented as part of the input.

### A.1.7 ADDITIONAL TASKS FROM LONGBENCH

We use the LM Evaluation Harness' implementation of LongBench's 2WikiMultiHopQA (Ho et al., 2020), MultiFieldQA (Bai et al., 2024), MuSiQue (Trivedi et al., 2022), QMSum (Zhong et al., 2021), SAMSum (Gliwa et al., 2019). We fix the double question error using

```
Question:
```

instead of

```
Question: Question:
```

and end the generation after a new line for 2WikiMultiHopQA, MultiFieldQA, MuSiQue and SAMSum, whereas after a double new line for QMSum. Evaluation is performed in a single shot setup.

### A.1.8 ADDITIONAL TASKS FROM RULER

We use the LM Evaluation Harness' implementation of RULER's CWE/FWE, HotPotQA (Yang et al., 2018) and SQuAD (Rajpurkar et al., 2016; 2018). We evaluate the models zero-shot.

## A.1.9 AIME 2024-2025

We implement AIME evaluation using LM Evaluation Harness. Following (Łańcucki et al., 2025) we utilize prompts adopted from the Open-R1 repository (Hugging Face, 2025a) and limit the generation to 30K tokens. AIME competitions are popular for the evaluation of reasoning models such as DeepSeek R1 (DeepSeek-AI et al., 2025). To check the correctness of an answer, we utilize the following code with Math-Verify (Hugging Face, 2024):

---

**AIME 2024-2025 evaluation code**

```python
from math_verify.metric import math_metric
from math_verify.parser import LatexExtractionConfig, ExprExtractionConfig

def grade_answer(problem, model_answer):
    gold_is_latex = False
    verify_func = math_metric(
        gold_extraction_target=(
            LatexExtractionConfig() if gold_is_latex else ExprExtractionConfig(),
        ),
        pred_extraction_target=(ExprExtractionConfig(), LatexExtractionConfig()),
        aggregation_function=max,
        precision=6,
    )
    gold_answer = problem["answer"]

    try:
        with timeout(seconds=30): # custom class to throw and exception if code does not complete
        ↪ under 30 seconds
            grade, extracted_answers = verify_func([gold_answer], [model_answer])
            return grade == 1
    except:
        return False
```

---

**AIME 2024-2025 prompt**

```
<|begin_of_sentence|><|User|>Solve the following math problem efficiently and clearly:

- For simple problems (2 steps or fewer):
Provide a concise solution with minimal explanation.

- For complex problems (3 steps or more):
Use this step-by-step format:

## Step 1: [Concise description]
[Brief explanation and calculations]

## Step 2: [Concise description]
[Brief explanation and calculations]

...

Regardless of the approach, always conclude with:

Therefore, the final answer is: $\boxed{answer}$. I hope it is correct.

Where [answer] is just the final number or expression that solves the problem.

Problem: {PROBLEM}

<|Assistant|><think>
```

---

## A.1.10 LIVECODEBENCH

For LiveCodeBench (Jain et al., 2025) evaluation, we utilize the official repository and, following (Łańcucki et al., 2025), limit the generation to 16K tokens and data range from 2024-08-01 to 2025-01-31. The benchmark consists of problems from sites like leetcode.com and codeforces.com.

### A.1.11 MATH-500

For evaluation on MATH-500 (Lightman et al., 2023), a subset of MATH (Hendrycks et al., 2021b) introduced by Lightman et al. (2023), we limit the generation to 5120 tokens following (Meta, 2024). We also change the sliding window size to $w = 256$. We utilize the following prompt adopted from MATH-500 evaluation, and the following code optimized for reproduction of Llama 3.3 70B results without the LLM judge:

---

**MATH-500 eval code**

```python
from math_verify import parse, verify

def answer_normalize(answer: str) -> str:
    answer = answer.split(r"\boxed{")[-1].split("}$")[0].strip()
    answer = answer.replace(r"\left", "")
    answer = answer.replace(r"\right", "")
    answer = answer.replace(r"\begin{align}", "")
    answer = answer.replace(r"\end{align}", "")
    answer = answer.replace(r"\begin{equation}", "")
    answer = answer.replace(r"\end{equation}", "")
    answer = answer.replace(" ", "")
    answer = answer.replace(r"\$", "")
    if answer.startswith(r"\text"):
        answer = answer.replace(r"\text{", "")
        answer = answer.replace(r"}", "")

    if answer.startswith(r"x\in"):
        answer = answer.replace(r"x\in", "")

    if answer.startswith(r"y="):
        answer = answer.replace(r"y=", "")

    return answer

def compare_answers(answer: str, model_answer: str) -> bool:
    gold = parse(answer)
    model = parse(model_answer)
    res = verify(gold, model)
    if not res:
        answer = answer_normalize(answer)
        model_answer = answer_normalize(model_answer)
        print(answer, model_answer)
        gold = parse(answer)
        model = parse(model_answer)
        if not verify(gold, model): # sometimes fails for improper
        ↪   expressions
            res = verify(answer, model_answer)
            gold = answer
            model = model_answer
        if res:
            return True
        else:
            return False
    else:
        return res
```

---

---

**MATH-500 prompt**

```
<|begin_of_text|><|start_header_id|>system<|end_header_id|>

Cutting Knowledge Date: December 2023
Today Date: 26 Jul 2024

<|eot_id|><|start_header_id|>user<|end_header_id|>

Solve the following math problem efficiently and clearly:

- For simple problems (2 steps or fewer):
Provide a concise solution with minimal explanation.

- For complex problems (3 steps or more):
Use this step-by-step format:

## Step 1: [Concise description]
[Brief explanation and calculations]

## Step 2: [Concise description]
[Brief explanation and calculations]

...

Regardless of the approach, always conclude with:

Therefore, the final answer is: $\boxed{answer}$. I hope it is correct.

Where [answer] is just the final number or expression that solves the problem.

Problem: {PROBLEM}<|eot_id|><|start_header_id|>assistant<|end_header_id|>
```

---

## A.2 BASELINES CONFIGURATION

### A.2.1 KIVI

We follow (Liu et al., 2024d, Section 4.1) and use `group size` = 32 and `residual length` = 128, with 2-bits per key and 2-bits per value. We utilize the official implementation.

### A.2.2 GEAR

We follow (Kang et al., 2024) github repository and set underlying quantization to KIVI with `group size` = 64, `streaming gap` = 64, 2-bit keys and values. Additionally, we set the rank of key/value correction to 4 for prefill and 2 for generation. We utilize the official implementation.

### A.2.3 xKV

We follow (Chang et al., 2025). For Llama 3.1 8B we group layers by 4 and set the pre-RoPE key rank to 512 (compression ratio 8) and value rank to 768 (compression rank $5\frac{1}{3}$). For Mistral NeMo and MN-Minitron, we group layers by 5 (those models have 1.25 more layers than Llama 3.1 8B) and set the pre-RoPE key rank to 640 (compression ratio 8) and value rank to 960 (compression rank $5\frac{1}{3}$). We utilize the official implementation.

### A.2.4 H2O

We utilize the official implementation of (Zhang et al., 2023) and set the recent and heavy hitter fractions to $\frac{1}{16}$ of (input + max possible output size). This results in up to an 8x compression ratio. Lower compression ratios occur when the model produces shorter outputs than the maximal specified value. We note that this can give an advantage to H2O over other methods.

### A.2.5 TOVA

We utilize the official implementation of (Oren et al., 2024) and set the max cache size to $\frac{1}{8}$ of (input + possible output size). This results in up to an 8x compression ratio. Lower compression ratios occur when the model produces shorter outputs than the maximal specified value. We note that this can give an advantage to TOVA over other methods.

### A.2.6 DMS

We report the results for DMS as published in Łańcucki et al. (2025).

## A.3 HARDWARE

Experiments were run on a node with 8 × NVIDIA H100 GPUs (80 GB each). All main jobs finished within 4 h, except MMLU and Llama-3.3-70B ($\leq$ 8 h) and xKV ($\leq$ 12 h). For baselines (KIVI, GEAR, xKV, TOVA, H2O) we used the authors' public code. All runs used batch size 1 (except Qwen and vLLM evaluations) to prevent padding that could bias some of the baselines.

# B  ABLATIONS AND ADDITIONAL DETAILS

We provide additional details and ablations related to `kvtc` in the following appendices:

- **Appendix B.1**: Additional details about the hyperparameters of `kvtc`, along with justifications and references to relevant ablation studies.
- **Appendix B.2**: Properties of key and value channels.
- **Appendix B.3**: Omitting sink tokens for KV cache compression can result in significant gains at higher compression ratios.
- **Appendix B.4**: The difference in compressibility between keys and values suggests that long-context retrieval tasks may benefit from using higher precision for keys.
- **Appendix B.5**: Increasing the amount of calibration data helps preserve performance at higher compression ratios, while smaller amounts remain competitive for lower compression.
- **Appendix B.6**: Influence of the calibration data domain on downstream performance.
- **Appendix B.7**: Influence of the sliding window size on downstream performance.
- **Appendix B.8**: Ablation of lossless compression algorithms.
- **Appendix B.9**: Benefits of DP quantization over pure PCA.
- **Appendix B.10**: Benefits of cross layer PCA.
- **Appendix B.11**: Ablation per-prompt vs one-time PCA
- **Appendix B.12**: Results from Table 2, with standard errors computed by LM Evaluation Harness (Gao et al., 2024).
- **Appendix B.13**: Additional results on RULER (Hsieh et al., 2024) and LongBench (Bai et al., 2024).
- **Appendix B.14**: Sizes of PCA projection matrices (stored per model), with an emphasis on the fact that their size is only a small fraction of the model parameters.
- **Appendix B.15**: Influence of sliding window on cache size.
- **Appendix B.16**: Latency evaluation in a simplified scenario with vLLM (Kwon et al., 2023) and LMCache (Cheng et al., 2025).
- **Appendix B.17**: Pseudocode for the Dynamic Programming (DP) precision assignment algorithm, along with complexity analysis and a sketch of the optimality proof.

## B.1 PCA CALCULATION PARAMETERS

As mentioned in the main text, we utilize $8$ iterations of the randomized algorithm from (Halko et al., 2011) for PCA. We utilize 160K calibration tokens for Llama 3.1 8B, Llama 3.3 70B instruct, Mistral NeMo 12B, and MN-Minitron 8B with a dimensionality cut-off of 10K. This choice is motivated by memory efficiency and the results presented in Figures 6, 10 and 11, where the initial boost from the increase in the number of calibration tokens from 10K to 100K is relatively large compared to the boost attained when increasing from 100K to 160K. We leave the exploration of larger calibration sets for future work. In Appendix B.5 we ablate the influence of the amount of the calibration data on downstream results. In Appendix B.14 we provide the sizes of the projection matrices, noting that they are relatively small when compared to the model size (2.4% of model parameters for Llama 3.3 70B) .

For Qwen models, due to their smaller number of KV heads, we utilize 200K calibration tokens and dimensionality reduction of $8$K. For all models (unless stated otherwise), we utilize a 50/50 mixture of FineWeb (Penedo et al., 2024) and OpenR1Math traces (Open R1, 2025) due to the duality of our benchmarks (reasoning and general purpose; see Appendix B.6 for an ablation on calibration data source). We take samples from both datasets with the only filters being minimum and maximum length (1K and 32K, respectively) for both datasets and quality score ($\geq 0.95$) for FineWeb (the quality score is attached to the dataset). Additionally, we ensure that the number of tokens from documents below 8000 and above 8000 tokens is roughly the same (except in the MN-Minitron case, as this model supports up to 8K context length). Token counts and cutoffs are chosen so that a single calculation of PCA fits on a single H100 GPU with 80GB of memory and completes within 10 minutes (for details see Appendix B.5).

We emphasize that for a given model, we use the same PCA matrix for all compression ratios. The only change between compression ratios is the precision assignment, which is done automatically via a dynamic programming algorithm. Additionally, we note that the manual adjustment needed by a practitioner to adapt `kvtc` to a new model is limited to choosing the initial PCA dimensionality cutoff (if the practitioner decides to use the efficient algorithm by (Halko et al., 2011)), the number of samples that the chosen PCA implementation can handle (based on GPU/CPU memory) and the calibration sample choice if special scenarios are desired for increased compression. In this paper, we note that across a wide range of tasks, the choice of a 50/50 mixture of both short and long context FineWeb (Penedo et al., 2024) (for generic text) and OpenR1Math (Open R1, 2025) (for thinking traces) data is sufficient for Llama, Mistral, and R1-distilled Qwen models for a variety of applications. In particular, we note that our method is not harder to adjust than xKV or SVDq, due to automatic precision assignment via dynamic programming. In particular, instead of aiming for the best reconstruction error for a specific compression ratio, the user can easily alter the algorithm to aim for the highest compression ratio within a given reconstruction error constraint.

## B.2 Additional Details About KV Cache Properties

We study the relative channel activation patterns of Llama 3.1 8B, Mistral-Nemo 12B and Qwen 2.5 R1 (Figures 7 and 8). The relative activation results are computed on a 50/50 mixture of FineWeb (Penedo et al., 2024) and OpenR1Math (Open R1, 2025), with document lengths between 1K and 8K tokens. We observe that keys and values of all models show potential for dimensionality reduction and quantization (low absolute activations and variance).

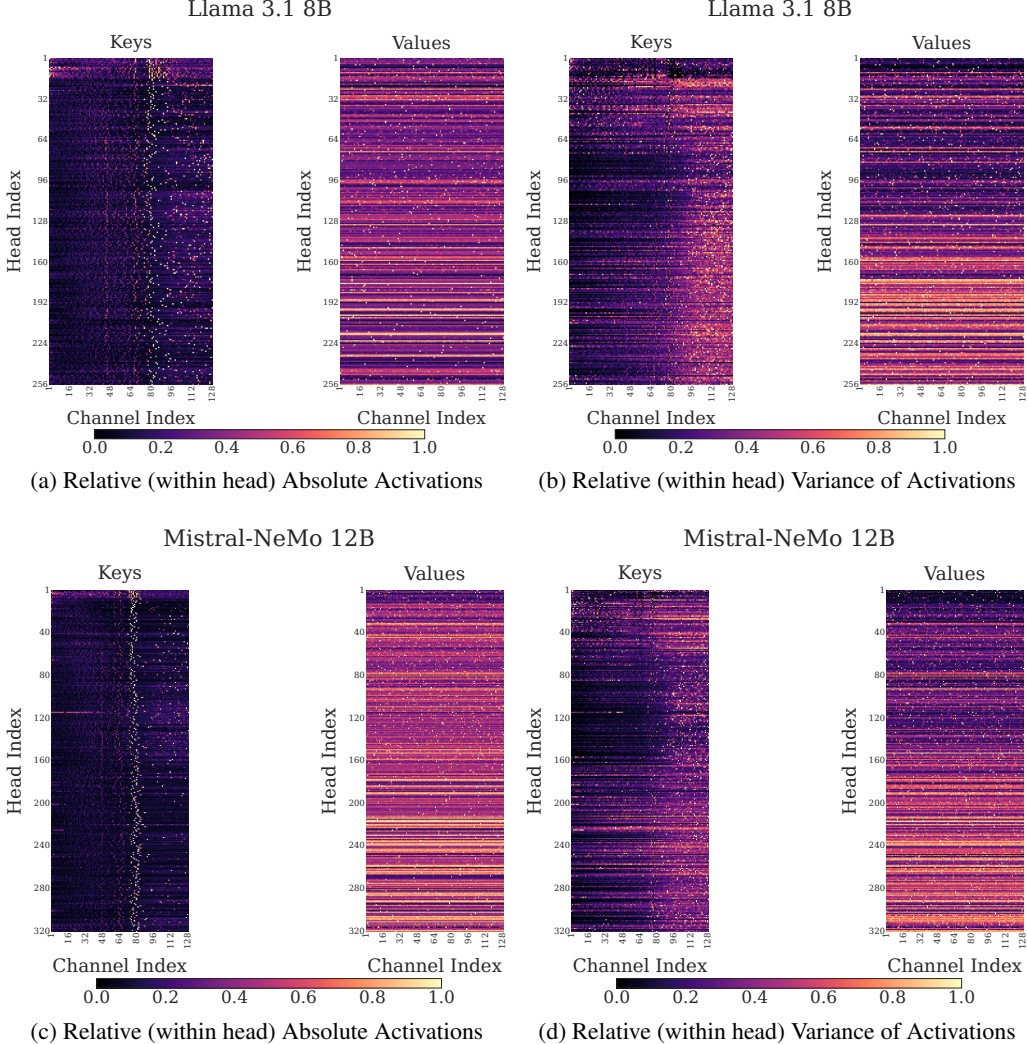

Figure 7: For key/value head $i$ and channel $j$, we define per-channel mean absolute activation $a_{i,j} = \frac{1}{n}\sum_{t=1}^{n}|\text{channel}_{t,i,j}|$ and plot the relative activation $\text{heatmap}_{i,j} = a_{i,j} / \max_b \{a_{i,b}\}$ — Panels (a),(c). Panels (b),(d) show $\frac{\text{varc}_{i,j}}{\max_c \{\text{varc}_{i,c}\}}$ where $\text{varc}_{i,j}$ is the variance of $\text{channel}_{t,i,j}$ over $t$.

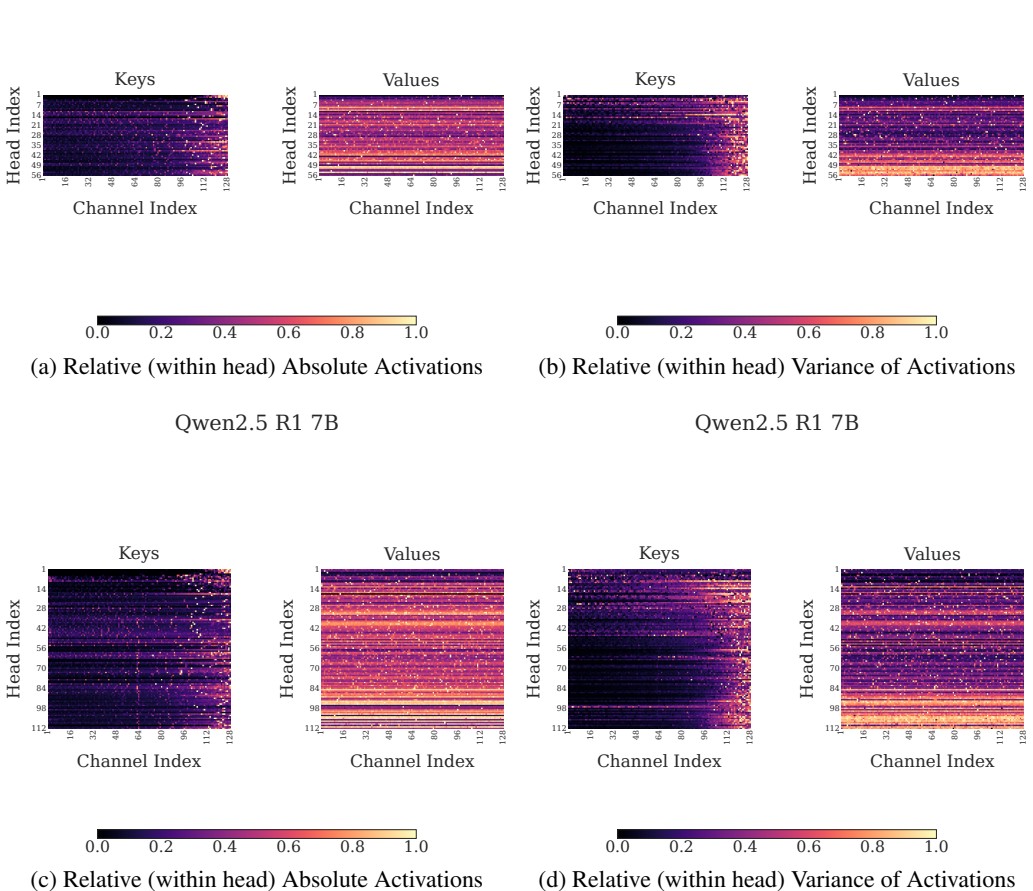

Figure 8: For key/value head $i$ and channel $j$, we define per-channel mean absolute activation $a_{i,j} = \frac{1}{n} \sum_{t=1}^{n} |\text{channel}_{t,i,j}|$ and plot the relative activation $\text{heatmap}_{i,j} = a_{i,j} / \max_b \{a_{i,b}\}$ — Panels (a),(c)(e). Panels (b),(d),(f) show $\frac{\text{varc}_{i,j}}{\max_c \{\text{varc}_{i,c}\}}$ where $\text{varc}_{i,j}$ is the variance of $\text{channel}_{t,i,j}$ over $t$.

## B.3 EXCLUDING SINK TOKENS FROM COMPRESSION

We consider the first four tokens of key and value caches (i.e., tokens at positions 0, 1, 2, and 3) to be sink tokens, motivated by the experimental results reported by (Xia et al., 2024). Reducing the dimensionality of key and value caches with PCA causes larger information loss of the initial tokens than the remaining ones, as shown in Figure 6. In order to assess the influence, we compare two setups: one that excludes first four tokens from compression (denoted $\mathtt{kvtc}^{\blacktriangleright 4}$) and one that compresses them along with other tokens (denoted $\mathtt{kvtc}^{\blacktriangleright 0}$). The results are shown in Table 6. High compression ratio of $64\times$ drastically reduces downstream task scores for the Llama 3.1 8B in the $\mathtt{kvtc}^{\blacktriangleright 0}$ case; for MN-Minitron 8B and Mistral NeMo 12B it causes regression on the long context tasks (Lost in the Middle and Variable Tracking) when compared with $\mathtt{kvtc}^{\blacktriangleright 4}$.

Table 6: Ablation on the skipping compression of the first four tokens. We note that the difference starts to be visible with larger compression ratios, and that it is in favor of skipping compression of potential attention sinks (Xiao et al., 2024). Results are presented as $\mathrm{score}_{\mathrm{stderr}}$ where $\mathrm{stderr}$ is bootstraped by LM Evaluation Harness (where available) (Gao et al., 2024). The differences in compression ratios, are due to the fact that we count the skipped tokens into compression ratio.

| Model | Method | CR | GSM8K | MMLU | QASPER | LITM 100 | RULER-VT |
|---|---|---|---|---|---|---|---|
| | | | Math & Knowledge | | | Long Context | |
| Llama 3.1 8B | $\mathtt{kvtc}^{\blacktriangleright 0}_{16\times}$ | 19-22 | $56.8_{1.4}$ | $60.3_{0.4}$ | 40.2 | $99.2_{0.1}$ | $98.2_{0.4}$ |
| | $\mathtt{kvtc}^{\blacktriangleright 4}_{16\times}$ | 18-22 | $56.9_{1.4}$ | $60.1_{0.4}$ | 40.7 | $99.3_{0.1}$ | $99.1_{0.3}$ |
| | $\mathtt{kvtc}^{\blacktriangleright 0}_{64\times}$ | 76-90 | $1.6_{0.3}$ | $9.9_{0.2}$ | 27.6 | $0.0_{0.0}$ | $61.8_{1.5}$ |
| | $\mathtt{kvtc}^{\blacktriangleright 4}_{64\times}$ | 60-88 | $57.2_{1.4}$ | $60.7_{0.4}$ | 37.8 | $90.2_{0.4}$ | $95.9_{0.6}$ |
| MN-Minitron 8B | $\mathtt{kvtc}^{\blacktriangleright 0}_{64\times}$ | 79-97 | $59.5_{1.4}$ | $61.9_{0.4}$ | 37.9 | $55.1_{0.6}$ | $91.5_{0.9}$ |
| | $\mathtt{kvtc}^{\blacktriangleright 4}_{64\times}$ | 53-95 | $57.8_{1.4}$ | $62.1_{0.4}$ | 38.1 | $59.5_{0.6}$ | $93.4_{0.8}$ |
| Mistral NeMo 12B | $\mathtt{kvtc}^{\blacktriangleright 0}_{64\times}$ | 77-89 | $61.9_{1.3}$ | $62.4_{0.4}$ | 37.5 | $84.8_{0.4}$ | $95.8_{0.6}$ |
| | $\mathtt{kvtc}^{\blacktriangleright 4}_{64\times}$ | 51-87 | $61.9_{1.3}$ | $61.4_{0.4}$ | 38.0 | $95.3_{0.3}$ | $98.0_{0.4}$ |

### B.4 SEPARATE ADJUSTMENT OF COMPRESSION RATIOS FOR KEY AND VALUE CACHES

All experimental results of kvtc (unless stated otherwise) have been obtained with 1:1 compression of keys and values. We present additional results of manually adjusting the compression ratio separately for keys and values (Table 7). The results suggest that for long-context retrieval tasks, the value cache could be compressed more than the key cache. We attribute this phenomenon to the necessity to precisely attend to selected tokens in the cache, which hinges on the high accuracy of key vectors. On the other hand, stronger compression of values shows a noticeable degradation on the GSM8K and MMLU tasks.

Table 7: Ablation of key/value compressibility with kvtc. We use the same kvtc configuration as in Table 2, but independently change key and value lossy compression rates. To denote that value compression ratio was set to 32 and key was set to 64 we use $\texttt{kvtc}^{\blacktriangleright 0}_{\frac{k:64\times}{v:32\times}}$

| Method | CR | GSM8K | MMLU | QASPER | LITM | RULER-VT |
|---|---|---|---|---|---|---|
| Llama 3.1 8B | | | | | | |
| $\texttt{kvtc}^{\blacktriangleright 4}_{\frac{k:256\times}{v:32\times}}$ | 55-80 | $57.1_{1.4}$ | 57.4 | 36.6 | $48.6_{1.6}$ | $91.9_{0.9}$ |
| $\texttt{kvtc}^{\blacktriangleright 4}_{\frac{k:32\times}{v:256\times}}$ | 55-77 | $56.8_{1.4}$ | 57.5 | 36.3 | $71.9_{1.4}$ | $95.9_{0.6}$ |
| Mistral NeMo 12B | | | | | | |
| $\texttt{kvtc}^{\blacktriangleright 0}_{\frac{k:256\times}{v:32\times}}$ | 69-79 | $62.2_{1.3}$ | 61.7 | 34.6 | $73.2_{1.4}$ | $90.4_{0.9}$ |
| $\texttt{kvtc}^{\blacktriangleright 0}_{\frac{k:32\times}{v:256\times}}$ | 69-77 | $57.1_{1.4}$ | 59.8 | 35.2 | $77.5_{1.3}$ | $89.5_{1.0}$ |

## B.5 CALIBRATION DATA TOKENS VS DOWNSTREAM PERFORMANCE

We test how the amount of calibration data affects calibration times and the downstream performance. From Figures 10 and 11 we already know that increasing the amount of the calibration data can bring down the reconstruction error. The question that remains is how such an increase correlates with the downstream performance. In Table 8 we show that an increase in calibration data clearly benefits a high compression ratio of $256\times$, with more moderate returns for smaller ($64\times$, $32\times$) compression ratios. This is a positive result, as it allows for trading calibration stage complexity for improved downstream performance, with 40K token budget bringing already competitive results for $64\times$ ratio. We additionally note that PCA calibration can be completed within 1.5 minutes for 160K tokens using (Halko et al., 2011) algorithm, and that the DP calculation time (performed once per model and compression ratio) can be finalized within 8 minutes.

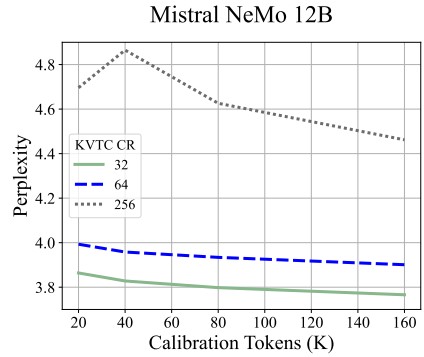

Figure 9: Model perplexity under different CRs and calibration token budgets.

Table 8: Ablation of the number of tokens used for `kvtc` calibration, along with respective PCA and DP calibration times. PCA calibration was performed using single H100 80GB GPU, calculation of DP tables (except simulation of quantization) was offloaded to the node cpu. We additionally limit DP calibration data to first 32K tokens, therefore we do not see increase in dp time after 40k calibration tokens. All calibration datapieces come from a 50/50 mixture of FineWeb (Penedo et al., 2024) and OpenMathR1 (Open R1, 2025) traces between 1K and 8K tokens. Reconstruction error and perplexity calculated on n a held-out 50/50 mixture of FineWeb and OpenR1Math data.

| Method | Data | Calibration PCA | DP | Error Key | Value | PPL | CR | GSM | MMLU | QASP | LITM | VT | AVG |
|---|---|---|---|---|---|---|---|---|---|---|---|---|---|
| | | | | | | | Mistral NeMo 12B | | | | | | |
| kvtc$_{32\times}$ | | | 6.5m | 0.184 | 0.365 | 3.864 | 31-42 | 63.0 | 63.9 | 34.8 | 97.3 | 98.9 | 71.6 |
| kvtc$_{64\times}$ | 20K | 41s | 5.4m | 0.222 | 0.440 | 3.993 | 63-87 | 63.9 | 61.7 | 31.8 | 88.8 | 96.9 | 68.6 |
| kvtc$_{256\times}$ | | | 3.9m | 0.293 | 0.565 | 4.696 | 148-340 | 59.6 | 51.5 | 23.3 | 10.4 | 66.8 | 42.3 |
| kvtc$_{32\times}$ | | | 7.2m | 0.172 | 0.341 | 3.828 | 31-43 | 64.2 | 63.0 | 35.3 | 98.7 | 99.5 | 72.1 |
| kvtc$_{64\times}$ | 40K | 48s | 6.1m | 0.212 | 0.421 | 3.958 | 63-88 | 63.7 | 61.9 | 32.4 | 95.7 | 97.8 | 70.3 |
| kvtc$_{256\times}$ | | | 4.6m | 0.285 | 0.555 | 4.865 | 148-344 | 58.3 | 50.1 | 23.7 | 9.2 | 70.6 | 42.4 |
| kvtc$_{32\times}$ | | | 7.2m | 0.163 | 0.322 | 3.798 | 31-43 | 63.6 | 63.9 | 36.0 | 98.0 | 99.3 | 72.2 |
| kvtc$_{64\times}$ | 80K | 60s | 6.1m | 0.204 | 0.405 | 3.934 | 63-88 | 63.3 | 62.0 | 32.2 | 94.5 | 97.8 | 70.0 |
| kvtc$_{256\times}$ | | | 4.6m | 0.279 | 0.543 | 4.626 | 148-339 | 60.5 | 51.7 | 23.3 | 13.6 | 83.7 | 46.6 |
| kvtc$_{32\times}$ | | | 7.2m | 0.156 | 0.308 | 3.766 | 31-43 | 63.2 | 64.3 | 36.6 | 99.5 | 99.4 | 72.6 |
| kvtc$_{64\times}$ | 160K | 86s | 6.1m | 0.198 | 0.394 | 3.901 | 63.2-87 | 64.6 | 62.2 | 32.8 | 96.9 | 97.7 | 70.8 |
| kvtc$_{256\times}$ | | | 4.6m | 0.275 | 0.535 | 4.462 | 148-341 | 61.4 | 55.5 | 24.3 | 34.2 | 79.5 | 51.0 |

| Correlation (Pearson) | PPL | AVG Score |
|---|---|---|
| Rec Error Key | 0.959 | -0.948 |
| Rec Error Value | 0.953 | -0.939 |
| PPL | 1.000 | -0.992 |
| AVG Score | -0.992 | 1.000 |

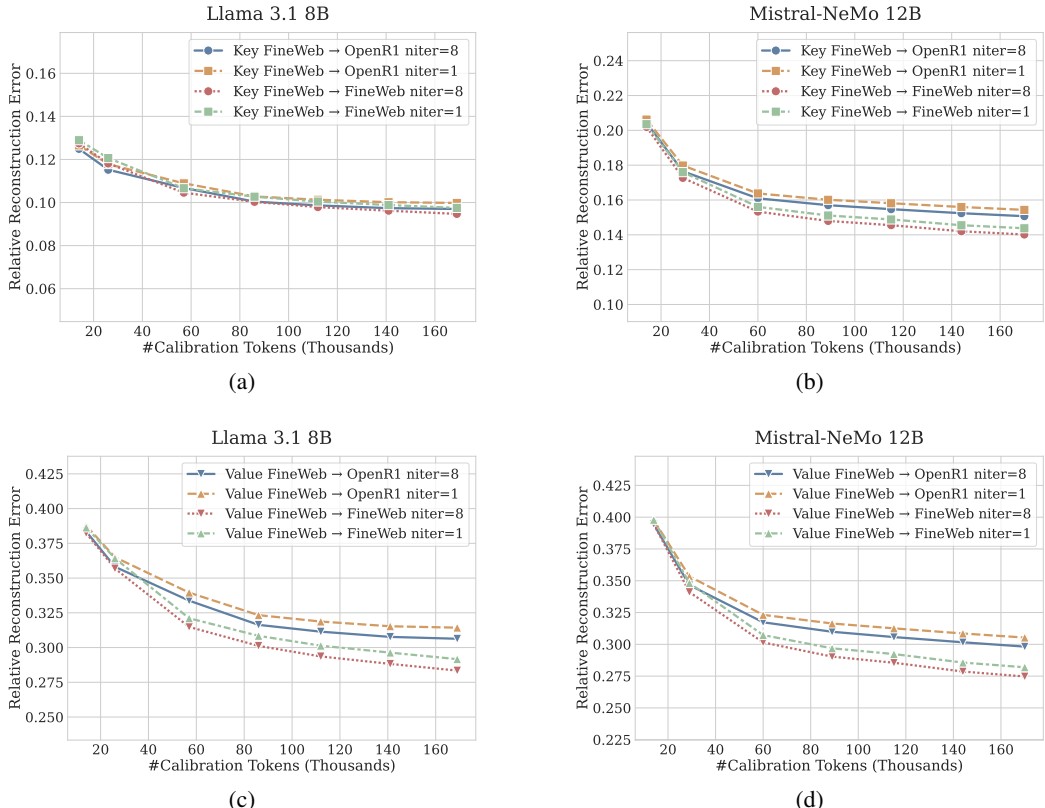

Figure 10: Relative reconstruction error when calibrating `kvtc` decorrelation step - ablation of the number of algorithm (Halko et al., 2011) iterations. Figures (a)-(b) show key reconstruction error, whereas (c)-(d) show value reconstruction error. Other parameters as in Figure 6. We note that the larger number of iterations provides slight improvements.

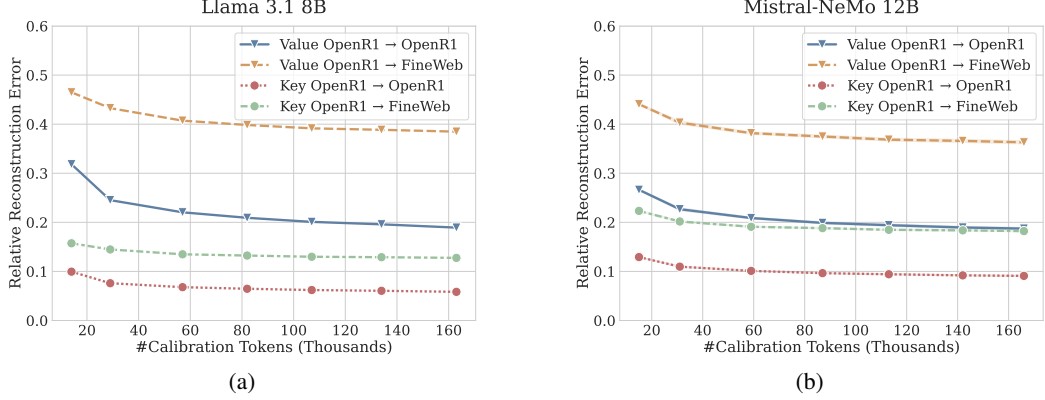

Figure 11: Relative reconstruction error when calibrating `kvtc` decorrelation step on OpenR1Math (Open R1, 2025) traces with the error calculation on FineWeb (Penedo et al., 2024). Parameters as in Figure 6. We note that OpenR1Math traces were published after the release of Llama 3.1 8B and Mistral NeMo 12B, and possibly due to their specificity result in higher generalization error.

### B.6 CALIBRATION DATA DOMAIN VS DOWNSTREAM PERFORMANCE

To examine how the calibration data domain influences downstream performance, we prepare two additional versions of `kvtc` for Mistral NeMo 12B: one using only FineWeb data and the other using only OpenR1Math data. Table 9 shows that using OpenR1Math calibration data better maintains MMLU and key-value retrieval scores than FineWeb at higher compression rates. This improvement is likely related to the question-think-answer structure of OpenR1Math, which may be more aligned with the evaluation tasks, compared to the more general web collection nature of FineWeb. However, for $32\times$ compression, both choices remain competitive. We note that for $256\times$ compression, the 50/50 mixture of FineWeb and OpenR1Math results in the best scores. For reconstruction errors, regarding calibrating on OpenR1Math/FineWeb and testing on FineWeb/OpenR1Math, see Figures 10 and 11.

We additionally check the standard deviation of task scores when sampling different calibration sets (see Table 10). We observe that `kvtc` is relatively stable when sampling 160K tokens — a 50/50 mixture of FineWeb and OpenR1Math data — as a calibration set. We also briefly check the influence of calibration data length distribution on downstream performance, showing the results in Table 11.

In Table 12, we test `kvtc` calibration under strong domain shifts. We observe that using Python, C, or Assembly from StarCoder (Li et al., 2023) instead of a 50/50 mixture of FineWeb and OpenR1Math data results in performance degradation on GSM8K. However, we note that despite a strong domain shift (code vs natural language), the model retains in-context retrieval abilities (LITM, RULER-VT) and for cr $16\times$ obtains scores that are either on par or better than KIVI/GEAR 2-bit (except the case when we calibrate on C). We note that files contained in StarCoder consist of both code and comments, which can potentially explain retention of lingual abilities.

Table 9: Ablation of the source of data. We consider `kvtc` calibrated fully on FineWeb (Penedo et al., 2024), fully on OpenMathR1 (Open R1, 2025) and a 50/50 mixture. For all cases we utilize a 50/50 mix of documents between 1K and 8K tokens along with documents between 8K and 32K tokens. We calibrate using 160K tokens.

| Method | Data | CR | GSM8K | MMLU | QASPER | LITM | RULER-VT |
|---|---|---|---|---|---|---|---|
| | Mistral NeMo 12B | | | | | | |
| $\text{kvtc}_{32\times}$ | FineWeb + OpenR1Math | 31-43 | $62.2_{1.3}$ | $63.8_{0.4}$ | 37.5 | $99.6_{0.1}$ | $98.7_{0.4}$ |
| | FineWeb | 31-43 | $63.5_{1.3}$ | $63.5_{0.4}$ | 37.5 | $98.5_{0.2}$ | $98.9_{0.3}$ |
| | OpenR1Math | 31-43 | $63.5_{1.3}$ | $64.8_{0.4}$ | 37.8 | $99.7_{0.1}$ | $99.3_{0.3}$ |
| $\text{kvtc}_{64\times}$ | FineWeb + OpenR1Math | 51-87 | $61.9_{1.3}$ | $61.4_{0.4}$ | 38.0 | $95.3_{0.3}$ | $98.0_{0.4}$ |
| | FineWeb | 63-87 | $63.2_{1.3}$ | $59.9_{0.4}$ | 36.5 | $92.0_{0.3}$ | $97.4_{0.5}$ |
| | OpenR1Math | 63-86 | $62.2_{1.3}$ | $63.7_{0.4}$ | 38.2 | $98.5_{0.2}$ | $96.5_{0.6}$ |
| $\text{kvtc}_{256\times}$ | FineWeb + OpenR1Math | 148-340 | $60.0_{1.3}$ | $52.2_{0.4}$ | 31.6 | $40.0_{0.6}$ | $84.3_{1.1}$ |
| | FineWeb | 148-343 | $57.9_{1.4}$ | $51.5_{0.4}$ | 31.0 | $14.1_{0.4}$ | $74.5_{1.4}$ |
| | OpenR1Math | 156-342 | $56.6_{1.4}$ | $54.0_{0.4}$ | 29.4 | $21.0_{0.5}$ | $81.3_{1.2}$ |

Table 10: Mean and standard deviation of downstream performance measured on 5 different calibrations sets (50/50 mixture of OpenR1Math (Open R1, 2025) and FineWeb (Penedo et al., 2024), 160K tokens, documents between 1K and 8K tokens). Results presented as $\text{mean}_{\pm\text{std}}$

| Method | CR | GSM8K | MMLU | QASPER | LITM | RULER-VT |
|---|---|---|---|---|---|---|
| | Mistral NeMo 12B | | | | | |
| Vanilla | 1 | $61.9_{1.3}$ | $64.5_{0.4}$ | 38.4 | $99.5_{0.1}$ | $99.8_{0.2}$ |
| $\text{kvtc}_{16\times}$ | 17-20 | $63.5_{\pm0.9}$ | $64.7_{\pm0.3}$ | $37.2_{\pm0.4}$ | $99.6_{\pm0.3}$ | $99.7_{\pm0.0}$ |
| $\text{kvtc}_{32\times}$ | 31-43 | $63.1_{\pm0.5}$ | $64.4_{\pm0.6}$ | $36.1_{\pm0.7}$ | $98.1_{\pm2.0}$ | $99.2_{\pm0.3}$ |
| $\text{kvtc}_{64\times}$ | 50-88 | $63.1_{\pm1.3}$ | $63.3_{\pm0.9}$ | $32.9_{\pm0.7}$ | $93.1_{\pm3.6}$ | $98.1_{\pm0.4}$ |

Table 11: Balanced 50/50 mixture of OpenR1Math (same amount of tokens from documents below 8K and above 8K) vs calibration on documents only up to $8k$ tokens. We observe that calibrating the model on shorter data can result in slightly better than vanilla performance on short context tasks.

| Method | Calibration Data Len | CR | GSM8K | MMLU | QASPER | LITM | RULER-VT |
|---|---|---|---|---|---|---|---|
| | | Mistral NeMo 12B | | | | | |
| Vanilla | - | 1 | $61.9_{1.3}$ | $64.5_{0.4}$ | 38.4 | $99.5_{0.1}$ | $99.8_{0.2}$ |
| $\texttt{kvtc}_{16\times}$ | 1K - 8K | 17-20 | $63.5_{\pm0.9}$ | $64.7_{\pm0.3}$ | $37.2_{\pm0.4}$ | $99.6_{\pm0.3}$ | $99.7_{\pm0.0}$ |
| $\texttt{kvtc}_{16\times}$ | 1K - 32K | 17-20 | $62.0_{1.3}$ | $64.4_{0.4}$ | 37.6 | $99.8_{0.0}$ | $99.5_{0.2}$ |
| $\texttt{kvtc}_{32\times}$ | 1K - 8K | 31-43 | $63.1_{\pm0.5}$ | $64.4_{\pm0.6}$ | $36.1_{\pm0.7}$ | $98.1_{\pm2.0}$ | $99.2_{\pm0.3}$ |
| $\texttt{kvtc}_{32\times}$ | 1K - 32K | 31-43 | $62.2_{1.3}$ | $63.8_{0.4}$ | 37.5 | $99.6_{0.1}$ | $98.7_{0.4}$ |
| $\texttt{kvtc}_{64\times}$ | 1K - 8K | 50-88 | $63.1_{\pm1.3}$ | $63.3_{\pm0.9}$ | $32.9_{\pm0.7}$ | $93.1_{\pm3.6}$ | $98.1_{\pm0.4}$ |
| $\texttt{kvtc}_{64\times}$ | 1K - 32K | 51-87 | $61.9_{1.3}$ | $61.4_{0.4}$ | 38.0 | $95.3_{0.3}$ | $98.0_{0.4}$ |

Table 12: Ablation of out of distribution source of data. We consider $\texttt{kvtc}$ calibrated on a 50/50 mixture of FineWeb (Penedo et al., 2024) and OpenMathR1 (Open R1, 2025) along with calibration on Python/C/Assembly from the StarCoder dataset (Li et al., 2023). In all cases we utilize files between 1K and 8K tokens and calibrate on 160K tokens. For configurations that were tested using several seeds we report $\text{score}_{\pm\text{std}}$, whereas for others we report $\text{score}_{\text{stderr}}$

| Method | Calibration Data | CR | GSM | MMLU | QASPER | LITM | RULER-VT |
|---|---|---|---|---|---|---|---|
| | | Mistral NeMo 12B | | | | | |
| Vanilla | | 1 | $61.9_{1.3}$ | $64.5_{0.4}$ | 38.4 | $99.5_{0.1}$ | $99.8_{0.2}$ |
| GEAR 2bit | - | 5 | $59.8_{1.4}$ | $64.0_{0.4}$ | 38.6 | $96.9_{0.2}$ | $99.4_{0.3}$ |
| KIVI 2bit | | 5 | $59.7_{1.4}$ | $64.3_{0.4}$ | 38.2 | $91.9_{0.3}$ | $98.3_{0.4}$ |
| $\texttt{kvtc}_{16\times}$ | FineWeb + OpenR1Math | 17-20 | $63.5_{\pm0.9}$ | $64.7_{\pm0.3}$ | $37.2_{\pm0.4}$ | $99.6_{\pm0.3}$ | $99.7_{\pm0.0}$ |
| | Python | 17-20 | $59.4_{1.4}$ | $65.2_{0.4}$ | 37.4 | $99.9_{0.0}$ | $99.7_{0.2}$ |
| | C | 17-20 | $55.0_{1.4}$ | $65.4_{0.4}$ | 38.0 | $99.2_{0.1}$ | $99.3_{0.3}$ |
| | Assembly | 17-20 | $58.9_{1.4}$ | $65.2_{0.4}$ | 37.0 | $99.9_{0.0}$ | $99.7_{0.2}$ |
| $\texttt{kvtc}_{32\times}$ | FineWeb + OpenR1Math | 31-43 | $63.1_{\pm0.5}$ | $64.4_{\pm0.6}$ | $36.1_{\pm0.7}$ | $98.1_{\pm2.0}$ | $99.2_{\pm0.3}$ |
| | Python | 35-43 | $59.0_{1.4}$ | $64.3_{0.4}$ | 36.3 | $99.7_{0.1}$ | $99.2_{0.3}$ |
| | C | 31-43 | $50.2_{1.4}$ | $64.7_{0.4}$ | 36.8 | $96.2_{0.2}$ | $99.3_{0.3}$ |
| | Assembly | 31-43 | $55.2_{1.4}$ | $64.2_{0.4}$ | 36.7 | $99.6_{0.1}$ | $99.4_{0.3}$ |
| $\texttt{kvtc}_{64\times}$ | FineWeb + OpenR1Math | 50-88 | $63.1_{\pm1.3}$ | $63.3_{\pm0.9}$ | $32.9_{\pm0.7}$ | $93.1_{\pm3.6}$ | $98.1_{\pm0.4}$ |
| | Python | 63-87 | $56.3_{1.4}$ | $62.0_{0.4}$ | 33.9 | $98.0_{0.2}$ | $97.4_{0.5}$ |
| | C | 63-87 | $54.4_{1.4}$ | $63.3_{0.4}$ | 34.5 | $95.4_{0.3}$ | $97.8_{0.5}$ |
| | Assembly | 50-86 | $60.9_{1.3}$ | $62.2_{0.4}$ | 33.0 | $99.1_{0.1}$ | $98.3_{0.4}$ |

## B.7 SLIDING WINDOW SIZE VS DOWNSTREAM PERFORMANCE

We ablate the influence of sliding window of recent not-compressed tokens on downstream performance in Table 13. We observe that increasing the length of sliding window improves downstream performance of the model, with most noticeable difference between sliding windows $\leq 16$ and sliding windows $\geq 64$.

Table 13: Ablation of the sliding window size of recent tokens that are not compressed. For a fair compression in this table we simulate a sequence of short conversations by running compressing/eviction on the cache every $s = 1$ instead every $s = 16$ tokens. Therefore, the $w = 16$ token window is left int the range 15-16 instead of 0-16.

| Method | Window Size | CR | GSM8K | MMLU | QASPER | LITM | RULER-VT |
|---|---|---|---|---|---|---|---|
| | | | Llama 3.1 8B | | | | |
| | 1 | 59-88 | $41.2_{1.4}$ | $53.8_{0.4}$ | 36.5 | $68.3_{0.6}$ | $87.7_{1.0}$ |
| $\texttt{kvtc}_{64\times}$ | 16 | 58-88 | $53.0_{1.4}$ | $56.0_{0.4}$ | 38.1 | $80.0_{0.5}$ | $88.0_{1.0}$ |
| | 64 | 57-88 | $55.8_{1.4}$ | $59.2_{0.4}$ | 37.7 | $89.8_{0.4}$ | $95.8_{0.6}$ |
| | 128 | 60-88 | $56.8_{1.4}$ | $60.5_{0.4}$ | 40.4 | $99.4_{0.1}$ | $99.8_{0.2}$ |
| | | | Mistral NeMo 12B | | | | |
| | 1 | 61-87 | $53.0_{1.4}$ | $54.9_{0.4}$ | 36.4 | $88.6_{0.4}$ | $90.7_{0.9}$ |
| $\texttt{kvtc}_{64\times}$ | 16 | 60-87 | $60.1_{1.3}$ | $56.4_{0.4}$ | 37.7 | $88.9_{0.4}$ | $95.8_{0.6}$ |
| | 64 | 59-87 | $62.2_{1.3}$ | $59.0_{0.4}$ | 37.8 | $93.0_{0.3}$ | $98.0_{0.4}$ |
| | 128 | 51-87 | $61.9_{1.3}$ | $61.4_{0.4}$ | 38.0 | $95.3_{0.3}$ | $98.0_{0.4}$ |

B.8    ABLATIONS OF LOSSLESS COMPRESSION

We ablate the differences between:

- ANS (Duda, 2014)
- Bitcomp with default nvCOMP settings (NVIDIA, 2020)
- DEFLATE (Wu, 2017) (Huffman (Huffman, 1952) + LZ77 (RASIP working group, 1997))
- GDeflate (NVIDIA, 2022)
- LZ4 (Collet, 2011)
- Snappy (Google, 2011)
- Zstandard (Facebook, 2015)
- Identity — no additional lossless compression

We present the results in Table 14. We note that DEFLATE can be easily substituted by a faster GDeflate optimized for GPUs, as the highest measured difference in compression ratio is $\leq 0.1$. We observe that the cache generated for RULER Variable Tracking is significantly more compressible than the cache generated for other tasks. We hypothesize that it may be an effect of repeated noise used as context filler by the authors (see Appendix A). We observe that ANS, DEFLATE, GDeflate, and Zstandard improve significantly over Identity in all studied cases. For a detailed study of throughput vs compression ratio of tested algorithms, we refer to (NVIDIA, 2024).

Table 14: Ablation of the lossless compression algorithms, performed using Mistral Nemo 12B with kvtc$_{32\times}$. Identity stands for no additional lossless compression (just PCA + DP Quantization; note that we count the omitted sinks into compression ratio, what is notable for tasks with shorter contexts). We mark results that have no compression ratio advantage over Identity smaller than 1 in red.

| Algorithm | Compression Ratio | | | | |
|---|---|---|---|---|---|
| | Min-Max | GSM8K | QASPER | LITM | RULER-VT |
| ANS | 32.8-37.6 | 32.8 | 37.6 | 36.8 | 37.0 |
| Bitcomp | 29.6-32.3 | 29.6 | 32.3 | 32.2 | 32.2 |
| DEFLATE | 34.7-42.9 | 34.7 | 39.5 | 39.7 | 42.9 |
| GDeflate | 34.6-42.8 | 34.6 | 39.4 | 39.6 | 42.8 |
| Identity | 29.6-32.4 | 29.6 | 32.4 | 32.2 | 32.3 |
| LZ4 | 29.5-35.5 | 29.5 | 32.4 | 32.7 | 35.5 |
| Snappy | 29.4-34.5 | 29.4 | 32.2 | 32.4 | 34.5 |
| zStandard | 34.6-46.3 | 34.6 | 39.4 | 39.9 | 46.3 |

## B.9 BENEFITS OF DP QUANTIZATION

We compare `kvtc` to a variant that does not use DP quantization, but instead removes a fraction of the least important principal components (denoted -DPQ). The results are presented in Table 15. We observe that removing PCA components, rather than applying DP quantization, leads to significant performance degradation on long context tasks. Additionally, the performance of DPQ on short context tasks deteriorates further as the length of the sliding window of uncompressed elements is reduced. We also note that DEFLATE can be much more efficient on quantized data. In general, our findings demonstrate that quantization is a crucial component of `kvtc`, and omitting it hinders scaling to larger compression ratios.

Table 15: Ablation of the importance of DP quantization over pure PCA. For alteration of sliding window size we follow the protocol from Appendix B.7.

| Window Size | Method | Modification | CR | GSM8K | MMLU | QASPER | LITM | RULER-VT |
|---|---|---|---|---|---|---|---|---|
| | | | | Llama 3.1 8B | | | | |
| 16 | $kvtc_{8\times}$ | - | 9-10 | $57.5_{1.4}$ | $59.4_{0.4}$ | 40.1 | $99.1_{0.1}$ | $98.2_{0.4}$ |
| | | -DPQ | 8-9 | $56.9_{1.4}$ | $57.5_{0.4}$ | 40.1 | $98.6_{0.1}$ | $93.0_{0.8}$ |
| | $kvtc_{16\times}$ | - | 18-22 | $57.2_{1.4}$ | $59.5_{0.4}$ | 40.6 | $99.4_{0.1}$ | $98.3_{0.4}$ |
| | | -DPQ | 16-17 | $52.0_{1.4}$ | $53.8_{0.4}$ | 36.9 | $69.2_{0.6}$ | $87.0_{1.1}$ |
| | $kvtc_{32\times}$ | - | 34-44 | $55.5_{1.4}$ | $58.2_{0.4}$ | 39.6 | $98.8_{0.1}$ | $94.8_{0.7}$ |
| | | -DPQ | 31-34 | $42.9_{1.4}$ | $36.8_{0.4}$ | 34.2 | $44.4_{0.6}$ | $83.0_{1.2}$ |
| | $kvtc_{64\times}$ | - | 58-88 | $53.0_{1.4}$ | $56.0_{0.4}$ | 38.1 | $80.0_{0.5}$ | $88.0_{1.0}$ |
| | | -DPQ | 52-68 | $21.8_{1.1}$ | $6.5_{0.2}$ | 26.7 | $5.1_{0.3}$ | $55.5_{1.6}$ |
| 128 | $kvtc_{8\times}$ | - | 9-10 | $56.7_{1.4}$ | $59.9_{0.4}$ | 40.0 | $99.3_{0.1}$ | $99.1_{0.3}$ |
| | | -DPQ | 8-9 | $57.0_{1.4}$ | $60.7_{0.4}$ | 40.2 | $99.5_{0.1}$ | $98.5_{0.4}$ |
| | $kvtc_{16\times}$ | - | 17-22 | $57.1_{1.4}$ | $60.1_{0.4}$ | 40.7 | $99.3_{0.1}$ | $99.0_{0.3}$ |
| | | -DPQ | 16-17 | $55.7_{1.4}$ | $59.7_{0.4}$ | 37.1 | $85.0_{0.4}$ | $95.5_{0.7}$ |
| | $kvtc_{32\times}$ | - | 33-44 | $58.4_{1.4}$ | $60.8_{0.4}$ | 39.3 | $99.1_{0.1}$ | $98.9_{0.3}$ |
| | | -DPQ | 31-34 | $55.1_{1.4}$ | $56.9_{0.4}$ | 35.3 | $64.5_{0.6}$ | $89.6_{1.0}$ |
| | $kvtc_{64\times}$ | - | 60-88 | $56.8_{1.4}$ | $60.5_{0.4}$ | 40.4 | $99.4_{0.1}$ | $99.8_{0.2}$ |
| | | -DPQ | 47-68 | $57.2_{1.4}$ | $49.3_{0.4}$ | 28.1 | $13.1_{0.4}$ | $58.7_{1.5}$ |

## B.10 PCA FEATURE CONCAT SIZE VS PERFORMANCE

We study how the number of layers over which we concatenate key/value heads influences the performance of `kvtc`. We present the results in Tables 16 and 17. To better isolate the influence of the number of concatenated key/value heads, we run `kvtc` without dynamic programming quantization (that is, we run `kvtc`-DPQ introduced in Appendix B.9) and with a sliding window $w = 16$. We note that results support our hypothesis about cross-layer similarity between key/value heads from Section 2, as the more layers we concatenate for calibration (PCA), the better the downstream performance.

Table 16: Ablation of the number of layers used for PCA. To better isolate the influence of the number of concatenated key/value heads, we run `kvtc` without dynamic programming quantization (that is, we run `kvtc`-DPQ introduced in Appendix B.9) and with a sliding window $w = 16$.

| Method | PCA Layers | GSM8K | MMLU | QASPER | LITM | RULER-VT |
|---|---|---|---|---|---|---|
| | | | Llama 3.1 8B | | | |
| `kvtc`$_{8\times}$-DPQ | 1 | $27.5_{1.2}$ | $24.3_{0.3}$ | 28.2 | $49.1_{0.6}$ | $70.8_{1.4}$ |
| | 2 | $44.8_{1.4}$ | $41.1_{0.4}$ | 34.2 | $77.6_{0.5}$ | $84.9_{1.1}$ |
| | 4 | $53.3_{1.4}$ | $51.9_{0.4}$ | 37.5 | $95.9_{0.2}$ | $93.0_{0.8}$ |
| | 8 | $55.9_{1.4}$ | $55.3_{0.4}$ | 39.7 | $99.2_{0.1}$ | $90.7_{0.9}$ |
| | 16 | $56.0_{1.4}$ | $57.1_{0.4}$ | 38.8 | $98.8_{0.1}$ | $92.8_{0.8}$ |
| | 32 | $56.9_{1.4}$ | $57.5_{0.4}$ | 40.1 | $98.6_{0.1}$ | $93.0_{0.8}$ |
| `kvtc`$_{16\times}$-DPQ | 1 | $2.5_{0.4}$ | $0.2_{0.0}$ | 18.4 | $0.0_{0.0}$ | $0.5_{0.2}$ |
| | 2 | $13.9_{1.0}$ | $2.3_{0.1}$ | 22.1 | $0.7_{0.1}$ | $12.9_{1.0}$ |
| | 4 | $33.3_{1.3}$ | $19.3_{0.3}$ | 29.1 | $32.0_{0.6}$ | $62.7_{1.5}$ |
| | 8 | $49.6_{1.4}$ | $43.4_{0.4}$ | 34.4 | $60.7_{0.6}$ | $85.8_{1.1}$ |
| | 16 | $51.6_{1.4}$ | $49.1_{0.4}$ | 36.0 | $72.7_{0.6}$ | $89.0_{1.0}$ |
| | 32 | $52.0_{1.4}$ | $53.8_{0.4}$ | 36.9 | $69.2_{0.6}$ | $87.0_{1.1}$ |
| `kvtc`$_{32\times}$-DPQ | 1 | $1.1_{0.3}$ | $0.1_{0.0}$ | 17.8 | $0.0_{0.0}$ | $0.1_{0.1}$ |
| | 2 | $1.5_{0.3}$ | $0.2_{0.0}$ | 18.1 | $0.0_{0.0}$ | $0.2_{0.1}$ |
| | 4 | $3.9_{0.5}$ | $0.9_{0.1}$ | 20.0 | $0.0_{0.0}$ | $1.6_{0.4}$ |
| | 8 | $28.0_{1.2}$ | $6.4_{0.2}$ | 24.7 | $2.6_{0.2}$ | $33.0_{1.5}$ |
| | 16 | $40.1_{1.4}$ | $26.0_{0.4}$ | 31.2 | $24.6_{0.5}$ | $60.4_{1.5}$ |
| | 32 | $42.9_{1.4}$ | $36.8_{0.4}$ | 34.2 | $44.4_{0.6}$ | $83.0_{1.2}$ |
| `kvtc`$_{64\times}$-DPQ | 1 | $0.9_{0.3}$ | $0.3_{0.0}$ | 18.0 | $0.0_{0.0}$ | $0.1_{0.1}$ |
| | 2 | $1.4_{0.3}$ | $0.1_{0.0}$ | 17.7 | $0.0_{0.0}$ | $0.1_{0.1}$ |
| | 4 | $1.5_{0.3}$ | $0.1_{0.0}$ | 18.1 | $0.0_{0.0}$ | $0.1_{0.1}$ |
| | 8 | $2.4_{0.4}$ | $0.6_{0.1}$ | 19.4 | $0.0_{0.0}$ | $0.2_{0.1}$ |
| | 16 | $13.9_{1.0}$ | $1.9_{0.1}$ | 22.1 | $0.0_{0.0}$ | $10.0_{0.9}$ |
| | 32 | $21.8_{1.1}$ | $6.5_{0.2}$ | 26.7 | $5.1_{0.3}$ | $55.5_{1.6}$ |

Table 17: Ablation of the number of layers used for PCA with $32\times$ dimensionality reduction applied separately to keys and values (that is, either keys or values are compressed, not both). We follow the protocol from Table 16. We observe that keys benefit more from global concatenation, whereas values show a significant boost when increasing the number of concatenated layers from 8 to 16.

| Cache | PCA Layers | GSM8K | MMLU | QASPER | LITM | RULER-VT |
|---|---|---|---|---|---|---|
| | | Llama 3.1 8B with `kvtc`$_{32\times}$-DPQ | | | | |
| Keys | 1 | $3.2_{0.5}$ | $4.4_{0.2}$ | 21.7 | $0.0_{0.0}$ | $0.1_{0.1}$ |
| | 4 | $25.9_{1.2}$ | $27.5_{0.4}$ | 25.9 | $0.2_{0.0}$ | $2.5_{0.5}$ |
| | 8 | $49.9_{1.4}$ | $43.6_{0.4}$ | 35.2 | $44.5_{0.6}$ | $74.9_{1.4}$ |
| | 16 | $52.5_{1.4}$ | $52.3_{0.4}$ | 38.6 | $86.0_{0.4}$ | $90.1_{0.9}$ |
| | 32 | $53.2_{1.4}$ | $56.3_{0.4}$ | 38.8 | $96.7_{0.2}$ | $97.4_{0.5}$ |
| Values | 1 | $2.0_{0.4}$ | $0.9_{0.1}$ | 18.0 | $0.0_{0.0}$ | $2.8_{0.5}$ |
| | 4 | $28.9_{1.2}$ | $29.0_{0.4}$ | 30.0 | $15.6_{0.5}$ | $68.2_{1.5}$ |
| | 8 | $40.9_{1.4}$ | $41.8_{0.4}$ | 34.6 | $53.1_{0.6}$ | $76.4_{1.3}$ |
| | 16 | $49.2_{1.4}$ | $45.7_{0.4}$ | 37.1 | $82.6_{0.5}$ | $88.6_{1.0}$ |
| | 32 | $48.7_{1.4}$ | $48.6_{0.4}$ | 38.7 | $81.4_{0.5}$ | $89.0_{1.0}$ |

### B.11 PCA DIFFERENT PER-PROMPT VS ONE-TIME

Table 18: Ablation of PCA Calibration. We follow the protocol described in Table 17. Conversation simulation proceeds as follows: the model maintains a window of only 16 uncompressed recent tokens and performs compression each time a new token is generated. We note that per-prompt calibration results in significantly lower compression ratios, owing to the overhead of storing the per-prompt projection matrix $V^T$. Moreover, a per-prompt $V^T$ can generalize poorly and fail to compress the continuation of the conversation effectively.

| Method | Calibration Per-Prompt | Simulate Conversation | CR | GSM8K Math | LITM 100 Long Context |
|---|---|---|---|---|---|
| | | | Llama 3.1 8B | | |
| $\texttt{kvtc}_{8\times}$-DPQ | False | False | 7.9-8.0 | $56.8_{1.4}$ | $98.8_{0.1}$ |
| | False | True | 7.9-8.0 | $56.9_{1.4}$ | $98.6_{0.1}$ |
| | True | False | 1.0-1.1 | $56.7_{1.4}$ | $99.5_{0.1}$ |
| | True | True | 1.0-1.1 | $31.4_{1.3}$ | $99.5_{0.1}$ |
| $\texttt{kvtc}_{16\times}$-DPQ | False | False | 15.4-15.9 | $54.6_{1.4}$ | $73.7_{0.5}$ |
| | False | True | 15.4-15.9 | $52.0_{1.4}$ | $69.2_{0.6}$ |
| | True | False | 1.0-2.1 | $56.7_{1.4}$ | $99.4_{0.1}$ |
| | True | True | 1.0-2.1 | $31.4_{1.3}$ | $99.5_{0.1}$ |
| $\texttt{kvtc}_{32\times}$-DPQ | False | False | 29.4-31.9 | $48.7_{1.4}$ | $49.5_{0.6}$ |
| | False | True | 29.5-31.9 | $42.9_{1.4}$ | $44.4_{0.6}$ |
| | True | False | 1.0-6.2 | $56.7_{1.4}$ | $99.4_{0.1}$ |
| | True | True | 1.0-6.2 | $31.4_{1.3}$ | $99.3_{0.1}$ |
| $\texttt{kvtc}_{64\times}$-DPQ | False | False | 54.2-63.4 | $38.6_{1.3}$ | $5.3_{0.3}$ |
| | False | True | 54.3-63.4 | $21.8_{1.1}$ | $5.1_{0.3}$ |
| | True | False | 1.3-12.4 | $58.3_{1.4}$ | $99.3_{0.1}$ |
| | True | True | 1.3-12.4 | $28.4_{1.2}$ | $98.8_{0.1}$ |

## B.12 STANDARD ERROR OF THE MAIN RESULTS

In Table 19 we attach the results from Table 2 with their standard error, as bootstrapped by LM Evaluation Harness (where available) (Gao et al., 2024). We note that downstream evaluation runs, except the Qwen models, were performed using 1 seed and greedy evaluation.

Table 19: Downstream task results, presented also in Table 2, here shown with standard error as reported by LM Evaluation Harness (where available) and CR computed both with and without sliding window (where applicable).

| Method | CR -Window | CR +Window | GSM8K | MMLU | QASPER | LITM 100 | RULER-VT |
|---|---|---|---|---|---|---|---|
| | | | Math & Knowledge | | | Long Context | |
| Llama 3.1 8B | | | | | | | |
| vanilla | 1 | 1 | $56.8_{1.4}$ | $60.5_{0.4}$ | 40.4 | $99.4_{0.1}$ | $99.8_{0.2}$ |
| GEAR 2bit | 5 | 3-5 | $52.8_{1.4}$ | $59.6_{0.4}$ | 40.4 | $96.9_{0.2}$ | $99.8_{0.2}$ |
| KIVI 2bit | 5 | 3-5 | $52.8_{1.4}$ | $59.6_{0.4}$ | 39.1 | $88.8_{0.4}$ | $98.9_{0.3}$ |
| $\text{xKV}^{2/16\ 4\text{key}}_{3/16\ 4\text{value}}$ | 1-5 | 1-5 | $56.6_{1.4}$ | $59.5_{0.4}$ | 35.6 | $99.9_{0.0}$ | $99.8_{0.2}$ |
| FP8 | 2 | 2 | $55.2_{1.4}$ | $60.1_{0.4}$ | 40.8 | $99.4_{0.1}$ | $99.9_{0.1}$ |
| $\text{kvtc}_{8\times}$ | 9-10 | 3-9 | $57.0_{1.4}$ | $59.8_{0.4}$ | 40.1 | $99.3_{0.1}$ | $99.1_{0.3}$ |
| $\text{kvtc}_{16\times}$ | 18-22 | 4-17 | $56.9_{1.4}$ | $60.1_{0.4}$ | 40.7 | $99.3_{0.1}$ | $99.1_{0.3}$ |
| $\text{kvtc}_{32\times}$ | 34-44 | 5-29 | $57.8_{1.4}$ | $60.6_{0.4}$ | 39.4 | $99.1_{0.1}$ | $98.9_{0.3}$ |
| $\text{kvtc}_{64\times}$ | 60-88 | 5-45 | $57.2_{1.4}$ | $60.7_{0.4}$ | 37.8 | $90.2_{0.4}$ | $95.9_{0.6}$ |
| $\text{H}_2\text{O}^{1/16\ \text{recent}}_{1/16\ \text{past}}$ | 8 | 8 | $54.3_{1.4}$ | $44.3_{0.4}$ | 34.3 | $20.2_{0.5}$ | $50.4_{1.6}$ |
| $\text{TOVA}\frac{1}{8}$ | 8 | 8 | $54.5_{1.4}$ | $44.8_{0.4}$ | 38.6 | $1.2_{0.1}$ | $99.7_{0.2}$ |
| MN-Minitron 8B | | | | | | | |
| Vanilla | 1 | 1 | $59.1_{1.4}$ | $64.3_{0.4}$ | 38.2 | $99.8_{0.0}$ | $99.4_{0.3}$ |
| GEAR 2bit | 5 | 3-5 | $57.9_{1.4}$ | $63.6_{0.4}$ | 38.2 | $96.0_{0.2}$ | $98.3_{0.4}$ |
| KIVI 2bit | 5 | 3-5 | $58.0_{1.4}$ | $63.2_{0.4}$ | 38.2 | $86.3_{0.4}$ | $96.8_{0.6}$ |
| $\text{xKV}^{2/16\ 5\text{key}}_{3/16\ 5\text{value}}$ | 1-5 | 1-5 | $59.3_{1.4}$ | $63.1_{0.4}$ | 34.5 | $99.6_{0.1}$ | $99.1_{0.3}$ |
| FP8 | 2 | 2 | $60.1_{1.3}$ | $64.3_{0.4}$ | 38.3 | $99.8_{0.1}$ | $99.2_{0.3}$ |
| $\text{kvtc}_{8\times}$ | 10-11 | 3-9 | $60.6_{1.3}$ | $64.2_{0.4}$ | 39.1 | $99.4_{0.1}$ | $98.8_{0.3}$ |
| $\text{kvtc}_{16\times}$ | 17-21 | 3-16 | $60.3_{1.3}$ | $64.1_{0.4}$ | 38.6 | $99.3_{0.1}$ | $98.8_{0.3}$ |
| $\text{kvtc}_{32\times}$ | 32-46 | 3-27 | $59.1_{1.4}$ | $63.7_{0.4}$ | 37.7 | $86.9_{0.4}$ | $96.0_{0.6}$ |
| $\text{kvtc}_{64\times}$ | 53-95 | 3-38 | $57.8_{1.4}$ | $62.1_{0.4}$ | 38.1 | $59.5_{0.6}$ | $93.4_{0.8}$ |
| $\text{H}_2\text{O}^{1/16\ \text{recent}}_{1/16\ \text{past}}$ | 8 | 8 | $55.3_{1.4}$ | $43.5_{0.4}$ | 30.0 | $16.6_{0.5}$ | $39.2_{1.5}$ |
| $\text{TOVA}\frac{1}{8}$ | 8 | 8 | $59.2_{1.4}$ | $48.1_{0.4}$ | 33.9 | $0.3_{0.1}$ | $99.3_{0.3}$ |
| Mistral NeMo 12B | | | | | | | |
| Vanilla | 1 | 1 | $61.9_{1.3}$ | $64.5_{0.4}$ | 38.4 | $99.5_{0.1}$ | $99.8_{0.2}$ |
| GEAR 2bit | 5 | 3-5 | $59.8_{1.4}$ | $64.0_{0.4}$ | 38.6 | $96.9_{0.2}$ | $99.4_{0.3}$ |
| KIVI 2bit | 5 | 3-5 | $59.7_{1.4}$ | $64.3_{0.4}$ | 38.2 | $91.9_{0.3}$ | $98.3_{0.4}$ |
| $\text{xKV}^{2/16\ 5\text{key}}_{3/16\ 5\text{value}}$ | 1-5 | 1-5 | $61.9_{1.3}$ | $63.9_{0.4}$ | 33.5 | $97.9_{0.2}$ | $99.4_{0.3}$ |
| FP8 | 2 | 2 | $61.7_{1.3}$ | $64.5_{0.4}$ | 37.9 | $99.0_{0.1}$ | $99.8_{0.2}$ |
| $\text{kvtc}_{8\times}$ | 10-11 | 3-10 | $62.5_{1.3}$ | $64.6_{0.4}$ | 37.6 | $99.9_{0.0}$ | $99.5_{0.2}$ |
| $\text{kvtc}_{16\times}$ | 17-20 | 3-16 | $62.0_{1.3}$ | $64.4_{0.4}$ | 37.6 | $99.8_{0.0}$ | $99.5_{0.2}$ |
| $\text{kvtc}_{32\times}$ | 31-43 | 3-29 | $62.2_{1.3}$ | $63.8_{0.4}$ | 37.5 | $99.6_{0.1}$ | $98.7_{0.4}$ |
| $\text{kvtc}_{64\times}$ | 51-87 | 3-47 | $61.9_{1.3}$ | $61.4_{0.4}$ | 38.0 | $95.3_{0.3}$ | $98.0_{0.4}$ |
| $\text{H}_2\text{O}^{1/16\ \text{recent}}_{1/16\ \text{past}}$ | 8 | 8 | $57.0_{1.4}$ | $45.4_{0.4}$ | 29.5 | $16.2_{0.5}$ | $35.2_{1.5}$ |
| $\text{TOVA}\frac{1}{8}$ | 8 | 8 | $60.3_{1.3}$ | $49.0_{0.4}$ | 36.0 | $8.7_{0.4}$ | $99.6_{0.2}$ |

### B.13 ADDITIONAL LONGBENCH AND RULER RESULTS

We additionally evaluate `kvtc` on 2WikiMultiHopQA (2WQA) (Ho et al., 2020), MultiFieldQA (MFQA) (Bai et al., 2024), MuSiQue (MQUE) (Trivedi et al., 2022), QMSum (QMS) (Zhong et al., 2021), SAMSum (SAMS) (Gliwa et al., 2019) from LongBench (Bai et al., 2024). We also evaluate on Common/Frequent Words Extraction (CWE/FWE), Needle in a Haystack (NIAH) (Kamradt, 2023), HotPotQA (HPQA) (Yang et al., 2018), SQuAD (SQA) (Rajpurkar et al., 2016; 2018) from RULER (Hsieh et al., 2024). We present the results in Table 20, showing that `kvtc` can still maintain comparable performance to vanilla with $\approx 20\times$ compression ratio.

Table 20: Additional results (with stderr) on LongBench (1 host) and RULER (0 shot) tasks, methods configured as in Table 2. Given the RULER 0-shot QA results we hypothesize that both Llama 3.1 8B and MN-Minitron 8B base models were exposed to some amount of question answering data with format similar to the mentioned tasks, whereas it might have not been the case for Mistral NeMo 12B.

| Task | Vanilla | GEAR | KIVI | H$_2$O | TOVA | xKV | FP8 | kvtc$_{16\times}$ | kvtc$_{32\times}$ | kvtc$_{64\times}$ |
|---|---|---|---|---|---|---|---|---|---|---|
| | | | | | Llama 3.1 8B | | | | | |
| CR | 1 | 5 | 5 | 8 | 8 | 4-6 | 2 | 18-20 | 35-39 | 62-78 |
| 2WQA | $40.8_{3.3}$ | $40.7_{3.3}$ | $42.1_{3.3}$ | $38.5_{3.2}$ | $41.3_{3.3}$ | $39.5_{3.2}$ | $40.6_{3.3}$ | $40.3_{3.3}$ | $40.0_{3.2}$ | $40.6_{3.3}$ |
| MFQA | $50.3_{2.6}$ | $49.6_{2.6}$ | $50.1_{2.7}$ | $40.8_{2.4}$ | $50.4_{2.5}$ | $48.7_{2.6}$ | $50.9_{2.6}$ | $51.1_{2.6}$ | $49.5_{2.6}$ | $50.2_{2.5}$ |
| MQUE | $33.8_{3.1}$ | $33.8_{3.1}$ | $31.7_{3.0}$ | $32.4_{3.0}$ | $32.6_{3.1}$ | $31.5_{3.0}$ | $33.8_{3.1}$ | $33.7_{3.1}$ | $33.9_{3.1}$ | $34.1_{3.0}$ |
| QMS | $26.9_{0.7}$ | $27.2_{0.7}$ | $25.7_{0.6}$ | $25.4_{0.6}$ | $25.4_{0.6}$ | $24.4_{0.7}$ | $26.1_{0.7}$ | $26.4_{0.7}$ | $25.9_{0.6}$ | $25.2_{0.6}$ |
| SAMS | $47.3_{1.3}$ | $47.2_{1.3}$ | $45.8_{1.2}$ | $46.8_{1.3}$ | $46.1_{1.3}$ | $45.7_{1.3}$ | $47.1_{1.3}$ | $47.0_{1.3}$ | $46.6_{1.3}$ | $45.6_{1.3}$ |
| CWE | $94.7_{1.0}$ | $94.0_{1.1}$ | $91.0_{1.3}$ | $64.9_{2.1}$ | $76.5_{1.9}$ | $68.9_{2.1}$ | $94.5_{1.0}$ | $92.4_{1.2}$ | $90.7_{1.3}$ | $88.0_{1.5}$ |
| FWE | $92.1_{1.2}$ | $91.9_{1.2}$ | $89.9_{1.4}$ | $75.7_{1.9}$ | $69.5_{2.0}$ | $88.6_{1.4}$ | $92.3_{1.2}$ | $89.0_{1.4}$ | $88.1_{1.5}$ | $83.3_{1.7}$ |
| NIAH | $100_{0.0}$ | $100_{0.0}$ | $100_{0.0}$ | $6.2_{1.1}$ | $99.6_{0.3}$ | $99.8_{0.2}$ | $100_{0.0}$ | $100_{0.0}$ | $99.8_{0.2}$ | $99.6_{0.3}$ |
| HPQA | $57.2_{2.2}$ | $56.8_{2.2}$ | $57.2_{2.2}$ | $48.8_{2.2}$ | $54.8_{2.2}$ | $56.2_{2.2}$ | $57.6_{2.2}$ | $57.2_{2.2}$ | $57.2_{2.2}$ | $55.8_{2.2}$ |
| SQA | $55.7_{2.2}$ | $55.7_{2.2}$ | $54.0_{2.2}$ | $40.2_{2.2}$ | $51.3_{2.2}$ | $53.5_{2.2}$ | $55.6_{2.2}$ | $55.2_{2.2}$ | $53.1_{2.2}$ | $53.8_{2.2}$ |
| AVG | **59.9** | **59.7** | 58.8 | 42.0 | 54.8 | 55.7 | **59.9** | **59.2** | 58.5 | 57.6 |
| | | | | | MN-Minitron 8B | | | | | |
| CR | 1 | 5 | 5 | 8 | 8 | 4-6 | 2 | 19 | 39-40 | 78-80 |
| 2WQA | $45.8_{3.3}$ | $45.3_{3.3}$ | $45.4_{3.3}$ | $44.6_{3.3}$ | $45.5_{3.3}$ | $47.1_{3.3}$ | $45.8_{3.3}$ | $46.2_{3.3}$ | $46.4_{3.3}$ | $45.1_{3.3}$ |
| MFQA | $42.6_{2.9}$ | $42.2_{2.8}$ | $43.0_{2.9}$ | $34.8_{2.5}$ | $40.3_{2.7}$ | $41.6_{2.9}$ | $43.2_{2.9}$ | $42.6_{2.9}$ | $43.4_{2.9}$ | $43.2_{2.9}$ |
| MQUE | $27.4_{2.8}$ | $26.8_{2.8}$ | $26.2_{2.8}$ | $25.6_{2.7}$ | $26.0_{2.8}$ | $26.8_{2.8}$ | $27.3_{2.8}$ | $27.1_{2.8}$ | $26.6_{2.8}$ | $26.3_{2.8}$ |
| QMS | $23.4_{0.6}$ | $23.0_{0.6}$ | $23.2_{0.6}$ | $21.9_{0.5}$ | $22.5_{0.5}$ | $21.8_{0.6}$ | $23.5_{0.6}$ | $23.1_{0.6}$ | $22.7_{0.6}$ | $22.6_{0.6}$ |
| SAMS | $36.1_{1.6}$ | $36.5_{1.6}$ | $36.8_{1.6}$ | $36.7_{1.6}$ | $36.1_{1.6}$ | $34.7_{1.6}$ | $36.2_{1.6}$ | $35.9_{1.6}$ | $35.9_{1.6}$ | $35.8_{1.6}$ |
| CWE | $92.4_{1.2}$ | $90.5_{1.3}$ | $87.1_{1.5}$ | $64.7_{2.1}$ | $66.4_{2.1}$ | $69.7_{2.1}$ | $92.5_{1.2}$ | $85.3_{1.6}$ | $79.5_{1.8}$ | $75.2_{1.9}$ |
| FWE | $86.2_{1.5}$ | $86.5_{1.5}$ | $85.8_{1.6}$ | $70.3_{2.0}$ | $73.7_{2.0}$ | $85.4_{1.6}$ | $85.9_{1.6}$ | $86.3_{1.5}$ | $83.1_{1.7}$ | $81.0_{1.8}$ |
| NIAH | $100_{0.0}$ | $100_{0.0}$ | $99.8_{0.2}$ | $6.0_{1.1}$ | $99.8_{0.2}$ | $97.6_{0.7}$ | $100_{0.0}$ | $100_{0.0}$ | $100_{0.0}$ | $100_{0.0}$ |
| HPQA | $62.0_{2.2}$ | $63.2_{2.2}$ | $58.4_{2.2}$ | $49.0_{2.2}$ | $57.8_{2.2}$ | $56.6_{2.2}$ | $62.8_{2.2}$ | $62.2_{2.2}$ | $55.8_{2.2}$ | $54.6_{2.2}$ |
| SQA | $64.9_{2.1}$ | $65.5_{2.1}$ | $62.4_{2.2}$ | $48.0_{2.2}$ | $62.7_{2.2}$ | $63.9_{2.2}$ | $64.6_{2.1}$ | $64.7_{2.1}$ | $62.5_{2.2}$ | $62.2_{2.2}$ |
| AVG | **58.1** | **58.0** | 56.8 | 40.2 | 53.1 | 54.5 | **58.2** | **57.3** | 55.6 | 54.6 |
| | | | | | Mistral NeMo 12B | | | | | |
| CR | 1 | 5 | 5 | 8 | 8 | 4-6 | 2 | 19 | 39-40 | 78-80 |
| 2WQA | $43.3_{3.3}$ | $42.7_{3.3}$ | $42.1_{3.3}$ | $39.2_{3.3}$ | $42.7_{3.3}$ | $43.8_{3.3}$ | $43.3_{3.3}$ | $43.7_{3.3}$ | $43.8_{3.3}$ | $43.2_{3.3}$ |
| MFQA | $51.5_{2.7}$ | $51.1_{2.7}$ | $50.6_{2.7}$ | $40.8_{2.6}$ | $49.6_{2.6}$ | $50.2_{2.7}$ | $50.7_{2.7}$ | $51.0_{2.7}$ | $50.9_{2.6}$ | $51.3_{2.6}$ |
| MQUE | $27.2_{2.8}$ | $27.2_{2.8}$ | $27.2_{2.8}$ | $24.9_{2.7}$ | $27.1_{2.8}$ | $26.7_{2.8}$ | $26.9_{2.8}$ | $26.6_{2.8}$ | $26.1_{2.8}$ | $27.9_{2.8}$ |
| QMS | $25.7_{0.6}$ | $25.9_{0.6}$ | $25.7_{0.6}$ | $21.9_{0.6}$ | $24.2_{0.6}$ | $25.1_{0.6}$ | $25.7_{0.6}$ | $25.2_{0.6}$ | $24.7_{0.7}$ | $25.1_{0.7}$ |
| SAMS | $45.6_{1.4}$ | $45.2_{1.3}$ | $44.9_{1.3}$ | $42.2_{1.4}$ | $45.5_{1.4}$ | $45.3_{1.4}$ | $45.4_{1.4}$ | $45.9_{1.3}$ | $45.1_{1.4}$ | $45.6_{1.3}$ |
| CWE | $93.2_{1.1}$ | $92.6_{1.2}$ | $93.3_{1.1}$ | $65.5_{2.1}$ | $69.9_{2.1}$ | $75.2_{1.9}$ | $93.7_{1.1}$ | $90.9_{1.3}$ | $86.5_{1.5}$ | $85.9_{1.6}$ |
| FWE | $83.0_{1.7}$ | $82.6_{1.7}$ | $84.1_{1.6}$ | $77.5_{1.9}$ | $70.9_{2.0}$ | $83.4_{1.7}$ | $82.9_{1.7}$ | $82.1_{1.7}$ | $78.2_{1.8}$ | $78.9_{1.8}$ |
| NIAH | $100_{0.0}$ | $100_{0.0}$ | $99.8_{0.2}$ | $6.0_{1.1}$ | $100_{0.0}$ | $100_{0.0}$ | $100_{0.0}$ | $100_{0.0}$ | $100_{0.0}$ | $100_{0.0}$ |
| HPQA | $36.2_{2.1}$ | $35.8_{2.1}$ | $35.4_{2.1}$ | $28.8_{2.0}$ | $31.8_{2.1}$ | $35.6_{2.1}$ | $35.2_{2.1}$ | $33.8_{2.1}$ | $33.6_{2.1}$ | $33.2_{2.1}$ |
| SQA | $22.4_{1.9}$ | $25.0_{1.9}$ | $23.8_{1.9}$ | $18.3_{1.7}$ | $21.7_{1.8}$ | $21.1_{1.8}$ | $22.8_{1.9}$ | $22.9_{1.9}$ | $23.0_{1.9}$ | $21.9_{1.8}$ |
| AVG | **52.8** | **52.8** | **52.7** | 36.5 | 48.3 | 50.6 | **52.7** | **52.2** | 51.2 | 51.3 |

## B.14 PCA MATRIX SIZES

In Table 21 we present the sizes of PCA projection matrices ($V$ from $U\Sigma V^\top$) after being computed via (Halko et al., 2011) algorithm. We note that the sizes are only a relatively small fraction of the model parameters, and that they can be further reduced by the DP algorithm depending on the desired compression ratio.

Table 21: Number of parameters used by PCA matrices, before DP, for the tested models. For example, Llama 3.1 8B has 32 layers, each with 8 key/value heads, each head of size 128. Therefore, after cross-head concatenation, each key/value has $32 \times 8 \times 128 = 32768$ features. The PCA projection $V$ is cut to the first 10K principal components by (Halko et al., 2011) algorithm for efficiency, resulting in $32768 \times 10000 \simeq 328M$ parameters. Further DP bit allocation can remove additional principal directions depending on the desired compression ratio. Both models and PCA projection matrices are stored in 16bit precision.

| Model | #Params | Key/Value Features | Key/Value PCA Cap | Key/Value PCA Params | $\frac{\text{TotalPCAParam}}{\text{ModelParams}}$ |
|---|---|---|---|---|---|
| Qwen 2.5 R1 1.5B | 1.5B | $28 \times 2 \times 128 = 7168$ | 8K | 51M | 6.8% |
| Qwen 2.5 R1 7B | 7.1B | $28 \times 4 \times 128 = 14336$ | | 115M | 3.2% |
| Llama 3.1 8B | 7.5B | $32 \times 8 \times 128 = 32768$ | | 328M | 8.7% |
| Llama 3.3 70B | 69.5B | $80 \times 8 \times 128 = 81920$ | 10K | 819M | 2.4% |
| Mistral NeMo 12B | 11.6B | $40 \times 8 \times 128 = 40960$ | | 410M | 7.1% |

## B.15 COMPRESSION RATIO WITH SLIDING WINDOW

For the presented methods and a sliding window of $w$ uncompressed (high precision) keys/values corresponding to recent tokens, one can use the following formulas to calculate the compression ratio that includes the sliding window:

- $\texttt{kvtc}_{\text{cr}\times}^{\blacktriangleright s}$: $\frac{\text{ctx}}{\frac{\text{ctx}-w-s}{\text{cr}}+w+s/2}$ — we keep the first $s$ tokens in FP8
- $\texttt{kvtc}_{\text{cr}\times}$: $\frac{\text{ctx}}{\frac{\text{ctx}-w-4}{\text{cr}}+w+2}$ — we keep the first four tokens in FP8
- Other methods: $\frac{\text{ctx}}{\frac{\text{ctx}-w}{\text{cr}}+w}$

## B.16 LMCACHE + VLLM

We provide additional end-to-end measurements in a simplified, multi-user scenario. To be more precise, we have run vLLM (Kwon et al., 2023) with LMCache (Cheng et al., 2025) managing the KV cache on a workload in which, for $\approx 64K$ of initial input tokens, users ask questions that are between 16 and 100 tokens and receive 100-token answers (1 token for TTFT measure), with no delays in between conversation turns. We have used Llama 3.3 70B in FP8 precision, split over 2x H100 80GB GPUs using tensor parallelism. LMCache has been configured to use 128GiB of host (CPU DRAM) memory per GPU, which is equivalent to using 1TiB in an 8-GPU server. We present the latency results in Table 22.

We observe that with 12 or more clients, the dedicated amount of host memory becomes too little to hold the KV caches and forces recomputation, which is reflected in spiking latency. In a more realistic scenario, there would be more concurrent users taking pauses between conversation turns. We note that in this test, the KV cache is compressed with KVTC only for storage in host memory; it does not take advantage of compressing the KV cache for storage in GPU HBM, which would bring additional latency benefits, as its implementation is more involved.

Table 22: Response latency for generation of 100 tokens (left) and TTFT (right) vs. number of concurrent vLLM clients. In both cases, the initial input context is sampled to have between 62K and 66K tokens, and each user question is sampled to have between 16 and 100 tokens. We use the same GPU for prefill, generation, compression, and decompression, which can result in increased latency for `kvtc`.

| #Clients | Vanilla (s) | KVTC 16× (s) |
|---|---|---|
| 1 | 2.5 | 2.5 |
| 2 | 2.9 | 2.9 |
| 4 | 8.6 | 9.1 |
| 6 | 12.7 | 13.9 |
| 8 | 17.3 | 18.0 |
| 10 | 21.4 | 24.2 |
| 12 | 155.7 | 27.9 |
| 14 | 180.9 | 31.3 |
| 16 | 208.2 | 37.1 |

| #Clients | Vanilla (s) | KVTC 16× (s) |
|---|---|---|
| 1 | 0.2 | 0.2 |
| 2 | 0.3 | 0.3 |
| 4 | 1.2 | 1.7 |
| 6 | 2.0 | 2.7 |
| 8 | 2.8 | 3.5 |
| 10 | 3.4 | 4.5 |
| 12 | 136.6 | 5.6 |
| 14 | 159.0 | 6.4 |
| 16 | 181.6 | 7.3 |

## B.17 DYNAMIC PROGRAMMING ALGORITHM

Below, we present the pseudocode for the dynamic programming precision assignment along with a proof sketch.

---

**Dynamic Programming Precision Assignment Pseudocode**

```
D                                # calibration data matrix of shape (batch, num_features)
m = D.mean(dim=0, keepdim=True)  # of each feature across the batch dimension

U, S, V = svd(D - m)             # D - m = U @ S @ V.T

P = U@S                          # assume columns sorted by singular values
batch, num_considered_features = P.shape

# initial_reconstruction_error corresponds to quantizing everything with zero bits
initial_reconstruction_error = (P*P).sum()    # squared Frobenius norm

# set to initial_reconstruction_error as we assume that the data is initially quantized with 0 bits
# and we progressively consider non-zero quantization of more and more features
best_error = tensor(shape=(num_considered_features + 1, max_bit_budget + 1),
↪   values=initial_reconstruction_error)
best_error_type = array(shape=(num_considered_features + 1, max_bit_budget + 1), values=0)
best_error_block_size = tensor(shape=(num_considered_features + 1, max_bit_budget + 1), values=0)
best_error_bit_cost = tensor(shape=(num_considered_features + 1, max_bit_budget + 1), values=0)

# we assume that block sizes are > 0
allowed_block_sizes = [1, 16, 64, 256, 1024]

# We assume the presence of a None type that quantizes data to the array of zeros.
# We count bit usage of this type as 0,
# because it directly corresponds to the removal of principal components.
types = [None, int2, int4, fp8]

for i in range(1, num_considered_features + 1):
  for block_size in allowed_block_sizes:
    if block_size <= i:
      assert block_size > 0
      for budget in range(1, max_bit_budget + 1):
        if best_error[i, budget] > best_error[i, budget - 1]:
          best_error[i, budget] = best_error[i, budget - 1]
          best_error_type[i, budget] = best_error_type[i, budget - 1]
          best_error_block_size[i, budget] = best_error_block_size[i, budget - 1]
          best_error_bit_cost[i, budget] = best_error_bit_cost[i, budget - 1]

        for t in types:

          block_to_quantize = P[:, i - block_size:i]

          quantized_data, used_bits = simulate_quantization(block_to_quantize, t)
          if used_bits <= budget:
            zero_bit_quantize_error = (block_to_quantize * block_to_quantize).sum()
            quantization_error = block_to_quantize - quantized_data
            quantization_error = (quantization_error * quantization_error).sum()

            error_change = -zero_bit_quantize_error + quantization_error

            if best_error[i, budget] > error_change + best_error[i - block_size, budget - used_bits]:
              best_error[i, budget] = error_change + best_error[i - block_size, budget - used_bits]
              best_error_type[i, budget] = t
              best_error_block_size[i, budget] = block_size
              best_error_bit_cost[i, budget] = used_bits

# we can use best_error, best_error_type, best_error_block_size and best_error_bit_cost tables to
↪   get the quantization for a given budget
```

---

The proof of the optimality follows by simple induction on `i` and `budget`. To be more precise we want to prove that `best_error[j, q]` is the smallest reconstruction error (squared Frobenius norm) one can achieve when considering first j features of P (setting other features to 0 – 0-bit quantization) and quantization restricted to types from `types` that can only be used to quantize blocks of contiguous features of sizes in `allowed_block_sizes` sizes, while utilizing no more than `budget` bits. For simplicity we assume that the smallest reconstruction error (squared Frobenius norm) for q=0 budget cases is `initial_reconstruction_error = (P*P).sum()`. Then the proof by induction can be conducted as follows:

- For `i=0` or `budget=0` we have that if we consider quantization of the first 0 features and leave other features as zeros or budget of size 0, then the reconstruction error is indeed `(P*P).sum()`.
- Then to prove for `i>0` and `budget>0` we assume the optimality of `best_error[j, q]` for j<i, and for j=i and q<budget. Then we note that the algorithm enumerates all

possible quantization blocks that the quantization of the first `i` features can end with within the budget `budget`.

Computational complexity can be directly inferred from the pseudo-code:

$$\mathcal{O}(\text{num\_considered\_features} \times$$
$$|\text{allowed\_block\_sizes}| \times$$
$$\text{max\_bit\_budget} \times$$
$$|\text{types}| \times$$
$$q_{\text{sim}}(\max\{\text{allowed\_block\_sizes}\}, \text{batch}))$$

where

$$q_{\text{sim}}(\max\{\text{allowed\_block\_sizes}\}, \text{batch})$$

is the time taken to simulate quantization. Assuming that $|\text{allowed\_block\_sizes}|$ and $|\text{types}|$ are constant and quantization simulation can be performed in

$$\mathcal{O}(\max\{\text{allowed\_block\_sizes}\} \times \text{batch})$$

we can write the asymptotic bound on the algorithm runtime as:

$$\mathcal{O}(\text{num\_considered\_features} \times \text{max\_bit\_budget} \times \text{batch})$$

We additionally provide the runtime of the algorithm in Table 8 in Appendix B.5.

