# OpenReview forum: "KV Cache Transform Coding for Compact Storage in LLM Inference"
_ICLR.cc/2026/Conference — ICLR 2026 Poster_

### Official Review · Reviewer_jZ1g · 2025-10-30

**Soundness:** 3
**Presentation:** 3
**Contribution:** 3
**Rating:** 4
**Confidence:** 5

**Summary:**

The paper introduces KVTC, a lightweight method for compressing KV caches in LLM inference. Inspired by classical transform coding, it combines PCA-based feature decorrelation, adaptive quantization via dynamic programming, and entropy coding. KVTC achieves up to 20× compression with negligible accuracy loss and scales across models such as Llama 3.1, Mistral-NeMo, and Qwen 2.5. Experiments on multiple benchmarks show that it outperforms prior quantization and SVD-based baselines while reducing inference latency.

**Strengths:**

1. KVTC effectively reduces memory footprint by up to 20x with minimal degradation in model accuracy (<1 point), demonstrating strong practical value.

2. The combination of PCA-based transform coding, adaptive quantization via dynamic programming, and entropy coding is well-grounded in classical signal compression theory, yet thoughtfully adapted to the KV-cache structure in modern Transformers.

3. Evaluation spans multiple model families (Llama 3.1, Mistral-NeMo, Qwen 2.5) and a wide range of tasks (GSM8K, MMLU, Qasper, AIME25, MATH500, etc.), showing strong and consistent results.

**Weaknesses:**

1) The practicality of kvtc hinges critically on whether SVD components computed on calibration data can be effectively transferred to out-of-domain inference contexts. While the authors provide some analysis in Appendix C.6 on the effect of calibration data domain, I remain concerned about the potential error introduced when there is a significant distributional shift between calibration and actual input sequences. For instance, in Figure 8, the relative reconstruction error for "Value Open R1 to FineWeb" is notably higher than "Value Open R1 to Open R1".

2) Table 4 reports the TTFT comparison between recomputation and decompression, which is informative. However, it would be valuable to also understand the latency in a more typical scenario where the KV cache is already present in GPU HBM (i.e., a cache hit). In such cases, what is the TTFT overhead introduced by kvtc’s decompression pipeline?

3) It would be helpful to clarify the key technical distinctions of kvtc compared to prior SVD-based KV cache compression methods such as xKV in Introduction.

4) xKV leverages inter-layer redundancy, whereas kvtc concatenates all attention heads across layers before compression. Given that Figure 2 shows block-wise cosine similarity patterns in value states rather than uniform cross-layer similarity, the rationale for global concatenation, rather than structured, layer-aware compression, needs further justification.

5) Figure 3 appears largely illustrative and less directly tied to the core technical contributions of the paper.

6) Across Tables 2 and 3, accuracy does not consistently decrease with higher compression ratios, in some cases, KVTC even outperforms the vanilla models. This counter-intuitive result is not analyzed.

7) Baselines are compared using each method’s best reported configuration, which is reasonable to ensure strong baseline performance.However, this also makes it difficult to assess performance at equivalent compression ratios, as accuracy may not scale linearly with compression strength.

8) Figure 8 demonstrates that cross-domain evaluation leads to larger reconstruction errors, yet no mechanism for adaptive or domain-agnostic calibration is provided.

**Questions:**

see weakness

---

> ### Author Response · Authors · 2025-11-21
>
> We thank Reviewer jZ1g for taking the time to read our paper and for providing thoughtful and constructive feedback.
>
> **W1 Calibration data and generalization to other domains**
> We provide an additional evaluation of KVTC  calibrated on data from the code domain [StarCoder]which is either Python or Assembly.
>
> | Method               | Calibration Data     |    CR |  GSM8K | MMLU | QASPER | LITM | RULER-VT |
> | -------------------- | -------------------- | ----: | ---: | ---: | -----: | ---: | -------: |
> | **Mistral NeMo 12B** |                      |       |        |      |        |      |          |
> | Vanilla              | -                    |     1 |   61.9 | 64.5 |   38.4 | 99.5 |     99.8 |
> | GEAR 2bit            | -                    |     5 |   59.8 | 64.0 |   38.6 | 96.9 |     99.4 |
> | KIVI 2bit            | -                    |     5 |   59.7 | 64.3 |   38.2 | 91.9 |     98.3 |
> | KVTC 16x            | FineWeb + OpenR1Math | 17–20 |   63.5 | 64.7 |   37.2 | 99.6 |     99.7 |
> | KVTC 16x            | Python               | 17–20 |   59.4 | 65.2 |   37.4 | 99.9 |     99.7 |
> | KVTC 16x            | Assembly             | 17–20 |   58.9 | 65.2 |   37.0 | 99.9 |     99.7 |
> | KVTC 32x            | FineWeb + OpenR1Math | 31–43 |   63.1 | 64.4 |   36.1 | 98.1 |     99.2 |
> | KVTC 32x            | Python               | 35–43 |   59.0 | 64.3 |   36.3 | 99.7 |     99.2 |
> | KVTC 32x            | Assembly             | 31–43 |   55.2 | 64.2 |   36.7 | 99.6 |     99.4 |
> | KVTC 64x            | FineWeb + OpenR1Math | 50–88 |   63.1 | 63.3 |   32.9 | 93.1 |     98.1 |
> | KVTC 64x            | Python               | 63–87 |   56.3 | 62.0 |   33.9 | 98.0 |     97.4 |
> | KVTC 64x            | Assembly             | 50–86 |   60.9 | 62.2 |   33.0 | 99.1 |     98.3 |
>
> Changing the calibration dataset lowers accuracy on GSM8K, but the model retains in-context retrieval abilities (LITM, RULER-VT) and for CR $16\times$ obtains scores that are either on par or better than KIVI/GEAR 2-bit. In addition, downstream tasks like Lost in the Middle, in which a randomly generated UUID has to be retrieved from the context, is out-of-domain for FineWeb and OpenR1Math, and further demonstrates that PCA directions learned on the calibration set generalize to inputs with a different token distribution.
>
> [StarCoder] StarCoder: may the source be with you!, Raymond Li et al. 2023 Transactions on Machine Learning Research

---

> ### Author Response · Authors · 2025-11-21
>
> **W2 Latency/TTFT overhead when KV cache is read from HBM (cache hit)?**
> The three building blocks of KVTC, which are projection, quantization and entropy-coding, are memory-bound for long contexts. In the current implementation, in which these are invoked as separate kernels, the cost of decompression would be equal or less to the cost of generating three tokens at the same batch size. However, fusing three stages into a single kernel would speed up decompression roughly 3x. We would like to emphasize that the point of using KVTC to compress KV cache for its residency in HBM is to free up HBM memory for other conversation, and in the end increase the number of concurrently served users and throughput of the node. KVTC can be applied on-demand, even to old caches, so as long as there is enough free memory in HBM, the KV caches can remain uncompressed. In other words, if a KV cache in HBM needs to be decompressed with KVTC, it means that KV Cache Manager chose to compress it in the past, and the node benefited from higher throughput or the cache has been prevented from being offloaded/evicted.
>
> In addition, we provide the results of end-to-end TTFT measurements in a simplified, multi-user scenario. To this end, we have run vLLM with LMCache managing the KV cache on a workload in which, for 64K of initial input tokens, the users ask questions that are between 16 and 100 tokens and receive 100-token answers, with no delays in between conversation turns. We have used Llama 3.3 70B in FP8 precision, split over 2x H100 80GB GPUs using tensor parallelism. LMCache has been configured to use 128GiB of host (CPU DRAM) memory per GPU, which is equivalent to using 1TiB in an 8-GPU server. We adjust the load of the server by changing the number of clients that concurrently hold conversations.
>
> | vLLM num of concurrent clients | Response latency of vanilla (s) | Response latency of KVTC 16x (s) | TTFT of vanilla (s) | TTFT of  KVTC 16x (s) |
> | -----------------------------: | ------------------------------: | -------------------------------: | -------: | -----------------------: |
> |                              1 |                             2.5 |                              2.5 |      0.2 |                      0.2 |
> |                              2 |                             2.9 |                              2.9 |      0.3 |                      0.3 |
> |                              4 |                             8.6 |                              9.1 |      1.2 |                      1.7 |
> |                              6 |                            12.7 |                             13.9 |      2.0 |                      2.7 |
> |                              8 |                            17.3 |                             18.0 |      2.8 |                      3.5 |
> |                             10 |                            21.4 |                             24.2 |      3.4 |                      4.5 |
> |                             12 |                           155.7 |                             27.9 |    136.6 |                      5.6 |
> |                             14 |                           180.9 |                             31.3 |    159.0 |                      6.4 |
> |                             16 |                           208.2 |                             37.1 |    181.6 |                      7.3 |
>
> In this experiment, the cache has always been compressed, regardless of the remaining memory. As the number of concurrently served clients increases, KV caches no longer fit in the memory and need to be recalculated. For KVTC, that is not the case; the TTFT increases are noticeable, but acceptable in comparison to full response time or recalculation.
>
> [LMCache] Cheng et al., 2025, LMCache: An Efficient KV Cache Layer for Enterprise-Scale LLM Inference.
> [vLLM] Woosuk Kwon et al., 2023, Efficient Memory Management for Large Language Model Serving with PagedAttention.

---

> ### Author Response · Authors · 2025-11-21
>
> **W3 Clarify the difference between KVTC and xKV**
> The goal of both methods is different. In xKV, the KV cache is compressed in order to occupy less GPU memory during decoding. After the prefill phase, a fresh SVD is calculated on the prefilled tokens. This SVD is performed on KV caches for small groups of G consecutive layers (with k=2 and k=4 used in the experimental section). For decoding, the KV cache is decompressed, and newly generated tokens are stored uncompressed.
>
> In KVTC, we show how SVD could be calculated just once and used for all sequences. We additionally show that a dynamic programming guided quantization can be used to significantly improve the results (see Figure 5). In addition, this formulation enables grouping all layers (which would equal to k=32 in xKV for the Llama 3.1 8B model).
>
> [xKV] Chang et al., 2025, xkv: Cross-layer svd for kv-cache compression, 2025

---

> ### Author Response · Authors · 2025-11-21
>
> **W4 Global compression (all layers) vs. block-wise (like xKV). Why use global?**
> Following the reviewer's advice, we perform an ablation of the number of layers used for PCA, that is the number of consecutive layers concatenated together for PCA calculation (block-size). To better isolate the influence, we run KVTC without dynamic programming quantization (KVTC-No Quantization) and with a sliding window w=16. We clearly observe the benefits of global PCA vs local separate PCAs, and the limits of quantization-free approach.
>
> | Method       | PCA Layers | GSM8K | MMLU | QASPER | LITM | RULER-VT |
> | ------------ | ---------: | ----: | ---: | -----: | ---: | -------: |
> | **Llama 3.1 8B**  |           |  | |   | |     |
> | KVTC 8x-No Quantization |          1 |  27.5 | 24.3 |   28.2 | 49.1 |     70.8 |
> |              |          2 |  44.8 | 41.1 |   34.2 | 77.6 |     84.9 |
> |              |          4 |  53.3 | 51.9 |   37.5 | 95.9 |     93.0 |
> |              |          8 |  55.9 | 55.3 |   39.7 | 99.2 |     90.7 |
> |              |         16 |  56.0 | 57.1 |   38.8 | 98.8 |     92.8 |
> |              |         32 |  56.9 | 57.5 |   40.1 | 98.6 |     93.0 |
> | KVTC 16x-No Quantization |          1 |   2.5 |  0.2 |   18.4 |  0.0 |      0.5 |
> |              |          2 |  13.9 |  2.3 |   22.1 |  0.7 |     12.9 |
> |              |          4 |  33.3 | 19.3 |   29.1 | 32.0 |     62.7 |
> |              |          8 |  49.6 | 43.4 |   34.4 | 60.7 |     85.8 |
> |              |         16 |  51.6 | 49.1 |   36.0 | 72.7 |     89.0 |
> |              |         32 |  52.0 | 53.8 |   36.9 | 69.2 |     87.0 |
> | KVTC 32x-No Quantization |          1 |   1.1 |  0.1 |   17.8 |  0.0 |      0.1 |
> |              |          2 |   1.5 |  0.2 |   18.1 |  0.0 |      0.2 |
> |              |          4 |   3.9 |  0.9 |   20.0 |  0.0 |      1.6 |
> |              |          8 |  28.0 |  6.4 |   24.7 |  2.6 |     33.0 |
> |              |         16 |  40.1 | 26.0 |   31.2 | 24.6 |     60.4 |
> |              |         32 |  42.9 | 36.8 |   34.2 | 44.4 |     83.0 |
> | KVTC 64x-No Quantization |          1 |   0.9 |  0.3 |   18.0 |  0.0 |      0.1 |
> |              |          2 |   1.4 |  0.1 |   17.7 |  0.0 |      0.1 |
> |              |          4 |   1.5 |  0.1 |   18.1 |  0.0 |      0.1 |
> |              |          8 |   2.4 |  0.6 |   19.4 |  0.0 |      0.2 |
> |              |         16 |  13.9 |  1.9 |   22.1 |  0.0 |     10.0 |
> |              |         32 |  21.8 |  6.5 |   26.7 |  5.1 |     55.5 |
>
> We also perform ablation on separately, compressing either keys or values:
>
> | Cache                          | PCA Layers | GSM8K | MMLU | QASPER | LITM | RULER-VT |
> | ------------------------------ | ---------- | ----- | ---- | ------ | ---- | -------- |
> | *Llama 3.1 8B with KVTC 32x-No Quantization* |            |       |      |        |      |          |
> | Keys                           | 1          | 3.2   | 4.4  | 21.7   | 0.0  | 0.1      |
> | Keys                           | 4          | 25.9  | 27.5 | 25.9   | 0.2  | 2.5      |
> | Keys                           | 8          | 49.9  | 43.6 | 35.2   | 44.5 | 74.9     |
> | Keys                           | 16         | 52.5  | 52.3 | 38.6   | 86.0 | 90.1     |
> | Keys                           | 32         | 53.2  | 56.3 | 38.8   | 96.7 | 97.4     |
> | Values                         | 1          | 2.0   | 0.9  | 18.0   | 0.0  | 2.8      |
> | Values                         | 4          | 28.9  | 29.0 | 30.0   | 15.6 | 68.2     |
> | Values                         | 8          | 40.9  | 41.8 | 34.6   | 53.1 | 76.4     |
> | Values                         | 16         | 49.2  | 45.7 | 37.1   | 82.6 | 88.6     |
> | Values                         | 32         | 48.7  | 48.6 | 38.7   | 81.4 | 89.0     |
>
> We note that both keys and values benefit from the increase in the number of layers (block size) which are concatenated for single PCA, with the global PCA yielding the best results. We also emphasize that many value heads show similarities with heads that are far away in the model (see Figure 2).
>
> [xKV] Chang et al., 2025, xKV: Cross-layer SVD for KV-Cache Compression, 2025

---

> ### Author Response · Authors · 2025-11-21
>
> **W5 Figure 3 is not directly tied to the core technical contributions**
> While the figure does not relate directly to the technical contribution, we consider it relevant for defining the problem in the context of modern LLM serving environments. For instance, one of the use cases for KVTC is extending the lifespan of KV cache. The “Cache-aware Router” shown in Figure 3 is the component which chooses which prefill/decode node should be chosen by its already held KV cache (L147-L148). The Object Storage shown in the Figure might hold caches for longer periods of time, or facilitate their distribution (e.g., collect the cache from the decode node, so it could later be sent to a prefill node for a subsequent round of communication). Finally, the role of KV Cache Manager is tracking caches, moving them between hot/warm/cold tiers, and possibly applying KVTC. We will update the manuscript describing these nuances.
>
> **W6 Why KVTC sometimes outperforms vanilla, and accuracy does not steadily decline with CR?**
> Some of the results reported in Table 2 have a fairly high standard error, as reported in Table 11. However, it might not fully explain why KVTC consequently achieves higher scores on some downstream tasks, notably on GSM8K. We attribute this to the effect reported in [DMS], that compressing KV cache might increase the scores on some tasks by filtering out unimportant information/tokens that make attention calculation slightly noisier. For GSM8K,  compressing the context older than the sliding window of 128 tokens might slightly help the model focus on this window. Conversely, the tasks that require the full, long context, namely LITM and VT, show accuracy drops that increase with CR.
>
> In order to test this hypothesis, we provide additional evaluation in which we calibrate the model either on a balanced dataset, or only on short texts. The results align with our expectations, showing improvements in GSM8K scores at the cost of lower LITM and QASPER scores.
>
> | Model            | CR        |        GSM8K |         MMLU |   QASPER |         LITM |           VT |
> | ---------------- | --------- | -----------: | -----------: | -------: | -----------: | -----------: |
> |  **Mistral NeMo 12B**    |       |           |           |       |          |
> |  vanilla                 |   1   | 61.9 | 64.5 |   38.4 | 99.5 |     99.8 |
> |  12B short and long docs | 51–87 | 61.9 | 61.4 |   38.0 | 95.3 |     98.0 |
> | only short docs          | 50–88 | 63.1 | 63.3 |   32.9 | 93.1 |     98.1 |
>
> [DMS] Łańcucki et al. 2025, Inference-time hyper-scaling with kv cache compression
>
> **W7 Difficult to compare baselines, no common CR**
> Our goal for Table 1 was to provide context for methods in their optimal setting. For instance, GEAR and KIVI are directly tied to data format precision, and their CR cannot be easily adjusted. Since KVTC is nearly Pareto-optimal in comparison to other methods (comparing CR and downstream task scores), equalizing CRs has a lower practical value.
>
> **W8 Cross-domain eval in Figure 8 shows a higher cross-domain error, but no domain-agnostic calibration is provided**
> We emphasize the necessity of a diverse calibration set in Section 3.1, describing the detailed procedure on lines 216-218 of the manuscript. Figure 8 illustrates the consequences of deviating from this best practice by showing the higher reconstruction error that results when calibrating on a non-diverse, in-domain set (e.g., OpenR1 → FineWeb). Conversely, Figure 4 (left) demonstrates that when using a diverse calibration set, the cross-domain reconstruction error is significantly smaller and robustly decreases as the size of the calibration data increases. We will enhance the clarity of this discussion within the main text to better highlight this distinction. For additional results regarding cross-domain calibration we refer to our answer to W1.

---

### Official Review · Reviewer_4EeH · 2025-10-30

**Soundness:** 3
**Presentation:** 3
**Contribution:** 3
**Rating:** 6
**Confidence:** 4

**Summary:**

This paper proposes kvtc, a transform-coding pipeline for KV-cache compression that borrows from classic media codecs: PCA-based decorrelation, dynamic-programming bit-allocation for adaptive quantization, and Deflate entropy coding, with a one-time calibration and no weight changes. It targets the prefill↔decode boundary and multi-turn reuse, also handling practicalities like removing RoPE, skipping a recent-token window, and excluding early sink tokens. Empirically, kvtc attains ~16–20× (and up to ~40×) compression with accuracy within ~1 point of vanilla across GSM8K, MMLU, Qasper, RULER/VT, AIME/LCB, and MATH500, while reducing TTFT by up to 8× and outperforming or matching KIVI/GEAR (quantization), H2O/TOVA (eviction), and xKV/SVDq baselines in most settings.

**Strengths:**

This paper is well written and easy to follow. The paper’s main strengths lie in its practicality, simplicity, and effectiveness. It introduces a lightweight, system-friendly transform-coding approach (kvtc) that achieves up to 20× compression of KV caches with minimal accuracy loss and no model modifications.

**Weaknesses:**

The proposed method leverages the traditional transform-coding framework, achieving high compression ratios with minimal performance degradation. However, its limitation appears to be decompression latency. How does the decompression time compare with other methods listed in Table 2? In Table 4, the authors report only the compression and decompression times of the proposed approach, which seems insufficient for a fair comparison.

**Questions:**

1. In Figure 1, the authors state that $T$ denotes the time dimension. Could the authors clarify what the “time dimension” specifically refers to in this context? Does it correspond to the token dimension or another notion of temporal ordering within the sequence?

2. The authors propose using a calibration set to derive a generalizable transformation matrix $V$. However, an important question remains: how are the calibration samples selected to ensure representativeness across diverse input distributions? Moreover, is the calibration process sensitive to the choice or domain of these samples, and how does performance vary under distribution shift? In comparison to SVD-based methods that compute a separate decomposition for each prompt, how much performance degradation does this one-time calibration introduce?

3. Some results in Table 2 are confusing. The proposed method occasionally outperforms the vanilla model, even under compression. Could the authors provide an explanation or analysis for this unexpected improvement?

---

> ### Author Response · Authors · 2025-11-21
>
> We appreciate Reviewer 4EeH’s taking the time to comment on our paper and would like to thank them for their questions and remarks.
>
>
> **W1 How does decompression latency compare to that of methods in Table 2?**
> Most of the methods in Table 2 do not require decompression. However, in our main intended use case which is extending the lifetime of KV caches, the current decompression latency is low enough for large benefits in practical use. In addition, the latency could be further lowered with a fused decompression kernel, since the building blocks of KVTC (projection, quantization and entropy coding) are memory-bound, and their latency is bound by the time required to read the KV cache from HBM.
>
> Finally, KVTC compression/decompression can be executed on-demand. All KV caches can be stored by default uncompressed in the host memory (CPU DRAM), and compressed using the host CPU only when necessary. With such a strategy, when a KV cache needs to be decompressed, it means that without KVTC it would have been evicted, and repeating prefill would be much more costly than decompression.
>
> **Q1 What does “T” denote in the paper?**
> “T” denotes the temporal/sequence dimension over which the new tokens are appended to key and value caches. KVTC does not compress over this dimension, which is a conscious choice that allows storage of compressed attention pages that have unchanged beginning and ending points.

---

> ### Author Response · Authors · 2025-11-21
>
> **Q2 How are calibration samples selected? Is the calibration process sensitive to the choice or domain? How much accuracy degradation does one-time calibration introduce?**
> We provide information about the selection of calibration samples in Appendix C.2. Briefly speaking, we use a 50/50 mixture of FineWeb (for diversity) and OpenR1Math (for thinking traces), equalizing the number of tokens from documents below 8K (short) and above 8K tokens (long). We observe that simply sampling a document and taking all of its tokens works reasonably well, and therefore follow this approach. We note that using only short documents may boost short-context results (GSM8K) at the cost of long context ones (QASPER, LITM):
>
> | Model            | CR        |        GSM8K |         MMLU |   QASPER |         LITM |           VT |
> | ---------------- | --------- | -----------: | -----------: | -------: | -----------: | -----------: |
> |  **Mistral NeMo 12B**    |       |           |           |       |          |
> |  vanilla                 |   1   | 61.9 | 64.5 |   38.4 | 99.5 |     99.8 |
> |  12B short and long docs | 51–87 | 61.9 | 61.4 |   38.0 | 95.3 |     98.0 |
> | only short docs          | 50–88 | 63.1 | 63.3 |   32.9 | 93.1 |     98.1 |
>
> To check the sensitivity to the domain, we additionally evaluate the performance of KVTC when calibrated on data from code repositories and attach the results below. We observe that using 1-8K token files of either Python or Assembly from [StarCoder] instead of a 50/50 mixture of FineWeb and OpenR1Math data results in performance degradation on GSM8K for compression ratios above 16x. We note that despite a strong domain shift (code vs natural language), the model retains in-context retrieval abilities (LITM, RULER-VT) and for cr $16\times$ obtains scores that are either on par or better than KIVI/GEAR 2-bit. Briefly speaking, we observe relatively high robustness to the calibration domain; however, the best compression results are achieved with a balanced (akin to pre-training) calibration set.
>
> | Method               | Calibration Data     |    CR |  GSM | MMLU | QASPER | LITM | RULER-VT |
> | -------------------- | -------------------- | ----: | ---: | ---: | -----: | ---: | -------: |
> | **Mistral NeMo 12B** |                      |       |      |      |        |      |          |
> | Vanilla              | -                    |     1 | 61.9 | 64.5 |   38.4 | 99.5 |     99.8 |
> | GEAR 2bit            | -                    |     5 | 59.8 | 64.0 |   38.6 | 96.9 |     99.4 |
> | KIVI 2bit            | -                    |     5 | 59.7 | 64.3 |   38.2 | 91.9 |     98.3 |
> | KVTC 16x            | FineWeb + OpenR1Math | 17–20 | 63.5 | 64.7 |   37.2 | 99.6 |     99.7 |
> | KVTC 16x            | Python               | 17–20 | 59.4 | 65.2 |   37.4 | 99.9 |     99.7 |
> | KVTC 16x            | Assembly             | 17–20 | 58.9 | 65.2 |   37.0 | 99.9 |     99.7 |
> | KVTC 32x            | FineWeb + OpenR1Math | 31–43 | 63.1 | 64.4 |   36.1 | 98.1 |     99.2 |
> | KVTC 32x            | Python               | 35–43 | 59.0 | 64.3 |   36.3 | 99.7 |     99.2 |
> | KVTC 32x            | Assembly             | 31–43 | 55.2 | 64.2 |   36.7 | 99.6 |     99.4 |
> | KVTC 64x            | FineWeb + OpenR1Math | 50–88 | 63.1 | 63.3 |   32.9 | 93.1 |     98.1 |
> | KVTC 64x            | Python               | 63–87 | 56.3 | 62.0 |   33.9 | 98.0 |     97.4 |
> | KVTC 64x            | Assembly             | 50–86 | 60.9 | 62.2 |   33.0 | 99.1 |     98.3 |
>
> To check the sensitivity of the process itself we conduct the calibration with 5 different document sets, and provide result along with the standard deviation below:
>
> | Method               |    CR |            GSM8K |             MMLU |           QASPER |             LITM |         RULER-VT |
> | -------------------- | ----: | ---------------: | ---------------: | ---------------: | ---------------: | ---------------: |
> | **Mistral NeMo 12B** |       |                  |                  |                  |                  |                  |
> | Vanilla              |     1 |     $61.9_{1.3}$ |     $64.5_{0.4}$ |           $38.4$ |     $99.5_{0.1}$ |     $99.8_{0.2}$ |
> | KVTC 16x             | 17-20 | $63.5_{\pm 0.9}$ | $64.7_{\pm 0.3}$ | $37.2_{\pm 0.4}$ | $99.6_{\pm 0.3}$ | $99.7_{\pm 0.0}$ |
> | KVTC 32x             | 31-43 | $63.1_{\pm 0.5}$ | $64.4_{\pm 0.6}$ | $36.1_{\pm 0.7}$ | $98.1_{\pm 2.0}$ | $99.2_{\pm 0.3}$ |
> | KVTC 64x             | 50-88 | $63.1_{\pm 1.3}$ | $63.3_{\pm 0.9}$ | $32.9_{\pm 0.7}$ | $93.1_{\pm 3.6}$ | $98.1_{\pm 0.4}$ |

---

> > ### Author Response · Authors · 2025-11-21
> >
> > Below, we attach the brief ablation regarding the per-prompt calibration. We note that while compression of 64K tokens finishes in under half a second (see Table 4 in our paper), the calibration can take minutes on a single H100 GPU (see Table 8 in our paper) . What is more, storing the V matrix for each prompt significantly reduces the compression ratio. We observe that V computed on the prompt can result in poor compression quality of the response, but better compression quality of the prompt. In general, we note that our approach (single longer calibration instead of per-prompt calibration) is much better in terms of compression ratio, speed, and generalization. To speed-up evaluation, we use KVTC without quantization and limit ourselves to two tasks.
> >
> > | Method                     | Calibration Per-Prompt | Simulate Conversation | CR        | GSM8K      | LITM 100              |
> > | -------------------------- | ---------------------- | --------------------- | --------- | ---------- | --------------------- |
> > | KVTC 8x - No Quantization  | False                  | False                 | 7.9–8.0   | 56.8       | 98.8                  |
> > |                            | False                  | True                  | 7.9–8.0   | 56.9       | 98.6                  |
> > |                            | True                   | False                 | 1.0–1.1   | 56.7       | 99.5                  |
> > |                            | True                   | True                  | 1.0–1.1   | 31.4       | 99.5                  |
> > | KVTC 16x - No Quantization | False                  | False                 | 15.4–15.9 | 54.6       | 73.7                  |
> > |                            | False                  | True                  | 15.4–15.9 | 52.0       | 69.2                  |
> > |                            | True                   | False                 | 1.0–2.1   | 56.7       | 99.4                  |
> > |                            | True                   | True                  | 1.0–2.1   | 31.4       | 99.5                  |
> > | KVTC 32x - No Quantization | False                  | False                 | 29.4–31.9 | 48.7       | 49.5                  |
> > |                            | False                  | True                  | 29.5–31.9 | 42.9       | 44.4                  |
> > |                            | True                   | False                 | 1.0–6.2   | 56.7       | 99.4                  |
> > |                            | True                   | True                  | 1.0–6.2   | 31.4       | 99.3                  |
> > | KVTC 64x - No Quantization | False                  | False                 | 54.2–63.4 | 38.6       | 5.3                   |
> > |                            | False                  | True                  | 54.3–63.4 | 21.8       | 5.1                   |
> > |                            | True                   | False                 | 1.3–12.4  | 58.3       | 99.3                  |
> > |                            | True                   | True                  | 1.3–12.4  | 28.4       | 98.8                  |
> >
> > [StarCoder] StarCoder: may the source be with you!, Raymond Li et al. 2023 Transactions on Machine Learning Research

---

> ### Author Response · Authors · 2025-11-21
>
> **Q3 Why KVTC occasionally outperforms vanilla?**
> Some of the results reported in Table 2 have a fairly high standard error, as reported in Table 11. However, it might not explain why KVTC consequently achieves higher scores on some downstream tasks, for instance, GSM8K. We attribute this to the effect, already reported in [DMS], that KV cache compression might increase the scores on some tasks by filtering out unimportant information/tokens that make attention calculation slightly noisier. For GSM8K, we hypothesize that compressing the context older than the sliding window of 128 tokens might slightly help the model focus on this window. Conversely, the tasks that require the full, long context, namely LITM and VT, show accuracy drops that increase with CR. We also note that biasing the calibration data towards short context data improves results on GSM8K at the cost of LITM and QASPER scores (see first table in response to Q2).
>
> [DMS] Łańcucki et al. 2025, Inference-time hyper-scaling with kv cache compression

---

### Official Review · Reviewer_B6EH · 2025-10-30

**Soundness:** 3
**Presentation:** 3
**Contribution:** 3
**Rating:** 6
**Confidence:** 4

**Summary:**

The authors propose a method called KV Transform Coding (KVTC), which compresses the KV cache using a transform-coding pipeline inspired by classical image compression. The proposed approach does not modify model parameters or architectures and can be integrated seamlessly into existing inference systems. Experiments demonstrate compression up to 20$\times$ with negligible accuracy loss across multiple models and benchmarks.

**Strengths:**

1. The KV cache is a key bottleneck for efficient LLM inference. The paper identifies and effectively tackles this real-world challenge.
2. KVTC uses well-established techniques (PCA, quantization, entropy coding) in a novel context. It requires no retraining and can be plugged into existing frameworks.
3. The authors test on diverse LLMs (Llama 3.1, Mistral-NeMo, R1-Distilled Qwen2.5) and datasets (MMLU, GSM8K, RULER, MATH500, etc.). The paper reports improvements in both latency and memory usage.

**Weaknesses:**

1. **Limited algorithmic novelty**:  Transform coding with PCA+quantization is classical, and several SVD/quant methods exist (e.g., SVDq, xKV). The core components are classical in signal processing. The main contribution is an effective adaptation of known techniques, rather than a fundamentally new algorithm.

2. **Dependence on calibration data**: The PCA basis and bit allocation depend on a representative calibration dataset. When model structure changes, recalibration is required. There is limited end-to-end serving results (e.g., cross-node throughput, bandwidth savings, hit-rate improvements, multi-tenant scenarios).

3. **Unfair latency comparison setup**: The reported "8× faster TTFT" compares KVTC against a full recomputation baseline, which is not representative of how production LLM systems operate. A fairer comparison should include: 1) recomputing only the KV cache for new tokens 2) other competitive caching strategies.

4. The experimental protocol seems not align with the primary stated use case (compressing stale caches for storage between conversation turns). This mismatch, coupled with latency benchmarks from a "simple implementation" rather than a high-performance serving framework, makes it difficult to evaluate the true system-level overhead and viability of kvtc in its intended deployment.

5. Some comparison choices could bias CR: CR computed only on compressed tokens (excluding sliding window), while Deflate gains are content-dependent.  The link between Frobenius reconstruction error and downstream accuracy is only partially validated; no perplexity/CE analysis.  More ablations on calibration set composition/size and GQA specifics would help.

6. RoPE “undo” and reapplication details, and cross-layer/head concatenation vs. blockwise streaming claims, could use clearer algorithmic specification.

7. The assumption of a single, global low-rank subspace shared by all concatenated layers and heads is a strong and maybe potentially suboptimal design choice. The paper lacks a crucial ablation study to justify this global approach over more granular alternatives (e.g., per-layer or layer-block PCA), which might yield a better rate-distortion trade-off.

8. SVD-related scheme and the bit allocation applied directly to KV Cache, inherently minimizes the **cache reconstruction loss**($\|\mathbf{K} - \widehat{\mathbf{K}}\|_F^2$), which is Q-agnostic. However, the actual objective is minimizing the **attention output loss**, where Q acts as a dynamic projection (and maybe or maybe not can be approximated by the calibration process). Considering that the theoretical derivation and implementation may be more difficult after the introduction of Q, a brief discussion will make the paper more rigorous.

**Questions:**

- How exactly are RoPE rotations inverted and reapplied for keys during compression/decompression in mixed prefill/decode settings?
- Is the projection matrix V block-structured per layer/head to enable true layer-by-layer decompression, or is selective multiplication by submatrices used? Any accuracy/cost trade-offs?
- Can you report end-to-end CR including the uncompressed sliding window to better reflect practical storage savings?
- Which nvCOMP codec is used (Deflate vs. GDeflate/LZ4/Zstd)? How sensitive are results to codec choice and content?
- How robust is a single global V across domains? Any cross-domain generalization or per-domain calibration vs. global calibration ablation?
- What is the memory/latency overhead of storing and applying V, group scales/shifts, and metadata for large models (e.g., 70B) under pipeline/tensor parallelism?
- Can you provide perplexity/per-token CE vs. CR curves, and a stronger correlation study between reconstruction error and task accuracy?

---

> ### Author Response · Authors · 2025-11-21
>
> We appreciate Reviewer B6EH’s careful reading and helpful comments, which we address below.
>
> **W1 Limited algorithmic novelty**
> While KVTC uses widely known algorithms, we show that SVD can be calculated once, which improves upon the previous work that applies SVD per-prompt. By choosing a simple transform coding scheme, we hope to establish a strong, reproducible foundation for subsequent research and aim to encourage further exploration of other transform coding approaches. The dynamic bit allocation in our algorithm takes advantage of microscaling, which caught attention only recently, as it leads to better quantization of LLMs to 4 bits as shown with MXFP4 and NVFP4 [BR]. In addition, we frame the problem as compression for storage, which adds the constraint of fast decompression as it counts towards TTFT. This area is largely unexplored, but gains in importance with the wide adoption of LLMs.
>
> [BR] Egiazarian et al., 2025, Bridging the Gap Between Promise and Performance for Microscaling FP4 Quantization
>
> **W2/Q5 How robust is PCA across domains? Need more ablations on calibration set choice.**
> To answer the question, we additionally evaluate the performance of KVTC when calibrated on data from the code domain and attach the results below. We observe that using 1-8K token files of either Python or Assembly from [StarCoder] instead of a 50/50 mixture of FineWeb and OpenR1Math data results in performance degradation on GSM8K for compression ratios above 16x. However, we note that despite a strong domain shift (code vs natural language), the model retains in-context retrieval abilities (LITM, RULER-VT) and for CR $16\times$ obtains scores that are either on par or better than KIVI/GEAR 2-bit.
>
> | Method               | Calibration Data     |    CR |  GSM | MMLU | QASPER | LITM | RULER-VT |
> | -------------------- | -------------------- | ----: | ---: | ---: | -----: | ---: | -------: |
> | **Mistral NeMo 12B** |                      |       |      |      |        |      |          |
> | Vanilla              | -                    |     1 | 61.9 | 64.5 |   38.4 | 99.5 |     99.8 |
> | GEAR 2bit            | -                    |     5 | 59.8 | 64.0 |   38.6 | 96.9 |     99.4 |
> | KIVI 2bit            | -                    |     5 | 59.7 | 64.3 |   38.2 | 91.9 |     98.3 |
> | KVTC x 16            | FineWeb + OpenR1Math | 17–20 | 63.5 | 64.7 |   37.2 | 99.6 |     99.7 |
> | KVTC x 16            | Python               | 17–20 | 59.4 | 65.2 |   37.4 | 99.9 |     99.7 |
> | KVTC x 16            | Assembly             | 17–20 | 58.9 | 65.2 |   37.0 | 99.9 |     99.7 |
> | KVTC x 32            | FineWeb + OpenR1Math | 31–43 | 63.1 | 64.4 |   36.1 | 98.1 |     99.2 |
> | KVTC x 32            | Python               | 35–43 | 59.0 | 64.3 |   36.3 | 99.7 |     99.2 |
> | KVTC x 32            | Assembly             | 31–43 | 55.2 | 64.2 |   36.7 | 99.6 |     99.4 |
>
> To be more precise, we observe relatively high robustness to the calibration domain; however, the best compression results are achieved with a balanced (akin to pre-training) calibration set.
> What is more, LITM's random-UUID dictionary further demonstrates that PCA directions learned on the calibration set generalize to inputs with a very different token distribution.
>
> [StarCoder] StarCoder: may the source be with you!, Raymond Li et al. 2023 Transactions on Machine Learning Research

---

> ### Author Response · Authors · 2025-11-21
>
> **W2 Recalibration when model structure changes; limited end-to-end serving results**
> We would like to point out that the calibration completes within minutes on modern GPUs, and could be performed with CPUs. Recalibration would likely constitute a fraction of the cost and time required to alter the weights of the model, or reconfigure how the computation is split over accelerators, for instance, when the number of tensor- or pipeline-parallel instances changes.
> We provide additional end-to-end measurements in a simplified, multi-user scenario. To this end, we have run vLLM with LMCache managing the KV cache on a workload in which, for 64K of initial input tokens, the users ask questions that are between 16 and 100 tokens and receive 100-token answers, with no delays in between conversation turns. We have used Llama 3.3 70B in FP8 precision, split over 2x H100 80GB GPUs using tensor parallelism. LMCache has been configured to use 128GiB of host (CPU DRAM) memory per GPU, which is equivalent to using 1TiB in an 8-GPU server. We adjust the load of the server by changing the number of clients that concurrently hold conversations.
>
> | vLLM num of concurrent clients | Response latency of vanilla (s) | Response latency of KVTC 16x (s) | TTFT of vanilla (s) | TTFT of  KVTC 16x (s) |
> | -----------------------------: | ------------------------------: | -------------------------------: | -------: | -----------------------: |
> |                              1 |                             2.5 |                              2.5 |      0.2 |                      0.2 |
> |                              2 |                             2.9 |                              2.9 |      0.3 |                      0.3 |
> |                              4 |                             8.6 |                              9.1 |      1.2 |                      1.7 |
> |                              6 |                            12.7 |                             13.9 |      2.0 |                      2.7 |
> |                              8 |                            17.3 |                             18.0 |      2.8 |                      3.5 |
> |                             10 |                            21.4 |                             24.2 |      3.4 |                      4.5 |
> |                             12 |                           155.7 |                             27.9 |    136.6 |                      5.6 |
> |                             14 |                           180.9 |                             31.3 |    159.0 |                      6.4 |
> |                             16 |                           208.2 |                             37.1 |    181.6 |                      7.3 |
>
> With 12 or more clients, the dedicated amount of host memory becomes too little to hold the KV caches and forces recomputation, which is reflected in spiking latency. In a more realistic scenario, there would be more concurrent users taking pauses between conversation turns. We note that in this test, the KV cache is compressed with KVTC only for storage in host memory; it does not take advantage of compressing the KV cache for storage in GPU HBM, which would bring additional latency benefits, as its implementation is more involved.
>
> [LMCache] Cheng et al., 2025, LMCache: An Efficient KV Cache Layer for Enterprise-Scale LLM Inference.
> [vLLM] Woosuk Kwon et al., 2023, Efficient Memory Management for Large Language Model Serving with PagedAttention.

---

> ### Author Response · Authors · 2025-11-21
>
> **W3 Unfair latency comparison setup (vs. full recomputation)**
> Before we address this weakness, we would like to point out the fact that the compression could be enabled on demand, i.e., only for long sequences. Furthermore, the complexity of prefill is O(T^2), where T is the sequence length, whereas the complexity of KVTC compression is O(T). Considering a particular example, prefill of new 8k tokens with the existing KV cache for 128k old tokens is more expensive than prefill of 8k context with no old context. However, KVTC compression is contextless, and in both cases will cost the same, much less than prefill. If there are a few new tokens, for instance, 64 new tokens with 128k tokens of old context, KVTC might simply not be enabled, and these 64 tokens either reside in the memory uncompressed or need to be calculated from scratch.
>
> A full recomputation of the prefill phase might happen frequently in the production systems in scenario (1), which we describe in Section 2 (multiple conversation turns). Consider a conversation about a codebase,  prefilled with a very long context, which the user wishes to continue after an hour or more. During this time, the cache has likely been removed entirely from GPU memory and CPU DRAM, due to limited storage space, and needs to be recomputed from scratch or retrieved from cold storage. However, if all caches are compressed 20x, their lifetime in CPU DRAM would increase roughly proportionally.
>
> A comparison with other caching strategies is increasingly difficult, as the gains are strongly dependent on the use case. A measurement of gains should assume the lengths of conversations, inputs, outputs, typical response times, and server load. Moreover, many caching strategies (e.g., caching in pages or organizing the KV cache as a prefix tree) are compatible with KVTC and could bring compounded gains.
>
> **W4 The experimental protocol seems misaligned with the primary stated use case (conversations)**
> The addition of frequent compression/decompression to the standard benchmarks was a conscious effort to show that the compression preserves high accuracy on a wide range of challenging downstream tasks. In practice, the KV cache would be compressed after the decoding phase. Artificially compressing/decompressing the KV cache more frequently every 16 tokens increases the difficulty and is aimed at showing how robust the algorithm is.
>
> Performance-wise, please refer to the answer to W2 for a more realistic evaluation.
>
> **W5/Q3 Calculating CR end-to-end and taking into account the sliding window**
> We note that the following formula can be used to calculate the KVTC compression ratio without Deflate on the entire context:
> $$\mathrm{ReducedContext} = {\frac{\mathrm{ContextLength} - w - s}{\mathrm{cr}}  + w + s}$$
> $$\mathrm{FullCompressionRatio} = \frac{\mathrm{ContextLength}}{\mathrm{ReducedContext}}$$
>
> ​where $w=128$ is the sliding window size, $s=4$ is the number of sink tokens that we skip at the beginning, and
> $\mathrm{cr}$ is the KVTC compression ratio without Deflate (specified in the tables as $\mathrm{kvtc}_{\mathrm{cr}}$)
>
> Therefore for $\mathrm{kvtc}_{\mathrm{16}}$ the compression ratio for context 100K is
> $$\mathrm{ReducedContext} = {\frac{100000 - 128 - 4}{16}  + 128 + 4} = 6373.75$$
> $$\mathrm{FullCompressionRatio} = {\frac{100000}{6373.75}} = 15.7$$
>
> Furthermore, KVTC can be dynamically turned on and off based on the context length.
> ​We note that [KIVI] also uses a sliding window of the same size, whereas [DMS] employs a 2x larger 256 token sliding window. As the effect of the sliding window on the compression ratio approaches zero as the context length increases, in the main tables, we continue to report the compression ratio without the sliding window, for comparability, and following the reviewer's request, we will add an Appendix reporting the compression ratio that includes the sliding window. If the reviewer has any further questions, we would be happy to answer them.
>
> [DMS] Łańcucki et al. 2025, Inference-time hyper-scaling with kv cache compression
> [KIVI] Zirui Liu et al., 2023, Plug-and-play 2-bit kv cache quantization with streaming asymmetric quantization

---

> ### Author Response · Authors · 2025-11-21
>
> **W5.1/Q4 Deflate’s gains are content-dependent. What is the performance of other algorithms? What algorithm is used?**
> In our experiments, we use Deflate. Below, we attach an ablation of the lossless compression algorithms, performed using Mistral Nemo 12B with KVTC 32x. Identity stands for no additional lossless compression (just PCA + DP Quantization; note that we count the omitted sinks into the compression ratio, which is notable for tasks with shorter contexts). We observe that ANS, Deflate, GDeflate, and zStandard improve significantly over Identity in all studied cases. In paper tables, we additionally note the compression ratio of KVTC without Deflate and sinks as $\mathrm{kvtc}_{cr}$.
>
> | Algorithm |   Min–Max | GSM8K | QASPER | LITM | RULER-VT |
> | --------- | --------: | ----: | -----: | ---: | -------: |
> | ANS       | 32.8–37.6 |  32.8 |   37.6 | 36.8 |     37.0 |
> | Bitcomp   | 29.6–32.3 |  29.6 |   32.3 | 32.2 |     32.2 |
> | Deflate   | 34.7–42.9 |  34.7 |   39.5 | 39.7 |     42.9 |
> | GDeflate  | 34.6–42.8 |  34.6 |   39.4 | 39.6 |     42.8 |
> | Identity  | 29.6–32.4 |  29.6 |   32.4 | 32.2 |     32.3 |
> | LZ4       | 29.5–35.5 |  29.5 |   32.4 | 32.7 |     35.5 |
> | Snappy    | 29.4–34.5 |  29.4 |   32.2 | 32.4 |     34.5 |
> | zStandard | 34.6–46.3 |  34.6 |   39.4 | 39.9 |     46.3 |
>
> **W5.2 More ablations on GQA specifics**
> Modern models converge to similar architectures for attention, with GQA being the dominant one at the time of writing. We would like to point out the fact that Qwen2.5 models have a different GQA configuration than Llama 3.1 8B or Mistral models, which is reflected in the lower size of their KV caches in Table 1 in proportion to the number of parameters. In addition, the results reported in [xKV] suggest that SVD will bring similar benefits for Multi-head Latent Attention.
>
> [xKV] Chang et al., 2025, xKV: Cross-layer SVD for KV-Cache Compression, 2025
>
> **W5.3/W8 Optimizing with respect to attention output loss instead of key reconstruction; the Frobenius norm and the downstream accuracy link only partially validated**
> We agree that optimizing an objective that is closer to downstream performance is more promising. We already noted this as a limitation in section 5, lines 415-419: “We note that the reconstruction error based on the Frobenius norm is a proxy for measuring how a compression approach will result in the model’s downstream performance. We briefly explore the correlation between reconstruction error and downstream performance in Appendix C.5”.
> Additionally, in our response to Q7 below, we provide stronger evidence for the correlation between the Frobenius norm reconstruction error, perplexity and downstream accuracy.
>
> Regarding the “attention output loss”, optimization with respect to the Frobenius norm over keys/values reconstruction has optimal substructure, which is a necessary condition for using Dynamic Programming. The bit-budgets q_i independently contribute to the reconstruction error. Optimization with respect to attention output loss would not have this property due to the non-linearities involved in the calculation, preventing us from using DP. In addition, regardless of the optimization algorithm used, multiple repeated evaluations of attention would significantly increase the computational cost.

---

> ### Author Response · Authors · 2025-11-21
>
> **W6/Q1 How is RoPE inverted and re-applied?**
> To revert the RoPE, we note that it is an invertible operation as it boils down to the application of rotation to pairs of key channels. To be more precise, consider the following RoPE code from the transformers implementation of Llama 3:
> ```
> def rotate_half(x):
>     x1 = x[..., : x.shape[-1] // 2]
>     x2 = x[..., x.shape[-1] // 2 :]
>     return torch.cat((-x2, x1), dim=-1)
>
>
> def apply_rotary_pos_emb(q, k, cos, sin, position_ids=None, unsqueeze_dim=1):
>     cos = cos.unsqueeze(unsqueeze_dim)
>     sin = sin.unsqueeze(unsqueeze_dim)
>     q_embed = (q * cos) + (rotate_half(q) * sin)
>     k_embed = (k * cos) + (rotate_half(k) * sin)
>     return q_embed, k_embed
> ```
>
> Application and removal of RoPE come down to invoking the same function with the following arguments:
> ```
> q_rotated, k_rotated = apply_rotary_pos_emb(q, k, cos, sin)
> q_unrotated, k_unrotated = apply_rotary_pos_emb(q_rotated, k_rotated, cos, -sin)
> ```
>
> Below, we explain in detail why this formulation is correct.
> ```
> cos(-x) = cos(x)
> sin(-x) = -sin(x)
> ```
> To be more precise
> ```
> # for i \in [0, half) where half is the number of k channels // 2
> # we omit other dims for clarity
> k_rotated[i] = k[i] * cos - k[i + half] * sin
> k_rotated[i + half] = k[i+half] * cos + k[i] * sin
>
> k_unrotated[i]  = k_rotated[i]*cos + k_rotated[i + half]*sin
> k_unrotated[i]  = (k[i] * cos - k[i + half] * sin)*cos + ( k[i+half] * cos + k[i] * sin)*sin
> k_unrotated[i]  = k[i] * cos*cos - k[i + half] * sin *cos + k[i+half] * cos * sin + k[i] * sin * sin
> k_unrotated[i]  = k[i] * cos*cos + k[i] * sin * sin
> k_unrotated[i]  = k[i] (cos*cos +  sin * sin)
> k_unrotated[i]  = k[i]
>
> k_unrotated[i+ half]  = k_rotated[i + half]*cos - k_rotated[i]*sin
> k_unrotated[i+ half]  = (k[i+half] * cos + k[i] * sin)*cos - (k[i] * cos - k[i + half] * sin)*sin
> k_unrotated[i+ half]  = k[i+half] * cos * cos + k[i] * sin *cos - k[i] * cos * sin + k[i + half] * sin*sin
> k_unrotated[i+ half]  = k[i+half] * cos * cos + k[i + half] * sin*sin
> k_unrotated[i+ half]  = k[i+half] * (cos * cos + sin*sin)
> k_unrotated[i+ half]  = k[i+half]
> ```
> To be more precise in the KVTC pipeline, the RoPE is removed from keys just before compression (note that the same function that applied the RoPE can be used, with the same arguments, except the negated sin values)  and applied directly after decompression (using the function that would apply it originally).
> We also note that to recreate sines and cosines, one needs only the length/positions of the sequence and the per-model RoPE configuration.

---

> ### Author Response · Authors · 2025-11-21
>
> **W7 Missing ablation on block-wise PCA**
> Following the reviewer's advice, we perform an ablation of the number of layers used for PCA, that is the number of consecutive layers concatenated together for PCA calculation (block-size). To better isolate the influence, we run KVTC without dynamic programming quantization (KVTC-No Quantization)) and with a sliding window w=16. We clearly observe the benefits of global PCA vs local separate PCAs, and the limits of quantization-free approach.
> | Method       | PCA Layers | GSM8K | MMLU | QASPER | LITM | RULER-VT |
> | ------------ | ---------: | ----: | ---: | -----: | ---: | -------: |
> | **Llama 3.1 8B**  |           |  | |   | |     |
> | KVTC 8x-No Quantization |          1 |  27.5 | 24.3 |   28.2 | 49.1 |     70.8 |
> |              |          2 |  44.8 | 41.1 |   34.2 | 77.6 |     84.9 |
> |              |          4 |  53.3 | 51.9 |   37.5 | 95.9 |     93.0 |
> |              |          8 |  55.9 | 55.3 |   39.7 | 99.2 |     90.7 |
> |              |         16 |  56.0 | 57.1 |   38.8 | 98.8 |     92.8 |
> |              |         32 |  56.9 | 57.5 |   40.1 | 98.6 |     93.0 |
> | KVTC 16x-No Quantization |          1 |   2.5 |  0.2 |   18.4 |  0.0 |      0.5 |
> |              |          2 |  13.9 |  2.3 |   22.1 |  0.7 |     12.9 |
> |              |          4 |  33.3 | 19.3 |   29.1 | 32.0 |     62.7 |
> |              |          8 |  49.6 | 43.4 |   34.4 | 60.7 |     85.8 |
> |              |         16 |  51.6 | 49.1 |   36.0 | 72.7 |     89.0 |
> |              |         32 |  52.0 | 53.8 |   36.9 | 69.2 |     87.0 |
> | KVTC 32x-No Quantization |          1 |   1.1 |  0.1 |   17.8 |  0.0 |      0.1 |
> |              |          2 |   1.5 |  0.2 |   18.1 |  0.0 |      0.2 |
> |              |          4 |   3.9 |  0.9 |   20.0 |  0.0 |      1.6 |
> |              |          8 |  28.0 |  6.4 |   24.7 |  2.6 |     33.0 |
> |              |         16 |  40.1 | 26.0 |   31.2 | 24.6 |     60.4 |
> |              |         32 |  42.9 | 36.8 |   34.2 | 44.4 |     83.0 |
> | KVTC 64x-No Quantization |          1 |   0.9 |  0.3 |   18.0 |  0.0 |      0.1 |
> |              |          2 |   1.4 |  0.1 |   17.7 |  0.0 |      0.1 |
> |              |          4 |   1.5 |  0.1 |   18.1 |  0.0 |      0.1 |
> |              |          8 |   2.4 |  0.6 |   19.4 |  0.0 |      0.2 |
> |              |         16 |  13.9 |  1.9 |   22.1 |  0.0 |     10.0 |
> |              |         32 |  21.8 |  6.5 |   26.7 |  5.1 |     55.5 |
>
> We also perform ablation on them separately, compressing either keys or values:
> | Cache                          | PCA Layers | GSM8K | MMLU | QASPER | LITM | RULER-VT |
> | ------------------------------ | ---------- | ----- | ---- | ------ | ---- | -------- |
> | *Llama 3.1 8B with KVTC 32x-No Quantization* |            |       |      |        |      |          |
> | Keys                           | 1          | 3.2   | 4.4  | 21.7   | 0.0  | 0.1      |
> | Keys                           | 4          | 25.9  | 27.5 | 25.9   | 0.2  | 2.5      |
> | Keys                           | 8          | 49.9  | 43.6 | 35.2   | 44.5 | 74.9     |
> | Keys                           | 16         | 52.5  | 52.3 | 38.6   | 86.0 | 90.1     |
> | Keys                           | 32         | 53.2  | 56.3 | 38.8   | 96.7 | 97.4     |
> | Values                         | 1          | 2.0   | 0.9  | 18.0   | 0.0  | 2.8      |
> | Values                         | 4          | 28.9  | 29.0 | 30.0   | 15.6 | 68.2     |
> | Values                         | 8          | 40.9  | 41.8 | 34.6   | 53.1 | 76.4     |
> | Values                         | 16         | 49.2  | 45.7 | 37.1   | 82.6 | 88.6     |
> | Values                         | 32         | 48.7  | 48.6 | 38.7   | 81.4 | 89.0     |
>
> We note that both keys and values benefit from the increase in the number of layers (block size) which are concatenated for single PCA, with the global PCA yielding the best results. We will include these additional ablations in the Appendix.
>
> **Q2 What is the structure of the projection matrix V? Are there any accuracy/cost trade-offs?**
> In our experiments, V is a dense matrix. When the KV cache is offloaded to CPU DRAM, the compression might be carried out using the CPU in the background. Decompression layer-by-layer does not require a special structure of V. Consider the reverse projection $V^T\in\mathbb{R}^{r\times 32768}$, which projects the cache from the latent dimension $r$ back to 32768 dimensions - 32 layers with 1024 dimensions each. A contiguous submatrix of $V^T$ of size $r\times 1024$ will suffice for decompression of the cache for a single layer. Naturally, many optimizations might be applied to this process in order to make it more efficient, like enforcing a particular structure of V or applying SVD hierarchically (first for individual layers, then for the concatenation of layers). We leave these as the objective for future work.

---

> ### Author Response · Authors · 2025-11-21
>
> **Q6 What is the memory/latency overhead of storing V and quantization parameters for large models with PP/TP?**
> We count the size of quantization parameters (group sizes, shift and scaling factors) towards the compression ratios in all results presented in the manuscript. As for the projection matrices, for small models that fit on a single GPU, we construct V matrices of size $(p \times r)$, where $p$ is the number of all features (num layers $\times$ num heads $\times$ head dimension) and $r$ the number of dimensions retained by PCA, per the notation on L219-L230. For large models, regardless of the tensor-parallel (TP) and pipeline-parallel (PP) setup, the dimension $p$ will be split evenly over (TP+PP) accelerators, shrinking V on each GPU roughly to size (p/(TP+PP) $\times$ r/(TP+PP)).
>
> For instance, the size of V for keys and values on each GPU serving Llama 3.3 70B with PP=2 TP=1 for KVTC 16x is approximately ~2 GiB at 16-bit precision.
>
> **Q7 Additional perplexity or cross-entropy evaluation and a stronger correlation study between reconstruction error and task accuracy**
> Following the Reviewer's advice, we present a version of Table 8 augmented with per CR reconstruction error and perplexity (on a held-out 50/50 mixture of FineWeb and OpenR1Math data), along with the correlation matrix below.
>
> | Method            | Data  | PCA   | DP      | Error Key   | Error Value | PPL   | CR      | GSM8K        | MMLU         | QASPER | LITM         | VT           | AVG  |
> |-------------------|-------|-------|---------|-------|-------|-------|---------|--------------|--------------|--------|--------------|--------------|------|
> | **Mistral NeMo 12B**  |       |       |         |       |       |       |         |              |              |        |              |              |      |
> | KVTC 32x          | 20K   | 41.3s | 6.5 min | 0.184 | 0.365 | 3.864 | 31-42   | $63.0_{1.3}$ | $63.9_{0.4}$ | $34.8$ | $97.3_{0.2}$ | $98.9_{0.3}$ | 71.6 |
> | KVTC 64x          | 20K   | 41.3s | 5.4 min | 0.222 | 0.440 | 3.993 | 63-87   | $63.9_{1.3}$ | $61.7_{0.4}$ | $31.8$ | $88.8_{0.4}$ | $96.9_{0.5}$ | 68.6 |
> | KVTC 256x         | 20K   | 41.3s | 3.9 min | 0.293 | 0.565 | 4.696 | 148-340 | $59.6_{1.4}$ | $51.5_{0.4}$ | $23.3$ | $10.4_{0.4}$ | $66.8_{1.5}$ | 42.3 |
> | KVTC 32x          | 40K   | 47.8s | 7.2 min | 0.172 | 0.341 | 3.828 | 31-43   | $64.2_{1.3}$ | $63.0_{0.4}$ | $35.3$ | $98.7_{0.1}$ | $99.5_{0.2}$ | 72.1 |
> | KVTC 64x          | 40K   | 47.8s | 6.1 min | 0.212 | 0.421 | 3.958 | 63-88   | $63.7_{1.3}$ | $61.9_{0.4}$ | $32.4$ | $95.7_{0.3}$ | $97.8_{0.5}$ | 70.3 |
> | KVTC 256x         | 40K   | 47.8s | 4.6 min | 0.285 | 0.555 | 4.865 | 148-344 | $58.3_{1.4}$ | $50.1_{0.4}$ | $23.7$ | $9.2_{0.4}$  | $70.6_{1.4}$ | 42.4 |
> | KVTC 32x          | 80K   | 60.4s | 7.2 min | 0.163 | 0.322 | 3.798 | 31-43   | $63.6_{1.3}$ | $63.9_{0.4}$ | $36.0$ | $98.0_{0.2}$ | $99.3_{0.3}$ | 72.2 |
> | KVTC 64x          | 80K   | 60.4s | 6.1 min | 0.204 | 0.405 | 3.934 | 63-88   | $63.3_{1.3}$ | $62.0_{0.4}$ | $32.2$ | $94.5_{0.3}$ | $97.8_{0.5}$ | 70.0 |
> | KVTC 256x         | 80K   | 60.4s | 4.6 min | 0.279 | 0.543 | 4.626 | 148-339 | $60.5_{1.3}$ | $51.7_{0.4}$ | $23.3$ | $13.6_{0.4}$ | $83.7_{1.2}$ | 46.6 |
> | KVTC 32x          | 160K  | 85.6s | 7.2 min | 0.156 | 0.308 | 3.766 | 31-43   | $63.2_{1.3}$ | $64.3_{0.4}$ | $36.6$ | $99.5_{0.1}$ | $99.4_{0.3}$ | 72.6 |
> | KVTC 64x          | 160K  | 85.6s | 6.1 min | 0.198 | 0.394 | 3.901 | 63-87 | $64.6_{1.3}$ | $62.2_{0.4}$ | $32.8$ | $96.9_{0.2}$ | $97.7_{0.5}$ | 70.8 |
> | KVTC 256x         | 160K  | 85.6s | 4.6 min | 0.275 | 0.535 | 4.462 | 148-341 | $61.4_{1.3}$ | $55.5_{0.4}$ | $24.3$ | $34.2_{0.6}$ | $79.5_{1.3}$ | 51.0 |
>
> We note that both key and value reconstruction errors strongly correlate with perplexity and average downstream performance.
>
> | Correlation |    PPL |   AVG Score |
> |:------------|-------:|------------:|
> | Error Key   |  0.959 |      -0.948 |
> | Error Value |  0.953 |      -0.939 |
> | PPL         |  1.000 |      -0.992 |
> | AVG Score   | -0.992 |       1.000 |

---

> > ### Comment · Reviewer_B6EH · 2025-11-27
> > **Thanks for the rebuttal; maintaining positive but cautious rating**
> >
> > I thank the authors for their comprehensive response and the additional experiments, particularly the vLLM integration and the Global PCA ablation. These additions have clarified the implementation details and demonstrated the practical utility of KVTC in specific system configurations (e.g., CPU offloading).
> >
> > However, after carefully considering the new data, I have decided to maintain my score. I have remaining concerns:
> >
> > 1. Algorithmic Novelty Remains Limited:
> >
> > Even with the new results, the core contribution remains the application of classical signal processing techniques (PCA + Quantization) to the KV cache. While the engineering realization is solid, the algorithmic insight is somewhat incremental. The finding that "Global PCA works best" is empirically interesting but lacks a deeper theoretical justification for why a single low-rank subspace suffices across all layers and heads for a 70B model, other than empirical observation on the tested benchmarks.
> >
> > 2. Sensitivity to Calibration Data (OOD Concerns):
> >
> > The authors’ new ablation on calibration data (W2/Q5) confirms my worry. Using code data for calibration resulted in a performance drop on GSM8K (Math) for higher compression ratios. This indicates that the "Global PCA" basis is indeed sensitive to the domain distribution. In real-world deployment, where user queries are highly unpredictable and multimodal, this reliance on a representative calibration set poses a robustness risk compared to calibration-free or adaptive methods.
> >
> > 3. Scope of Impact (Storage vs. Computation):
> >
> > The vLLM experiment clarifies that the primary gain comes from reducing bandwidth pressure when offloading to CPU/Host memory (reducing TTFT from cold storage). While valuable for serving systems that rely heavily on swapping, it does not directly address the GPU memory bottleneck during active computation (batch size limits) in the same way that non-offloading compression methods do. It is effectively a "bandwidth optimization" technique rather than a "compute/memory capacity" optimization for the active path.

---

> > > ### Author Response · Authors · 2025-11-27
> > >
> > > We thank the reviewer for their detailed feedback and for considering our additional experiments. We address the remaining concerns below.
> > >
> > >
> > >
> > > ### 1 Regarding the concerns about algorithmic novelty and deeper theoretical justification.
> > >
> > > In terms of novelty, our approach builds on classical transform coding, yet even such a simple approach yields unprecedented compression ratios from 16x up to 80x, and exposes a smooth accuracy-compression Pareto frontier, far surpassing the other tested methods. Those results are made possible by extending the PCA to more than just one/few layers of the model and dynamic programming guided quantization, which are non-trivial contributions.
> > >
> > > Finally, in a simplified setting (no RoPE, keys=queries), if two heads share attention patterns, their key spaces must lie in similar low-rank subspaces (up to orthogonal transforms, by uniqueness of Gram realizations [Matrix Analysis]). Sharing of attention patterns in transformer-based models was already observed in [MInference] and can be applied to key/query subsets. This suggests that a global PCA basis can capture much of the variation across heads and layers; we will add a precise statement and discussion in the appendix.
> > >
> > > [MInference] Jiang et al. 2024, MInference 1.0: Accelerating pre-filling for long-context LLMs via dynamic sparse attention
> > > [Matrix Analysis] Roger A. Horn and Charles R. Johnson, 2013, Matrix Analysis,  Cambridge University Press, Cambridge, UK, 2 Edition

---

> > > ### Author Response · Authors · 2025-11-27
> > >
> > > ### 2. Regarding the Sensitivity to Calibration Data (OOD Concerns).
> > > We agree with the reviewer, and we will emphasize in the limitations section the potential performance degradation when using OOD data. However,  we note that KVTC with 17-20x compression (16x without Deflate) achieves scores that are either on par or better than calibration-free methods such as KIVI, GEAR, which are limited to 5x compression, and methods such as TOVA and H2O. We also note that KVTC calibrated on Assembly is better on long context tasks, such as Qasper, LITM, RULER-VT, than xKV calibrated using in-domain data. We highlight that phenomenon in the table below by bolding the results of calibration methods that are no more than 1.5 points below the results of the best non-calibration compression/quantization/eviction method.
> > > | Method | Calibration Data | CR | GSM | MMLU | QASPER | LITM | RULER-VT |
> > > | -------------------- | -------------------- | ----: | ---: | ---: | -----: | ---: | -------: |
> > > | Mistral NeMo 12B | | | | | | | |
> > > | Vanilla | - | 1 | 61.9 | 64.5 | 38.4 | 99.5 | 99.8 |
> > > | Calibration Free | | | | | | | |
> > > | GEAR 2bit | - | 5 | 59.8 | 64.0 | 38.6 | 96.9 | 99.4 |
> > > | KIVI 2bit | - | 5 | 59.7 | 64.3 | 38.2 | 91.9 | 98.3 |
> > > | TOVA | - | 8 | 60.3 | 49.0 | 36.0 | 8.7 | 99.6 |
> > > | H2O | - | 8 | 57.0 | 45.4 | 29.5 | 16.2 | 35.2 |
> > > | Requires Calibration | | | | | | | |
> > > | xKV | - | 1-5 | **61.9** | **63.9** | 33.5 | **97.9** | **99.4** |
> > > | KVTC x 16 | FineWeb + OpenR1Math | 17–20 | **63.5** | **64.7** | **37.2** | **99.6** | **99.7** |
> > > | KVTC x 16 | Python | 17–20 | **59.4** | **65.2** | **37.4** | **99.9** | **99.7** |
> > > | KVTC x 16 | Assembly | 17–20 | **58.9** | **65.2** | 37.0 | **99.9** | **99.7** |
> > >
> > >
> > >
> > > In terms of KVTC being less robust than calibration-free methods, the experimental results show a reverse tendency. KVTC is generally better in long context cases than the tested methods. To be more precise, we observe that only KVTC 18-20x  and GEAR 5x  average scores stay within 1 point from Vanilla, whereas KIVI 5x  and KVTC 35-39x stay within 1.5 points from Vanilla. Additionally, we note that KVTC x16 > xKV, despite xKV performing per-prompt calibration. What is more, we note that in  a lot of use cases (e.g., coding assistants), users operate within more predictable domains.
> > >
> > > | Task | Vanilla | GEAR | KIVI | H2O | TOVA | xKV | KVTC 16x | KVTC 32x | KVTC 64x |
> > > | ---- | --------------- | --------------- | ------------ | ------------ | ------------ | ------------ | --------------- | --------------- | --------------- |
> > > | **Llama 3.1 8B** | | | | | | | | | |
> > > | CR | 1 | 5 | 5 | 8 | 8 | 4-6 | 18-20 | 35-39 | 62-78 |
> > > | 2WQA | $40.8$ | $40.7$ | $42.1$ | $38.5$ | $41.3$ | $39.5$ | $40.3$ | $40.0$ | $40.6$ |
> > > | MFQA | $50.3$ | $49.6$ | $50.1$ | $40.8$ | $50.4$ | $48.7$ | $51.1$ | $49.5$ | $50.2$ |
> > > | MQUE | $33.8$ | $33.8$ | $31.7$ | $32.4$ | $32.6$ | $31.5$ | $33.7$ | $33.9$ | $34.1$ |
> > > | QMS | $26.9$ | $27.2$ | $25.7$ | $25.4$ | $25.4$ | $24.4$ | $26.4$ | $25.9$ | $25.2$ |
> > > | SAMS | $47.3$ | $47.2$ | $45.8$ | $46.8$ | $46.1$ | $45.7$ | $47.0$ | $46.6$ | $45.6$ |
> > > | CWE | $94.7$ | $94.0$ | $91.0$ | $64.9$ | $76.5$ | $68.9$ | $92.4$ | $90.7$ | $88.0$ |
> > > | FWE | $92.1$ | $91.9$ | $89.9$ | $75.7$ | $69.5$ | $88.6$ | $89.0$ | $88.1$ | $83.3$ |
> > > | NIAH | $100$ | $100$ | $100$ | $6.2$ | $99.6$ | $99.8$ | $100$ | $99.8$ | $99.6$ |
> > > | HPQA | $57.2$ | $56.8$ | $57.2$ | $48.8$ | $54.8$ | $56.2$ | $57.2$ | $57.2$ | $55.8$ |
> > > | SQA | $55.7$ | $55.7$ | $54.0$ | $40.2$ | $51.3$ | $53.5$ | $55.2$ | $53.1$ | $53.8$ |
> > > | AVG | $\mathbf{59.9}$ | $\mathbf{59.7}$ | $\mathbf{58.8}$ | $42.0$ | $54.8$ | $55.7$ | $\mathbf{59.2}$ | $\mathbf{58.5}$ | $57.6$ |
> > >
> > > LongBench: 2WikiMultiHopQA (2WQA), MultiFieldQA (MFQA), MuSiQue (MQUE), QMSum (QMS), SAMSum (SAMS)
> > > RULER: Common/Frequent Words Extraction (CWE/FWE), Needle In A Haystack (NIAH), HotPotQA (HPQA), SQuAD (SQA).
> > >
> > > [2WikiMultihopQA], Ho et al., 2020, Constructing a multi-hop qa dataset for comprehensive evaluation of reasoning steps
> > > [HotpotQA], Yang et al., 2018, HotpotQA: A Dataset for Diverse, Explainable Multi-hop Question Answering
> > > [LongBench] Bai et al. 2023, LongBench: A Bilingual, Multitask Benchmark for Long Context Understanding
> > > [MuSiQue], Trivedi et al., 2022, Multihop questions via single-hop question composition.
> > > [NIAH], Kamradt, 2023, Needle In A Haystack - pressure testing LLMs
> > > [QMSum], Zhong et al., 2021, Qmsum: A new benchmark for query-based multi-domain meeting summarization
> > > [Quest], Tang et al., 2024, Quest: Query-Aware Sparsity for Efficient Long-Context LLM Inference
> > > [RULER] Yang et al., 2018, RULER: What's the Real Context Size of Your Long-Context Language Models?
> > > [SAMSum], Gliwa et al., 2019, Samsum corpus: A human annotated dialogue dataset for abstractive summarization.
> > > [SQuAD], Rajpurkar et al., 2018, Know what you don’t know: Unanswerable questions for SQuAD

---

> > > ### Author Response · Authors · 2025-11-27
> > >
> > > ### 3. Regarding scope of impact (Storage vs. Computation)
> > > In principle, KVTC can be combined with eviction methods like DMS or TOVA for compounded benefits due to their different objectives. Such methods focus solely on reducing the size of the KV cache during inference in order to free up GPU memory and increase the concurrency.
> > >
> > > We agree with the reviewer that a method that also improves decoding performance would be more beneficial to the community, which we already note in the limitations section and will further expand in the camera-ready version.
> > >
> > > [DMS] Łańcucki et al. 2025, Inference-time hyper-scaling with kv cache compression

---

### Official Review · Reviewer_rALC · 2025-11-01

**Soundness:** 3
**Presentation:** 2
**Contribution:** 3
**Rating:** 6
**Confidence:** 3

**Summary:**

This paper introduces kvtc, a lightweight transform coder designed to compress KV cache for efficient storage in LLM inference. Drawing on classical media compression techniques, kvtc integrates PCA-based feature decorrelation, adaptive quantization, and entropy coding, requiring only brief initial calibration without modifying model parameters . It achieves up to 20× compression ratio on KV cache of models like Llama 3.1, Mistral NeMo, and Qwen 2.5 R1, while maintaining reasoning and long-context accuracy across benchmarks such as GSM8K, MMLU, and LiveCodeBench.

**Strengths:**

+ The paper presents an interesting approach by using a transform-coding framework for KV cache compression. Experiments demonstrate that this method can maintain model performance even with a high compression rate.

+ The ablation experiments are very thorough.

**Weaknesses:**

+ The paper uses a limited number of evaluation benchmarks. Recent related works typically conduct comprehensive experiments on benchmarks like LongBench and RULER, whereas this paper only uses one dataset from each (Qasper and VT).

+ The "Related Work" section should also include content related to transform coding, as it is central to the proposed approach in this paper.

**Questions:**

See above

---

> ### Author Response · Authors · 2025-11-21
>
> We appreciate Reviewer rALC’s taking the time to comment on our paper and would like to thank them for their questions.
>
> **W1 Limited number of benchmarks (LongBench, RULER)**
> We provide additional evaluation of KVTC on tasks from LongBench and RULER using Llama 3.1 8B. The additional tasks include 2WikiMultihopQA, MuSiQue, QMSum, SAMSum, Needle In A Haystack (NIAH), Common/Frequent words extraction (CWE/FWE), HotpotQA, and SQuAD.
>
> | Task   | Vanilla | KVTC 16x | KVTC 32x | KVTC 64x |
> |--------------|--------:|-----------------------:|------------------------:|-----------------------:|
> | CR           | 1.0     | [19;20]               | [39;39]               | [75;78]           |
> | 2WIKIMQA     | 40.8    | 40.3                  | 40.0                  | 40.6                  |
> | MUSIQUE      | 33.8    | 33.7                  | 33.9                  | 34.1                  |
> | QMSUM        | 26.9    | 26.4                  | 25.9                  | 25.2                  |
> | SAMSUM       | 47.3    | 47.0                  | 46.6                  | 45.6                  |
> | CWE          | 94.7    | 92.4                  | 90.7                  | 88.0                  |
> | FWE          | 92.1    | 89.0                  | 88.1                  | 83.3                  |
> | NIAH         | 100.0   | 100.0                 | 99.8                  | 99.6                  |
> | HOTPOTQA     | 57.2    | 57.2                  | 57.2                  | 55.8                  |
> | SQUADQA      | 55.7    | 55.2                  | 53.1                  | 53.8                  |
>
> The accuracy of KVTC remains high across the compression rates, with the largest accuracy drops recorded on the CWE and FWE tasks.
>
> [2WikiMultihopQA], Ho et al., 2020, Constructing a multi-hop qa dataset for comprehensive evaluation of reasoning steps
> [HotpotQA], Yang et al., 2018, HotpotQA: A Dataset for Diverse, Explainable Multi-hop Question Answering
> [LongBench] Bai et al. 2023, LongBench: A Bilingual, Multitask Benchmark for Long Context Understanding
> [MuSiQue], Trivedi et al., 2022, Multihop questions via single-hop question composition.
> [NIAH], Kamradt, 2023, Needle In A Haystack - pressure testing LLMs
> [QMSum], Zhong et al., 2021, Qmsum: A new benchmark for query-based multi-domain meeting summarization
> [RULER] Hsieh et al., 2024, RULER: What's the Real Context Size of Your Long-Context Language Models?
> [SAMSum], Gliwa et al., 2019, Samsum corpus: A human annotated dialogue dataset for abstractive summarization.
> [SQuAD], Rajpurkar et al., 2018, Know what you don’t know: Unanswerable questions for SQuAD
>
>
> **W2 Related works on transform coding are missing**
> We will  extend the Related Work with a section dedicated to transform coding with references to seminal works. We will also add references to works that employ transform coding for compressing the context of deep neural networks, in particular  for model compression [TQ, CAT], or more recently, for re-purposing hardware video codecs for tensor compression [LLM.265], with KV cache compression reaching up to 5.5x.
>
> [TQ] Young et al., 2021, Transform Quantization for CNN Compression
> [CAT] Baskin et al., 2021, CAT: Compression-Aware Training for bandwidth reduction
> [LLM.265] Xu et al., 2025, LLM.265: Video Codecs are Secretly Tensor Codecs

---

> ### Author Response · Authors · 2025-11-27
> **Extended response to W1**
>
> Since our initial response to W1, we have managed to extend the set of evaluation methods, tasks, and models. We attach the results in the table below, noting that only KVTC 18-20x (16x without Deflate) and GEAR (cr 5x) are able to stay within 1 point from the average (AVG) Vanilla score. We will attach those results in the appendix.
> | Task | Vanilla | GEAR | KIVI | H2O | TOVA | xKV | KVTC 16x | KVTC 32x | KVTC 64x |
> | ---- | --------------- | --------------- | ------------ | ------------ | ------------ | ------------ | --------------- | --------------- | --------------- |
> | **Llama 3.1 8B** | | | | | | | | | |
> | CR | 1 | 5 | 5 | 8 | 8 | 4-6 | 18-20 | 35-39 | 62-78 |
> | 2WQA | $40.8$ | $40.7$ | $42.1$ | $38.5$ | $41.3$ | $39.5$ | $40.3$ | $40.0$ | $40.6$ |
> | MFQA | $50.3$ | $49.6$ | $50.1$ | $40.8$ | $50.4$ | $48.7$ | $51.1$ | $49.5$ | $50.2$ |
> | MQUE | $33.8$ | $33.8$ | $31.7$ | $32.4$ | $32.6$ | $31.5$ | $33.7$ | $33.9$ | $34.1$ |
> | QMS | $26.9$ | $27.2$ | $25.7$ | $25.4$ | $25.4$ | $24.4$ | $26.4$ | $25.9$ | $25.2$ |
> | SAMS | $47.3$ | $47.2$ | $45.8$ | $46.8$ | $46.1$ | $45.7$ | $47.0$ | $46.6$ | $45.6$ |
> | CWE | $94.7$ | $94.0$ | $91.0$ | $64.9$ | $76.5$ | $68.9$ | $92.4$ | $90.7$ | $88.0$ |
> | FWE | $92.1$ | $91.9$ | $89.9$ | $75.7$ | $69.5$ | $88.6$ | $89.0$ | $88.1$ | $83.3$ |
> | NIAH | $100$ | $100$ | $100$ | $6.2$ | $99.6$ | $99.8$ | $100$ | $99.8$ | $99.6$ |
> | HPQA | $57.2$ | $56.8$ | $57.2$ | $48.8$ | $54.8$ | $56.2$ | $57.2$ | $57.2$ | $55.8$ |
> | SQA | $55.7$ | $55.7$ | $54.0$ | $40.2$ | $51.3$ | $53.5$ | $55.2$ | $53.1$ | $53.8$ |
> | AVG | $\mathbf{59.9}$ | $\mathbf{59.7}$ | $58.8$ | $42.0$ | $54.8$ | $55.7$ | $\mathbf{59.2}$ | $58.5$ | $57.6$ |
> | **MN-Minitron  8B** | | | | | | | | | |
> | CR | 1 | 5 | 5 | 8 | 8 | 4-6 | 19 | 39-40 | 78-80 |
> | 2WQA | $45.8$ | $45.3$ | $45.4$ | $44.6$| $45.5$| $47.1$ | $46.2$ | $46.4$ | $45.1$ |
> | MFQA | $42.6$ | $42.2$ | $43.0$ | $34.8$| $40.3$| $41.6$ | $42.6$ | $43.4$ | $43.2$ |
> | MQUE | $27.4$ | $26.8$ | $26.2$ | $25.6$| $26.0$| $26.8$ | $27.1$ | $26.6$ | $26.3$ |
> | QMS | $23.4$ | $23.0$ | $23.2$ | $21.9$| $22.5$| $21.8$ | $23.1$ | $22.7$ | $22.6$ |
> | SAMS | $36.1$ | $36.5$ | $36.8$ | $36.7$| $36.1$| $34.7$ | $35.9$ | $35.9$ | $35.8$ |
> | CWE | $92.4$ | $90.5$ | $87.1$ | $64.7$| $66.4$| $69.7$ | $85.3$ | $79.5$ | $75.2$ |
> | FWE | $86.2$ | $86.5$ | $85.8$ | $70.3$| $73.7$| $85.4$ | $86.3$ | $83.1$ | $81.0$ |
> | NIAH | $100$ | $100$ | $99.8$ | $6.0$ | $99.8$| $97.6$ | $100$ | $100$ | $100$ |
> | HPQA | $62.0$ | $63.2$ | $58.4$ | $49.0$| $57.8$| $56.6$ | $62.2$ | $55.8$ | $54.6$ |
> | SQA | $64.9$ | $65.5$ | $62.4$ | $48.0$| $62.7$| $63.9$ | $64.7$ | $62.5$ | $62.2$ |
> | AVG | $\mathbf{58.1}$ | $\mathbf{58.0}$ | $56.8$ | $40.2$ | $53.1$ | $54.5$ | $\mathbf{57.3}$ | $55.6$ | $54.6$ |
>
> LongBench (1-shot): 2WikiMultiHopQA (2WQA), MultiFieldQA (MFQA), MuSiQue (MQUE), QMSum (QMS), SAMSum (SAMS)
> RULER (0-shot): Common/Frequent Words Extraction (CWE/FWE), Needle In A Haystack (NIAH), HotPotQA (HPQA), SQuAD (SQA).

---

### Meta-Review · Area_Chair_QPjA · 2026-01-07

**Summary:**

This paper tries to solve an important problem, while the main concern from the reviewers is based on some detailed concerns about the experimental configuration and results. The authors have added a clear explanation and solid additional experimental results to improve their submission. Overall, I would expect the concerns could be addressed, which leads to an acceptance of the paper.

**Reviewer Concerns:**

Solid and detailed additional experimental results would be sufficient to address the critical concerns from the reviewers.

**Reviewer Scores:**

I would expect Reviewer rALC, 4EeH, jZ1g to increase the score considering the additional experimental results.

---

### Decision · Program_Chairs · 2026-01-26

Accept (Poster)